# Dynamic savanna burning emission factors based on satellite data using a machine learning approach

Roland Vernooij[1], Tom Eames[1], Jeremy Russell-Smith[2,3], Cameron Yates[2,3], Robin Beatty[3,4], Jay Evans[2,3], Andrew Edwards[2,3], Natasha Ribeiro[5], Martin Wooster[6,7], Tercia Strydom[8], Marcos Vinicius Giongo Alves[9], Marco Assis Borges[10], Máximo Menezes Costa[10], Ana Carolina Sena Barradas[10], Dave van Wees[1], Guido R. Van der Werf[1]

[1]Department of Earth Sciences, Faculty of Science, Vrije Universiteit Amsterdam, Amsterdam, The Netherlands
[2]Darwin Centre for Bushfire Research, Charles Darwin University, Darwin, 0909, Northern Territory, Australia
[3]International Savanna Fire Management Initiative (ISFMI), New South Wales, Australia
[4]321Fire, Lda, Inhambane, Mozambique
[5]Faculty of Agronomy and Forest Engineering, Eduardo Mondlane University, Maputo, Mozambique
[6]King's College London, Environmental Monitoring and Modelling Research Group, Department of Geography, UK
[7]National Centre for Earth Observation (NERC), UK
[8]South African National Parks (SANParks), Kruger National Park, South Africa
[9]Center for Environmental Monitoring and Fire Management (CEMAF), Federal University of Tocantins, Gurupi, Brazil
[10]Chico Mendes institute for Conservation of Biodiversity (ICMBio), Rio da Conceição, Brazil

Correspondence to: Roland Vernooij (r.vernooij@vu.nl)

**Abstract.** Landscape fires, predominantly in the frequently burning global savannas, are a substantial source of greenhouse gases and aerosols. The impact of these fires on the atmospheric composition is partially determined by the chemical breakup of the constituents of the fuel into individual emitted chemical species, which is described by emission factors (EFs). These EFs are known to be dependent on, amongst other things, the type of fuel consumed, the moisture content of the fuel and the meteorological conditions during the fire, indicating that savanna EFs are temporally and spatially dynamic. Global emission inventories, however, rely on static biome-averaged EFs which makes them ill-suited for the estimation of regional biomass burning (BB) emissions and for capturing the effects of shifts in fire regimes. In this study we explore the main drivers of EF-variability within the savanna biome and assess which geospatial proxies can be used to estimate dynamic EFs for global emission inventories. We made over 4500 bag measurements of $CO_2$, $CO$, $CH_4$ and $N_2O$ EFs using an unmanned aerial system (UAS), and also measured fuel parameters and fire severity proxies during 129 individual fires. The measurements cover a variety of savanna ecosystems under different seasonal conditions, sampled over the course of six fire seasons between 2017 and 2022. We complemented our own data with EFs from 85 fires with locations and dates provided in the literature. Based on the locations, dates and time of the fires we retrieved a variety of fuel-, weather- and fire severity proxies (i.e. possible predictors) using globally available satellite and reanalysis data. We then trained random forest (RF) regressors to estimate EFs for $CO_2$, $CO$, $CH_4$ and $N_2O$ at a spatial resolution of 0.25° and a monthly timestep. Using these modelled EFs, we calculated their spatiotemporal impact on BB emission estimates over the 2002–2016 period using the Global Fire Emissions Database version 4 with small fires (GFED4s). We found that the most important field indicators for the EFs of $CO_2$, $CO$ and $CH_4$ were tree cover density, fuel moisture content and the grass to litter ratio. The grass to litter ratio and the nitrogen to

carbon ratio were important indicators for $N_2O$ EFs. RF models using satellite observations performed well for the prediction of EF variability in the measured fires with out-of-sample correlation coefficients between 0.80 and 0.99, reducing the error between measured and modelled EFs by 60–85% compared to using the static biome average. Using dynamic EFs, total global savanna emission estimates for 2002–2016 were 1.8% higher for CO while $CO_2$, $CH_4$ and $N_2O$ emissions were respectively 0.2%, 5% and 18% lower compared to GFED4s. On a regional scale we found a spatial redistribution compared to GFED4s with higher CO, $CH_4$ and $N_2O$ EFs in mesic regions and lower ones in xeric regions. Over the course of the fire season, drying resulted in gradually lower EFs of these species. Relatively speaking, the trend was stronger in open savannas than in woodlands where towards the end of the fire season they increased again. Contrary to the minor impact on annual average savanna fire emissions, the model predicts localized deviations from static averages of the EFs of CO, $CH_4$ and $N_2O$ exceeding 60% under seasonal conditions.

# 1 Introduction

Landscape fires emit substantial amounts of gases, including the greenhouse gases $CO_2$, $CH_4$, and $N_2O$ which affect the Earth's climate. To quantify the impact of these fire emissions, and track the role of fire in the biogeochemical system, fire emission inventories like the Global Fire Emissions Database (GFED, van der Werf et al., 2017) and the Global Fire Assimilation System (GFAS, Kaiser et al., 2012) use satellite observations to monitor global landscape fires. They estimate that, due to their high burning frequency, savannas account for roughly 60% of the gross (i.e. not considering regrowth) global carbon emissions from biomass burning (BB). The impact of fire emissions on atmospheric radiative forcing is strongly dependent on the partitioning of consumed biomass into individual emission species, which in part depends on the combustion efficiency (often simplified as the $CO_2$ emissions divided by the combined $CO_2$ and CO emissions, referred to as the modified combustion efficiency or MCE) during the fire. For this partitioning, inventories currently use biome-specific emission factors (EFs), expressed in grams of a molecule emitted for each kilogram of dry matter (DM) burned. However, measurements from both laboratory and landscape fires indicate that important drivers of fire intensity and combustion efficiency, e.g. the moisture content of the fuel (Chen et al., 2010) and the curing state of grasses (Korontzi et al., 2003), are seasonal and that therefore EFs are both spatially and temporally dynamic.

Earlier studies targeted a most representative EF for individual biomes. This single value was based on averaging numerous usually randomly sampled fires mostly from aircraft at the peak of fire season in the most active areas. These sophisticated measurements revealed much about the species that are emitted from fires but there is little opportunity for detailed measurements of the actual fire in this approach. Although they quantify overall variability (as summarized in for example Akagi et al., 2011 and Andreae, 2019), to date we cannot quantify how specific factors such as moisture content impact EFs (van Leeuwen and van der Werf, 2011). Thus, current global inventories are not designed to quantify any variation in emissions at local nor temporal scales. This results, for example, in the same EFs being assumed for a savanna woodland and an open grassland. Using historic averages also means that EFs do not dynamically change while fire regimes, weather patterns and environmental burning conditions can shift as a result of climate change or human interaction. One additional field of research that requires a better understanding of spatiotemporal dynamics involves fire management strategies in savannas to reduce fire-related greenhouse gas emissions, with the aim of mitigating climate change. Over the past decennium,

significant efforts have been directed at shifting the temporal patterns of savanna fire regimes in order to make them more sustainable and abate greenhouse gas emissions (e.g. Russell-Smith et al., 2013; Schmidt et al., 2018). EFs used for the accreditation of such projects currently assume a dichotomy of early- and late dry season averages, determined by a cut-off date. However, as discussed by Laris (2021), the fuel and meteorological

conditions thought to drive EFs vary more gradually over the season and are subjected to substantial inter-annual and spatial variability. Incorporating spatiotemporal variability in inventories makes emission inventories more dynamic and better equipped for assessing seasonal fluctuations.

Over the past six years (2017-2022), a series of savanna burning experiments measuring EFs using unmanned

aerial systems (UAS) has resulted in a large amount of new data with broad spatiotemporal coverage (e.g. Vernooij et al., 2021, 2022; Russell-Smith et al., 2021). While lacking the extensive species coverage and precision of instruments found in advanced aircraft campaigns, these UAS measurements can effectively focus on particular vegetation types, facilitating the connection between ground conditions and emissions. In this study we describe the variability in over 4500 individual bag-measured EFs of $CO_2$, CO, $CH_4$ and $N_2O$ covering 129 fires. Combined

with the EFs from fires already reported in literature, these new EF measurements allow us to study the variability in BB EFs in more detail by using unexplored non-linear statistical methods like decision-tree-based machine learning algorithms. The non-linear nature of these models makes them suitable to quantify distinctive dynamics under different conditions in complex natural processes such as landscape fires. This approach does require large datasets for training and validation which were not available until now. We first determine the dominant drivers

of EF variability based on field measurements and then apply random forest regression methods to estimate dynamic EFs for the abovementioned species using globally available satellite data and geospatial reanalysis data. Depending on the application, these dynamic EFs can be computed at various spatiotemporal resolutions, limited by the resolution of the underlying features (i.e. starting from 500-meter and with hourly timesteps). Finally, we use GFED4s, in combination with the dynamic EFs –computed on a monthly basis at 0.25°– to estimate the

emission dynamics over the 2002-2016 period.

## 2  Methods

The main objectives of this study are: (1) to identify the drivers of EF variability in the savanna biome and (2) to implement this variability into global emission inventories and assess the implications of using dynamic EFs

instead of static ones. The first objective requires a large dataset of EFs and a thorough assessment of a wide range of possible drivers, including direct field measurements of vegetation composition, meteorological conditions and fire intensity dynamics. This is described in section 2.1. The second objective requires a more globalized approach which allows BB EFs to be predicted based on satellite and reanalysis data with broad spatiotemporal coverage, see sections 2.2 and 2.3.

## 2.1    Field measurements

### 2.1.1    Measurement setup

Using a UAS-mounted sampling system we measured BB EFs of $CO_2$, CO, $CH_4$ and $N_2O$ in fresh smoke during savanna fires following the methodology described by Vernooij et al. (2021, 2022). Fires were lit with the aim of being representative of early dry season (EDS, often prescribed) fires and late dry season (LDS) non-prescribed fires. Although some backing fires were sampled during the initial phase of the fires, the majority of samples were obtained from the faster heading fires, which consumed most of the biomass. Fire sizes generally ranged between 2 to 10 hectares based on UAS drone imagery described by Eames et al. (2021), with exceptions of some fires that would not light and conversely, some fires that burned several hundred hectares. In the EDS, fire size was primarily limited by environmental conditions and fires ceased burning as humidity increased overnight whereas in the LDS, fire size was confined by low-fuel areas like burn scars, roads and prepared fire breaks. Particularly in the LDS, this means a limited fire size does not necessarily indicate limited fire intensity. Emissions were sampled at altitudes between 5−50 m depending on flame height for a duration of 35 seconds, resulting in 0.7 litres per gas sample. On average, we took 35 samples per fire. The sampling methodology involved taking samples from a fire passing a certain point –while correcting for wind direction and severity– until no more visual smoke passed the drone anymore. From earlier work (Vernooij et al., 2022a), where we compared the average of these measurements to results using continuous measurements taken at a mast, we have some confidence in the fidelity of this approach. Within 12 hours, the samples were measured using cavity-ringdown spectroscopy for atmospheric mixing ratios of $CO_2$ and $CH_4$ (Los Gatos Research, Microportable gas analyser), and CO and $N_2O$ (Aeris Technologies, Pico series). We calculated EFs using the carbon mass balance method (Ward and Radke, 1993), using ground measurements of the weighted average (WA) carbon content of the combusted fuel and emissions of $CO_2$, CO, $CH_4$ and $N_2O$. The carbon emitted in non-methane hydrocarbons (NMHC) and particulates was estimated based on the linear relations with EFs of CO (for particulates) and $CH_4$ (for NMHCs), which were derived from previous savanna literature (Andreae, 2019; Vernooij et al., 2022). EFs of $N_2O$ were calculated using $CO_2$ as the co-emitted carbonaceous reference species.

### 2.1.2    Sample coverage and literature studies

The dataset obtained using the abovementioned UAS methodology includes both previously published data collected in Mozambique, South Africa, and Brazil (Russell-Smith et al., 2021; Vernooij et al., 2021, 2022) and new measurements from xeric and mesic savannas in Botswana, Zambia and Australia, measured during the fire seasons of 2021 and 2022. The measurements cover three continents and the full length of the dry season, ranging from early dry season (EDS) campaigns in which fuel conditions sometimes prevented successful ignition to late dry season (LDS) campaigns with high-intensity fires. The 129 fires that we measured using the abovementioned methodology were supplemented with 85 previous savanna fires for which EFs of the measured species were reported in the updated database by Andreae (2019). This literature compilation only includes samples taken within minutes after emission to avoid significant chemical changes during atmospheric aging. For the comparison with geospatial data, we only included fires for which the fire date and coordinates were provided, a prerequisite to get relevant satellite features. These criteria mean that laboratory studies, satellite studies covering wider regions, and most aircraft campaigns were excluded. Fig. 1 provides an overview of the UAS (red for previously published and orange for our new measurements), and literature (blue) sample locations included in the study.

### 2.1.3    Fuel measurements

During more recent fieldwork campaigns, we not only measured EFs but also other parameters including fuel characteristics and fire severity indicators. Before the fire, we collected fuel load and fuel composition from various classes (e.g., grass, litter, coarse woody debris, shrubs and trees) and meteorological parameters. After the fire, we revisited the plots and recorded the combustion completeness of various fuel classes as well as fire intensity proxies (e.g. patchiness of the fire, and scorch and char heights) following the methodology outlined by Eames et al. (2021) and Russell-Smith et al. (2020). Table 1 lists the individual UAS EF-measurement campaigns, and whether fuel was collected following the abovementioned methodology. Fires were lit on the windward side of the plot and generally burned through 2-6 individual randomly scattered 50×10-meter fuel transects covering the fuel of a homogenous vegetation type and time since the last fire. We took the average of the affected fuel transects as the fire-averaged value, to correspond to the fire-averaged EF which is calculated over all the bag samples taken from that specific fire. Although the measurements were linearly correlated using the calibration bags for the individual fires, the standard deviations between the calibration samples were 2.58% for $CO_2$, 7.06% for CO, 2.32% for $CH_4$ and 4.04% for $N_2O$, indicating larger measurement uncertainties than reported by the manufacturers, which possibly arises from the bag methodology. The difference in the mean calibration value compared to the calibration gasses was -4.75% for $CO_2$, -1.32% for CO, -3.97% for $CH_4$ and -1.28% for $N_2O$.

## 2.2    Regression analysis

Field measurements provide the most accurate description of the vegetation conditions during the fire and yielded the most reliable insights in the drivers of EF dynamics. However, these measurements are sparse and thus unsuitable for spatiotemporal extrapolation. We therefore built machine learning algorithms, for which we selected a subset of satellite and reanalysis features with global coverage and temporal data availability for at least the past 20 years.

### 2.2.1    Global feature selection

To avoid the model becoming a black box, we did not include features with no intuitive significance or cogent link to EFs (e.g. individual satellite retrieval bands). Table 2 lists the different satellite and reanalysis products included in this study, along with the observed range for each feature over the included fires. We used remote sensing products based on retrievals and reanalysis data with sufficient spatial and temporal coverage, primarily using products based on the Moderate Resolution Imaging Spectroradiometer (MODIS). This meant that at this stage, we did not include data from VIIRS or geostationary satellites. Based on the coordinates of the individual samples we obtained a broad range of features which we then averaged over the samples from each individual fire in order to obtain the fire-averaged feature scores. As proxies for the vegetation conditions and landscape parameters prior to the fire we used Fractional Tree Cover (FTC) and Fractional Bare soil Cover (FBC) from MOD44BV006 (DiMiceli et al., 2015), the Fraction of absorbed Photosynthetically Active Radiation (FPAR) and the Leaf Area Index (LAI), which were retrieved from MCD15A2HC6 (Myneni et al., 2015). Based on MOD09GAC6 surface spectral reflectance (Vermote, 2015), we determined the Normalized Difference Vegetation Index (NDVI) before the fire and the Pgreen (calculated as NDVI before the fire minus the minimum NDVI of the previous year, divided by the total NDVI range of previous year (Korontzi, 2005)).

To estimate the weather conditions during the fire, we used ERA5-land meteorological reanalysis data from the European Centre for Medium Range Weather Forecasts (ECMWF) (Muñoz-Sabater et al., 2021). Hourly meteorological data for air temperature, wind speed, relative humidity, evapotranspiration and potential evapotranspiration were used to obtain the feature score at the UTC-corrected time stamp of each sample. Based on the timing of the sample, the feature value was obtained using linear temporal interpolation. Temperature and relative humidity were subsequently used to derive the Vapor Pressure Deficit (VPD, i.e. the difference between the saturation vapor pressure and the actual vapor pressure) following the method described by Tetens (1930). The Evaporative Stress Index (ESI) was calculated as the actual evapotranspiration divided by the potential evapotranspiration (Anderson et al., 2007). We used ERA5-land monthly average rainfall data to estimate the mean annual rainfall (MAR) over the 1990–2022 period, as well as the cumulative rainfall in the 12 months prior to the fire.

Fire weather comprises combinations of weather and fuel parameters that determine the risk and behaviour of wildfires. Indices like the globally available Fire Weather Index (FWI) have been developed with the aim of estimating the risk of wildfires (De Groot, 1987; Van Wagner, 1987) and are based on global reanalysis data. In this assessment we have included the daily FWI along with some of the intermediate parameters used to calculate the FWI. These intermediate parameters include: (1) the Fine Fuel Moisture Code (FFMC), designed to capture changes in the moisture content of fine fuels and leaf litter, (2) the Drought Code (DC), which captures the moisture content of deep, compacted organic soils and heavy surface fuels, (3) the Build-up Index (BUI) which represents the total fuel availability, and (4) the Initial Spread Index (ISI), which is driven by wind speed and the FFMC, and represents the ability of a fire to spread immediately after ignition. We used the global fire weather indices based on ERA5 (Hersbach et al., 2020) with a 0.25 spatial resolution and 1950-present temporal coverage (Vitolo et al., 2020) that are calculated as part of the European Forest Fire Information System (EFFIS). Global fire weather indices based on ERA5 (Vitolo et al., 2020) showed significant inconsistencies compared to fire weather indices based on GEOS-5 and MERRA-2 obtained from the Global Fire Weather Database (GFWED; Field et al., 2015), meaning these data should not be used as substitutes. Because of the long temporal coverage and higher spatial resolution, we only included ERA5 in our analysis.

For fire severity proxies we used the differential Normalized Burn Ratio (dNBR) and the differential Normalized Difference Vegetation Index (dNDVI) retrieved before and after the fire. These were based on the MODIS surface spectral reflectance, corrected for atmospheric conditions (MOD09GAV6; Vermote, 2015). If the scene before or after the fire was cloud-covered, the preceding or successive scene was used with a limit of 14 days before or after the fire. If no cloud-free scene was available in that time window, the fire was removed from the dataset.

### 2.2.2 Machine learning methodology

We tested a variety of different regression methodologies for the prediction of the fire-WA EFs based on the abovementioned satellite and reanalysis features. Using the Scikit-learn library in Python (Pedregosa et al., 2011), we trained multiple linear regression, decision tree-, random forest-, gradient boosting machine- and neural network regressors to predict the MCE and the EFs of $CO$, $CO_2$, $CH_4$ and $N_2O$. Many of the meteorological and

fuel characteristics follow seasonal patterns and exhibit strong co-variation. While this may be problematic for linear models, it should not negatively impact the decision-tree-based modes and therefore these features were included in the initial modelling stages. We trained the models to reconstruct the measured EF dynamics using the in-situ EF measurements (both ours and those from literature). We removed measurements with missing values for any of the included features. The remaining data was divided into training (70%) and validation data (30%), and the training data was resampled using ten-fold cross validation. This means that the training dataset is divided into ten equal-sized parts or folds. The random forest model is trained and evaluated 10 times. In each iteration, one fold is used as the "temporary validation" set (different from the 30% which is not included in the training data), and the remaining nine folds are used as the training set. The folds are created while allowing sample replacement (i.e., bootstrap method), meaning that for each sample in the dataset, there is an equal chance of it being selected more than once or not selected at all. All regression methods were trained to maximize the explained variance in the data. The hyper parameters (model configurations like number of trees, minimum samples per leaf, maximum features, etc.) were tuned using the scikitlearn "GridsearchCV" algorithm (Pedregosa et al., 2011). Random Forest (RF) regressors gave the best results, closely followed by gradient boosting machine (GBM) regressors. We therefore decided to proceed using RF regressors to predict the MCE and the EFs of CO, $CO_2$, $CH_4$ and $N_2O$.

## 2.3    Spatial extrapolation for global savanna emission estimates

To assess the impact of EF dynamics on emission estimates, and study global spatiotemporal patterns, we developed gridded EF layers that can easily be incorporated into existing emission inventories. The remote-sensing proxies ("features") were resampled to the required spatial resolution by simply averaging the values of the relevant gridcells. For example, to compute the 0.25° fraction tree cover feature, we averaged the fraction tree cover of all 500-meter pixels classified as savanna or grassland. When computing to a higher resolution, e.g. 500-meter EFs, only the higher resolution (MODIS-based) features exhibit pixel-to-pixel variability, while meteorological conditions (derived from ERA5-Land at 0.10° resolution) remain consistent across many adjacent 500-meter pixels. However, due to MODIS-derived features like FTC EF estimates remain distinct between the grid cells. In contrast, temporal resolution within the models is more influenced by ERA5-Land-derived fluctuations. While FTC retrievals remain constant throughout the year, variations in factors like VPD, temperature, and FWI cause EF estimates to fluctuate on a daily basis.

Figure 2 provides an example for the estimation of the CO EF at 500-meter resolution for MODIS tile "h20v10" (covering parts of Zambia, Botswana, Angola, Namibia, Zimbabwe, Mozambique and the Democratic Republic of the Congo) on June 1st, 2019, using the features shown in Fig 2a-e. The temporal resolution of the computed gridded EFs in the example of Fig. 2 is daily, in which the day-to-day EF dynamics are being driven by daily variations in VPD, FPAR, FWI and soil moisture. Burned area products cannot differentiate the time of the day at which a grid cell was burned. For features with a typical diurnal pattern, we therefore weighed the hourly meteorological data by the average diurnal fire profile for the grid cell in the respective month of the year. This diurnal fire profile was based on the three-hourly fractions of daily emissions obtained from GFED4.1s, which is based on the timing of active fire detections from both MODIS and geostationary satellites (Mu et al., 2011; van der Werf et al., 2017). To study the impact of EF dynamics in savannas, we calculated monthly global savanna

emissions by multiplying the dynamic EFs computed by our models with dry matter consumption from GFED4s (Randerson et al., 2012; van der Werf et al., 2017) at 0.25° spatial resolution, for the 2002-2016 period (the period for which MCD64A1C5 as used in GFED4s was available). To classify the landcover type of the cell (Fig. 2f) we used the International Geosphere-Biosphere Program (IGBP) classification (Loveland and Belward, 1997), obtained from the MODIS annual MCD12Q1C6 product (Friedl and Sulla-Menashe, 2019), where the savanna biome comprised land cover types classes 6-11. We then calculated the dynamic monthly MCE and the EFs for CO, $CH_4$, $N_2O$ and $CO_2$ at 0.25° spatial resolution for the savanna biome using the RF models. For burned grid cells that were partially classified as savanna, the EF of the cell was obtained by averaging the EFs of the different biomes in the underlying 500-meter grid cells, weighted by their dry matter consumption. We ran GFED4s using both static (original) and dynamic (this study) EFs for the savanna biome to determine the impact on seasonal and spatial emission patterns using our approach.

## 3   Results

### 3.1   Variability of savanna EF measurements

During six fire seasons we have collected over 4500 bag samples containing emissions from 129 fires, in a variety of savanna ecosystems under different seasonal conditions. Figure 3 shows the range, averages (green diamond), and WA EFs (red crosses) measured during the campaigns listed in Table 1. For the calculation of the WA $N_2O$ EF we excluded samples which contained less than 10 moles of total carbon emissions following the findings described by Vernooij et al. (2021). Table A1 provides a short geomorphological and floristic description of the savanna ecosystems included in Fig. 3, including the seasonal behaviour of the dominant vegetation. The relatively small range in the boxplot describing previous savanna literature (Fig. 3, red box based on studies listed by Andreae (2019)) may be attributed to the fact that most studies report either fire-averages, vegetation type averages or even study averages, whereas the other boxplots based on our measurements show the variability observed between individual samples.

We observed substantial variability within EF bag samples from different savanna ecosystems which was strongly linked to tree-cover density and mean annual rainfall. EFs of CO and $CH_4$ were lower (i.e. higher MCE) in xeric open savannas compared to woodland savannas. Fire-WA EF measurements for CO, $CH_4$ and $N_2O$, using the UAS method were on average 13%, 29% and 44% lower than estimates listed in previous inventories. However, this may be largely attributable to the fact that xeric savannas were overly represented in our measurements in terms of annual biomass consumption (i.e. sample bias). Our measurements in higher rainfall savannas were much closer to the previous averages (Fig. 3). In humid areas like dambos (seasonally inundated grasslands) and riverine forests, we found large intra-seasonal differences in $N_2O$, CO and $CH_4$ EFs. Water availability in these landscape features is often strongly soil type and geomorphology related (Bullock, 1992; Gonçalves et al., 2022), making the correlation with seasonal rainfall less direct and drying patterns over the dry-season more diverse. The grasslands with the highest EFs (found in high-rainfall savanna Dambos) were uncharacteristically green for the time of the season, and under those conditions fires in these landscapes would therefore not be representative of more xeric grasslands.

## 3.2 EF seasonality, fire intensity dynamics and fuel consumption in xeric and mesic savannas

Table 3 lists the EDS and LDS pre- and post-fire fuel characteristics, averaged over all the transects we measured in the respective vegetation type and season. In both xeric- and mesic savannas, the moisture content of the fuel and the relative humidity were substantially lower in the LDS compared to the EDS. This resulted in increases in fire intensity proxies over the dry season. Particularly during measurement campaigns in the Miombo woodlands in Mozambique and Zambia, the fine fuel in the EDS plots predominantly consisted of tree litter and became even more litter-dominated with the progression of the dry season. EDS fires were patchy, and generally did not consume coarse woody debris and shrubs. As the dry season progressed, there was a clear shift towards the combustion of more live foliage and Residual Smouldering Combustion (RSC)-prone fuels like coarse woody debris, stems and densely packed litter, which after months of drought have become more receptive to combustion. RSC occurs after the passage of a flame front and its emissions are not lofted by strong fire-induced convection (Bertschi et al., 2003). The increase in the consumption of live and course fuels towards the end of the dry season coincided with higher EFs for CO and $CH_4$ in the LDS. This shift in combusted fuels also results in a seasonal increase in the WA carbon content of the consumed fuel of woody savannas (Table 3) which linearly scales the EFs of all measured species. For some characteristics (e.g., the total fuel load), it is important to note that the average time since the last fire was not necessarily equal between the listed vegetation types. The higher fuel loads we found in open savannas in Australian compared to Botswana, may be partially attributed to the longer fuel build-up.

Overall, our measurements of CO and $CH_4$ EFs in xeric, grass- and shrub dominated savannas (e.g., Australian spinifex grasslands and open savannas in the Kalahari) were slightly lower in the LDS compared to the EDS campaigns but much lower compared to woody savannas (Fig. 3). Contrary to the mesic savannas, where RSC-prone fuel is readily available and becomes more flammable with the progression of the fire season, fires in xeric shrub and grasslands tended to consume much of the available fuel in the EDS (Table 3). Overall, the WA nitrogen content of the combusted fuel decreased with the progression of the dry season through curing of grasses and litter decomposition. This was somewhat compensated for by an influx of leaf litter and an increased combustion of live shrubs, which were richer in nitrogen than grasses (that had commodiously already cured in the EDS). Overall, fires that consumed more litter emitted more $N_2O$ than grass-dominated fires. Between individual fires, the curing stage of the grasses affected the $N_2O$ EF, with green seasonally-inundated grasslands emitting more $N_2O$ compared to fully cured grasslands. In some miombo woodland fires in Kafue, which were measured in November when the vegetation already carried its first green flush, we also measured relatively high $N_2O$ EFs.

## 3.3 Estimation of BB EFs using random forest regression based on satellite proxies

To extrapolate these relations for use in global emission inventories we correlated the field measurements to satellite products. Table 4 lists the correlations of the individual field-measured ecosystem attributes to the MCE and fire-averaged EF measurements, as well as global satellite proxies. Direct correlations between fire-averaged EF measurements and global satellite proxies as well as intercorrelations between the satellite and reanalysis proxies are listed in Table A2. The strongest predictors for the MCE and the CO and $CH_4$ EF were the tree density in the plots, the grass to litter ratio, the combustion completeness and the WA moisture content of the consumed fuel (Table 4). In turn, these parameters were best correlated to the remotely sensed FTC, FBC, VPD and the FWI.

EFs for CO and $CH_4$ are primarily proportionate to the inverse combustion efficiency (i.e. the not fully oxidized compounds) which had a standard deviation of 90% relative to the mean. $CO_2$, on the other hand is proportionate to the fully combusted carbon fraction which is much larger and more stable with a relative standard deviation of 4.5% compared to its mean. Therefore, the carbon content of the fuel –with a standard deviation of roughly 5%– becomes a dominant factor explaining the variability in $CO_2$ EFs. The features that most strongly correlated with the $N_2O$ EF were the nitrogen to carbon ratio in the combusted fuel and the percentage of grass in the fine fuel (consisting of grass, litter and course woody debris), which in turn correlated with the FBC and the VPD.

For the global estimation of MCE and EFs, we found that RF models performed well with respective out-of-sample correlation coefficients ranging between 0.80 and 0.99. In Figures 4 and 5, feature importance represents the mean accumulation of the impurity decrease within each tree and is an indication of how much the variability in each feature is used as a split criteria by the models to explain the variability in the EF data. The average MCE in the measurements was slightly higher compared to earlier assessments which were used by GFED4s (red line in Figure 4), which may again be attributable to the dominance of relatively dry savannas. Overall, we found that using only globally available features covering a large (>20 year) timespan, we could estimate the field-measured MCE of the fires in the validation set with a mean absolute error (MAE) of 0.006. Using the static MCE in GFED4 (MAE of 0.015 compared to the measurements) as a baseline, this meant a MAE reduction of 60%.

Although the features listed in Fig. 4 all have sufficient spatiotemporal coverage for global emission modelling, some features exhibited strong co-variation. Other retrievals were hampered by LDS cloud-cover (e.g., dNBR and Pgreen), which meant we could not use consistent quality retrievals or had to remove samples from the data. Further simplification using a subset of features that are not directly correlated, reduced the data dependency and computational intensity of the model as well as the loss of training data due to cloud cover, without losing much explained variance. When using a 5-feature subset, we found that RF regressors still predicted much of the variability in the MCE and EFs. Figure 5a shows the predictive performance of a RF regression model that uses VPD, FTC, FWI, FPAR and soil moisture (SM) to estimate the MCE, which was relatively similar to the model predicting MCE using all features (r of 0.80 vs. 0.86).

We found that spatial variability dominated the total variability in the MCE within the savanna biome with higher combustion efficiency in more xeric and open savannas. To isolate the effect of combustion efficiency in the prediction of individual species and make the model more transparent, we added the computed MCE to the predictor features. Both models that were trained using the full set of features in Table 1 and the 5-feature models identified the computed MCE as one of the primary features explaining of the variability in other EFs. The largest deviation from static EFs (vertical red line in Fig. 5d) was predicted for $N_2O$. This is partially due to the large number of new fires which on average (vertical magenta line) were lower than the static reference used in GFED4s. The modelled MCE was the main predictor of the $N_2O$ EF, followed by the soil moisture in the top layer (0 - 7cm depth). Somewhat surprisingly, we found soil moisture to correlate more strongly with the tree density in the plot rather than the fuel moisture content (Table 4).

## 3.4 Impact on global emission estimates using variable savanna emission factors

Figure 6 shows the relative impact of using variable EFs on annual global savanna fire emissions of CO (a), $CH_4$ (b) and $N_2O$ (c), averaged over the 2002-2016 period based on GFED4s. The map only shows cells for which the partial coverage of savannas exceeds 50%. In grid cells that are partially (50-99%) covered by savanna, the total impact on emissions is to some degree diluted as the EFs of the non-savanna biomes remained constant. For CO and $CH_4$, the dominant effect is a spatial redistribution with higher CO and $CH_4$ EFs in mesic, high-tree-cover savannas and lower EFs in xeric savannas compared to previous estimates. For $CO_2$ (not shown), we find the opposite pattern to CO. Relatively speaking, however, changes in $CO_2$ emission are much smaller because most carbon is emitted as $CO_2$, even when MCE values are low. Although CO and $CH_4$ followed the same spatial pattern, we found that MCE affected the $CH_4$ EF more strongly than the CO EF which resulted in lower $CH_4$ to MCE ratios in savannas with lower tree density. Global savanna emissions of CO were 2% higher compared to the GFED4s reference scenario whereas $CO_2$, $N_2O$ and $CH_4$ emissions were respectively 0.2%, 18% and 5% lower. $N_2O$ emissions were lower for the entire savanna biome (Fig. 6c).

Figure 7 shows the seasonal patterns in the average CO EF for different savanna vegetation classes in southern hemisphere Africa. The IGBP savanna subclasses are only used here to indicate the average patterns and are not involved in the EF calculation. Using the IGBP classification, our samples were classified as "Woody savannas" (24%), "Savannas" (42%), "Open shrubland" (21%), "Grassland" (4%), "Cropland/Natural vegetation mosaic" (6%) and "Croplands" (1%). The latter two classes are misclassifications and were all situated in protected areas with no crops. These classes are listed in the accompanied dataset (Vernooij, 2023). We found a stronger and more persistent seasonal decline of the CO EF in xeric grass- and shrublands compared to woody savannas. $N_2O$ EFs showed a similar pattern characterised by a decline over the dry season in the more xeric grass and shrubland savannas while EFs in woody savannas are more stable. The model indicates a reversal of the seasonal trend in woody savannas around August-September, long before these rains start. The coloured areas represent the timing of our field campaigns in this region. Although LDS campaigns were conducted before the first seasonal rains, the graph indicates they may not be indicative of peak-season fires. Figure 8 shows an overview of the relative changes in emissions for the various savanna rich GFED regions. Many of these regions contain both xeric and mesic savannas with contrasting spatial patterns, meaning local differences may be much larger (Fig. 6).

Both our measurements and the savanna biome averages in literature compilations (e.g. Akagi et al., 2011; Andreae, 2019) are subject to sampling bias when representing global savannas. A disproportionate number of field studies are clustered around reactively accessible locations with a well-developed research infrastructure, whereas other fire-prone areas lack direct field measurements. Rather than comparing the average of our savanna measurements to the literature averages, we computed the dynamic EFs globally using the RF model and subsequently calculated the emissions for the entire savanna biome. We then divided these annual emissions by the consumed biomass from GFED4s to get the annual consumed-biomass-weighted-average EFs, which we will further refer to as the "effective" EFs. Over the 2002-2016 period, the effective EFs over the savanna biome were 1685 ($\pm$ 5) for $CO_2$, 64.3 ($\pm$ 0.6) for CO, 1.9 ($\pm$ 0.0) for $CH_4$ and 0.16 ($\pm$ 0.00) for $N_2O$, with the number in the parentheses indicating the interannual standard deviation. In Table 4, we compare the effective average EFs over

the 2002-2016 period calculated by our model to the static average EFs for savanna and grassland vegetation used by GFED4s and those suggested by Andreae (2019) and Wiedinmyer et al. (2023). Table 4 also lists the average EFs of the UAS measured fires and the average EFs of all included fires (including literature studies). Except for $N_2O$, the differences between the effective EFs compared to more recently updated static EFs from Andreae (2019) were larger (+1.3% for $CO_2$, -7.1% CO, -31.4% $CH_4$ and -3.7%) than the differences compared to static EFs from GFED4s.

## 4 Discussion

### 4.1 Comparison with previous studies

The largest difference compared to previous savanna burning emission estimates is the reduction in $N_2O$ emissions. Rather than being the effect of including spatiotemporal dynamics, this reduction resulted from a substantial influx of new $N_2O$ EF measurements that exhibited significantly lower values than the averages found in EF compilations. Our field measurements yielded an average EF of 0.11 g $kg^{-1}$, while EF compilations reported averages of 0.21 g $kg^{-1}$ (Andreae and Merlet, 2001), 0.20 g $kg^{-1}$ (Akagi et al., 2011), and 0.17 g $kg^{-1}$ (Andreae, 2019). However, in our measurements, xeric savannas are overrepresented. When using the global RF model to extrapolate the measurements over the entire savanna biome the "effective" average $N_2O$ EF –for savanna grid cells at the time of their burning– was 0.16 g $kg^{-1}$, which is similar to the value listed in Andreae (2019). It is known that older studies might overestimate $N_2O$, due to $N_2O$ formation in stainless steel sample containers (Muzio and Kramlich, 1988). Particularly compared to more recent studies, our EFs were in line with other savanna measurements from South America (0.05-0.07 g $kg^{-1}$; Hao et al., 1991; Susott et al., 1996), Australia (0.07 – 0.12 g $kg^{-1}$; Hurst et al., 1994; Meyer et al., 2012; Surawski et al., 2015) and Africa (0.16 g $kg^{-1}$; Cofer et al., 1996). In accordance with Winter et al. (1999b), we found $N_2O$ EFs to be closely correlated with the nitrogen content of the fuel. Through this relation, we can explain both the spatial distribution observed in Fig. 6c and the different seasonal trends. In line with Susott et al. (1996) and Ward et al. (1992) we found that woody vegetation has higher nitrogen content contained in the foliage (Table 3), causing higher $N_2O$ emissions from tree dominated areas. We found relatively low nitrogen content for Australian open woodland savannas, which was in line with previous studies (Bustamante et al., 2006). The seasonal reduction in the nitrogen content of the fuel as the vegetation cures (Table 3) coincides with a reduction of the $N_2O$ EF over the dry season (Yokelson et al., 2011; Vernooij et al., 2021). This tends to happen quicker in xeric grass- and shrublands compared to more mesic and tree-covered areas. On the other hand, as fuels get more receptive over the dry season, fires consume increasingly more litter, coarse fuels and live foliage, provided these fuels are available (Table 3). This increases the WA carbon- and nitrogen contents of the fuel.

For carbonaceous species our model predicts a spatial redistribution, characterized by higher combustion efficiency in lower tree-cover savannas and lower combustion efficiencies in more woody savannas. Previous research by van Leeuwen and van der Werf (2011) identified multi-linear correlations between EFs of $CO_2$, CO and $CH_4$ and environmental drivers resulting in coefficients of determination ($r^2$) ranging from 0.48 to 0.62. In accordance with their study as well as many other field studies (e.g. Laris et al. (2021) and Sinha et al. (2004)), we found the FTC to be a strong predictor of the MCE and the EFs of CO and $CH_4$ (Fig. 9). When denoted in grams per kilogram of dry biomass consumed, EFs of carbonaceous species are dependent on both the combustion

efficiency and the carbon content of the fuel. The carbon content is often fixed in global studies of EFs (e.g. 45% in Andreae (2019) and Andreae and Merlet (2001) or 50% in Akagi et al. (2011)), with the latter forming the basis of the EFs used in GFED4s that represent the static EF references in this study. However, both the combustion efficiency and the carbon content have a spatial component with higher carbon contents in shrubs and trees compared to grasses (Table 3). For the studied fires, the WA carbon content of the fuel ranged from 40.3 to 49.3%, which linearly scales to a 22% difference in EFs between those extremes. In line with Andreae (2019), we assigned a carbon content of 45% to literature studies for which the carbon content was not reported which was close to our average measured value of 45.8 ± 2.3%. Contrary to previous research which indicated that dryer conditions in the LDS would lead to higher-MCE fires in both grasslands and savanna woodlands (Korontzi, 2005), we found lower MCE in these regions under late-LDS conditions (Fig. 3). One potential explanation is that although the LDS fires were more intense, they consumed much more RSC-prone fuels (Table 3), which may explain the higher $CH_4$ and CO EFs. An alternative explanation to this fuel-driven MCE reduction is that in certain areas our measurement campaigns missed the peak-season when fires are driven by stronger winds (Laris et al., 2021; N'Dri et al., 2018), and that fire intensity and MCE in these areas would already be on the decline. Eck et al. (2013) studied seasonal changes of BB particles during 15 annual fire seasons in xeric (e.g. Etosha pan and Kruger national park) and mesic (e.g. Mongu) savannas in southern Africa using the Aerosol Robotic Network (AERONET). They found a linear trend of the single scattering albedo (SSA), increasing throughout the dry season, which would support a late dry season decrease in MCE (Liu et al., 2014; Pokhrel et al., 2016). We found that, in the xeric savannas, the composition of the fuel in LDS fires did not significantly differ from EDS fires, as most of the available fuel was consumed in both the EDS and LDS fires. In these areas, we did observe a slight seasonal decline in CO and $CH_4$ EFs.

In accordance with previous studies (e.g. Korontzi et al., 2003b; van Leeuwen and van der Werf, 2011; Barker et al., 2020), we found steeper $CH_4$ EF to MCE regression slopes in woodlands compared to grasslands. Our data indicated a positive correlation of the $CH_4$ EF to MCE slope with the FTC based on MOD44Bv006. The MCE is a simplified form of the combustion efficiency and only calculated using CO and $CO_2$ emissions. Being less oxidized than CO (which is still common in flaming combustion), $CH_4$ emissions have a stronger dependency on the actual combustion efficiency ($CO_2$ divided by all carbon emissions). While most studies describe the relationship between the $CH_4$ EF and the MCE as being linear (Korontzi et al., 2003; van Leeuwen and van der Werf, 2011; Selimovic et al., 2018; Yokelson et al., 2003), we found that for individual bag samples it was better described using a nonlinear function (Fig. 9), in line with findings by Meyer et al. (2012) for Australian savanna measurements. Figure 9 represents individual bag measurements rather than fire averages (for which the spread in MCE is much lower). Laboratory experiments described by Selimovic et al. (2018) and others showed that the $CH_4$ to CO ratio is more complex and variable in real-time than at the fire-average level. Individual bag samples sampled over a concise 35-second timeframe thus exhibit a broader range and more pronounced variation in comparison to fire averages. Stable carbon isotopes also point to $CH_4$ emissions being more depleted in heavy carbon ([13]C) compared to CO in both mixed (C3 and C4) and single-fuel-type experiments using wooden logs, indicating a stronger dominance of RSC and the pyrolysis of lignin in its total emissions (Vernooij et al. 2022b). Mainly within woody savannas, this clarifies why studies focused on either smouldering or flaming phase emissions exhibit diverse slopes for $CH_4$ EF to MCE when employing linear regressions. Additionally, this

phenomenon accounts for the inclination of the slope to intensify in fueltypes characterized by higher lignin content.

Although higher MAR generally coincides with high FTC, this was not the case for our measurements from Brazil. The measured areas in the Estação Ecológica Serra Geral do Tocantins (EESGT) received relatively high MAR (1250–1600 mm yr$^{-1}$) compared to 850–1250 mm yr$^{-1}$ for Zambian and 890–1100 mm yr$^{-1}$ for Mozambican Miombo woodlands. Nonetheless, although being strictly protected from logging and other land clearing practices, the MOD44BV006 FTC in the measured areas in EESGT was very low (1-10%, with an average of 2%) compared to 7-32% with an average of 19% for Zambian-, and 3-43% with an average of 22% for Mozambican miombo woodlands. That our measurements in the EESGT were skewed towards open savannas (that typically burn with higher MCE), may explain the relatively low CH$_4$ EF to MCE slope discussed in Vernooij et al. (2021). For the whole Cerrado, the average MOD44BV006 FTC is 17%, indicating that the measurements in EESGT may be underestimating the MCE in other parts of the Cerrado. According to its classification, MCD44BV006 FTC only includes canopies of trees exceeding 5m in height (Adzhar et al., 2021) which may be why some common Cerrado species are classified as shrubs. However, the EFs observed from these areas were similar to those observed in low tree-cover savannas.

Measurements of fuel loads were higher than previous measurements from African savannas described by Shea et al. (1996). They found average fine fuel loads (litter and grass) of 3.8 tonne ha$^{-1}$ in moist Miombo woodland. In semiarid Miombo woodland they found 3.1 tonne ha$^{-1}$, In comparison we found 5.6 tonne ha$^{-1}$ in Mozambican Miombo woodland and 5.6 tonne ha$^{-1}$ in Zambian Miombo woodland. The percentage of grasses in these fuels was similar; Shea et al. (1996) reported 24% in moist Miombo woodland and 18% in semi-arid Miombo woodland whereas we found 37% in Mozambican and 18% in Zambian Miombo woodlands. The combustion completeness of these fuels was slightly lower in our fires at 50-80% versus 80-92% reported by (Shea et al., 1996), albeit that the lower values in this range occurred in the EDS. Combustion completeness of shrub leaves and course woody debris were in the same range. For dambo grasslands our fuel loads were also much higher at 6.2 ($\pm$ 2.16) tonne ha$^{-1}$ of which 99% was grass versus 3.1 tonne ha$^{-1}$ from Shea et al. (1996). Although these differences are large, they may be attributed to the significant natural variability in productivity and decay related to water availability, fire frequency, and termite and grazing activities in these natural landscapes.

## 4.2   Model representativeness

This is the first study to quantify the spatial distribution of GHG EFs over the entire savanna biome using field measurements from a variety of savanna ecosystems and their relation to global data mainly from satellites. Although spatiotemporal coverage has improved, there are still many understudied savanna and grassland areas for which we have derived EFs based on our model. Figure 1 clearly illustrates the gaps in the spatial distribution of the training data. Particularly savannas bordering the tropical rainforest, northern hemisphere Africa, meso America, south-east Asia as well as temperate grassland ecosystems are understudied. Due to the lack of measurements in these ecosystems, EFs are presently computed based on measurements primarily taken in southern hemisphere Africa. Nevertheless, EF trends in other regions might considerably differ from those

observed in extensively studied savannas. To guarantee the model's relevance to specific regions, it remains essential to calibrate and evaluate the model using supplementary in-situ emission factor measurements.

Most of the fires used to train the models were prescribed fires set by scientists or park rangers in protected areas in order to facilitate collection of data pre and post burn on site. It is common practice to extrapolate these measurements in relatively undisturbed savanna vegetation to the wider savanna. Even though these protected natural areas tend to burn more frequently, they represent a minority of the area that is currently modelled using savanna and grassland emission factors by global inventories (e.g. Fig.1). Most of this area is to some degree affected by humans though cattle ranging, wood harvesting, slash and burn agriculture, etc. This means fires in this study may not always represent the burning practices by local farmers, and representativeness of our work for the larger savanna area remains therefore uncertain. The samples were predominantly collected over heading fires, which in the measured fires typically represented most of the burned area. A common approach for prescribed fires is burning against the wind (backing fire), to minimise both the impact on vegetation and risk of spread. In a heading fire, RSC can be increased because the high rate of spread and patchiness leaves fuels smouldering further from the convection associated with the advancing flame front. In accordance with Wooster et al. (2011) and Laris et al. (2021), we found higher MCE in samples from backing fires, indicating less RSC and thus $CH_4$ and CO emissions in these types of fires. Another possible explanation for the higher MCE in the backing fire samples is that slower lofting RSC smoke does not mix with the flaming combustion emissions in these measurements, like it does in heading fires.

## 4.3 Spatial resolution and model considerations

For this research, we computed the average attributes within the 0.25° grid cell before calculating the savanna EF. This spatial resolution was selected because GFED4s burned area data, including assumptions for small fires, being generated at a 0.25° resolution. Nonetheless, there are potential advantages to future EF estimations at greater spatial resolutions. Recent studies indicate that higher resolution modelling yield different emissions than those based on aggregated data due, in part, to improved representation of landscape heterogeneity (van Wees and van der Werf, 2019). Enhancing the resolution of meteorological data would further amplify the precision of these models. These advancements anticipate that future global emission inventories will adopt higher spatial resolutions, enabling better representation of local or regional dynamics. We found the highest variability of EFs within smaller landscape features that are bound to geomorphological niches, typically along rivers and valleys. While these features are likely to have low significance for global emission patterns, they represent vital ecosystems that may require special fire protection. In its current form, the model may not always pick up on those landscape features. High-resolution modelling allows for a better understanding of localized fire regimes, especially in areas with relatively heterogeneous landcover.

The model is limited by the accuracy and spatial resolution of the underlying products. Using the features included in the current models, EFs can be calculated up to the native spatial resolution of the included MODIS-based products (500×500m), which is also the resolution of globally available burned area products. New high-resolution burned area products, however, indicate that these global products, including the GFED4s data used for global emission analyses in this study, grossly underestimate burned area due to omission of small fires (Chen

et al., 2023; Roteta et al., 2021; Roy et al., 2019). This also pertains to a substantial proportion of the fires we measured. Of the UAS-measured fires in this study only 5 of the 45 EDS fires (11%) and 13 of the 65 LDS fires (20%) were registered by MCD64A1 as burned area (including adjacent pixels and a 4-day time lag). Out of the 45 EDS fires, only 4 (9%), and among the 65 LDS fires, just 32 (49%), were detected by VIIRS S-NPP as thermal anomalies, with the hotspot's center point (accounting for a 1-day time lag) falling within a 3.5 km radius of the sample. Depending on the spatiotemporal nature of these omissions, this may affect some of the results in this study concerning the effects of the EF dynamics on total emissions. Chen et al. (2023) indicate that in the savannas, disproportionately more burned area is added in higher tree-cover areas when using higher resolution satellite imagery. Giving more weight to these areas would mean our savanna-wide effective EFs of CO, $CH_4$ and $N_2O$ would increase. The Sentinel-2 based burned area product from Roteta et al. (2021) performed much better and registered 8 of our 14 EDS fires (57%) and all of our 16 LDS fires (100%) in Botswana and Mozambique in 2019 (including adjacent pixels and up to a 21-day time lag). Due to the fewer overpasses the temporal allocation of this product is less precise with an average time lag of 5.5 days. Figure 10 shows the portion of our EDS and LDS fires that were detected by various satellite algorithms.

Fire intensity proxies (dNDVI and dNBR from MODIS) were considered by the models to be poor predictors for the EFs. A potential explanation is that these features were not always representative, as many of the fires only affected part of the pixel. Similar misrepresentation errors can be expected for the NDVI before the fire, FPAR and the Pgreen. Particularly in the LDS, we were often limited to areas that were enclosed by recent fire scars (0-2 years old) or other non-flammable boundaries like roads or bare areas. Although the burnt areas were sizable (several hectares), many of the retrievals in these pixels may poorly represent the burned vegetation. Along with inconsistent retrievals related to cloud cover, this may contribute to these features being deemed poor predictors by the models. Enhanced resolution features could improve the accuracy of pixel representations for the actual burned vegetation.

The meteorological parameters obtained from the ERA5-Land dataset carry uncertainty. This uncertainty increases when examining earlier time periods or remote regions due to diminished validation data availability. To what extent uncertainty propagates to the EF predictions varies depends mostly on whether there is a bias that was also present in the training data or misinterpretation or uncertainty in general. As this model is trained using specific datasets, these datasets should not be replaced by other sources without evaluating the consistency of that source with the feature training data. FTC and FBC, based on MOD44Bv006 were found to be strong predictors of BB EFs. However, intercomparison with Tropical Biomes in Transition (TROBIT) field sites in African, Brazilian and Australian savannas has shown that this product consistently underestimates canopy cover in tropical savannas by between 9 to 15% (Adzhar et al., 2021). Products based on higher-resolution satellite retrievals (e.g. LandSat and Sentinel) have the potential to further enhance the spatial resolution of the EF estimates to include small landscape features and thus become more representative. Although all satellite data comes with some uncertainty, we feel the errors are small enough to have high confidence in the key findings such as lower EFs in dry regions and higher in wetter regions.

The interdependence among features led to varying feature importance scores (depicted in Fig. 4) across different model runs, driven by the test-train data division and bootstrap resampling. For instance, a decision tree split based on VPD might closely resemble soil moisture or RH, and FTC in national parks often exhibits strong correlation with the MAR, with the exception of our measurement sites in Brazil. While we conducted model runs considering different subsets of features and selected the optimal one, it is important to note that various features might also effectively account for a significant portion of the variance. In cases where features had substantial co-variation (such as FPAR and LAI, or FWI and ISI), this resulted in the selection of only one feature for the simplified model, even if both features demonstrated high initial scores.

The models are currently trained using meteorological features obtained from ERA5-Land (Muñoz-Sabater et al., 2021) which is available from 1950 to present and has a 2- to 3-month delay. When interested in longer time periods or for near-real-time (NRT) applications these features may be substituted with ERA5 (Hersbach et al., 2020) which is available from 1940 to present with a shorter latency period of 5 days, or even CMIP climate projections. Although supplementing the datasets on which the models are trained with alternative data always comes with additional uncertainty, we found meteorological parameters obtained from ERA5-Land to be in close accordance with ERA5, indicating the two may also be substituted. This means that the EFs computed using the methodology outlined in this paper could potentially also be used to improve NRT biomass burning emission estimates like those from CAMS-GFAS (Andela et al., 2015; Di Giuseppe et al., 2016).

## 5   Conclusions

Over the last decade, substantial progress has been made increasing the spatiotemporal coverage of savanna fire emission factor measurements (EFs). In this study we described the variability of GHG EFs measured during 18 new field campaigns over the 2017-2022 period during which we sampled 129 fires in different parts of the savanna biome using a UAS platform. On average CO, $CH_4$ and $N_2O$ EFs in these UAS measurements were respectively 13%, 29% and 44% lower compared to the biome-averaged EFs used in previous inventories. However, from a global savanna perspective, xeric savannas with relatively low EFs were over-represented in our measurements which could explain part of the mismatch. The measured fires were predominantly intentional burns conducted by scientists or park rangers in protected areas for data collection, and while these measurements are extended to undisturbed savanna, the majority of the broader savanna used in emission models is influenced by human activities such as cattle grazing and agriculture, raising some uncertainty about the representativeness of our findings for global savannas. Measurements of the pre and postfire fuel load and the fuel conditions during the fire indicated significant changes in fuel receptiveness resulting in increased fire intensity over the dry season. Particularly for mesic savannas, an increase in the combustion of RSC-prone fuels resulted in higher EFs of CO and $CH_4$ during LDS fires. The main drivers of variability in CO and $CH_4$ EFs were tree-cover, fuel moisture content and the prevalence of grasses while EFs for $N_2O$ strongly correlated with the nitrogen content of the fuel which, in turn, is strongly linked to the grass to litter ratio. Although these correlations are consistent with previous savanna EF studies, quantifying their impact on EFs for the use in global emission studies has so far been hampered by a lack of measurements.

We developed a random forest regressor that estimates dynamic EFs (monthly EFs at 0.25°) based on satellite products to replace the use of static biome averaged EFs in global emission inventories, or the use of a dichotomy of EDS vs LDS EFs (based on a cut-off date). The model-produced data resulted in significant fire-specific improvements compared to static biome-averaged EFs, reducing the mean absolute error in the modelled versus measured predictions by 64% for $CH_4$, 58% for $N_2O$, 85% for CO and 79% for $CO_2$. Except for $N_2O$ EFs, our study does not indicate that savanna averages have large errors, but rather that temporal and especially spatial variability is large and is better accounted for by using a more sophisticated model. We used the dynamic EF models to calculate the emissions for global savanna emissions over the 2002-2016 period, which is more indicative of the "effective" EF differences. This resulted in a spatial redistribution of emissions over the savanna biome, characterized by increases of average annual emissions of CO, and $CH_4$ in woody savannas and reductions in open savannas. While the model indicates an initial seasonal decrease in combustion efficiency as the vegetation dried out, there was a reversal for woody savannas towards the end of the dry season, occurring before the first seasonal rains. This shift coincides with the increased consumption of live vegetation and RSC-prone fuels like densely packed litter, coarse woody debris). Xeric savannas had much lower EFs with a longer and more profound seasonal decrease in CO and $CH_4$. Although $N_2O$ EFs were lower for the entire savanna biome, they followed a similar spatiotemporal pattern.

The proposed dynamic EF method resulted in a 18% reduction in the estimated annual global $N_2O$ emissions from savanna fires, compared to static averages, with emission reductions of up to 60% in xeric regions. The impact on the global savanna emission estimates for $CO_2$ (decrease of 0.2%), CO (increase of 1.8%) and $CH_4$ (decrease of 2.1%) was low, indicating the use of static EFs did not lead to biases for studies focusing on global emissions. However, the regional impact on these EF estimates was as high as 60% and even 80% under extreme seasonal conditions, highlighting its variability at a more local level. Overall, the model results are a first step towards more dynamic and area specific emission inventories, which we plan to make available in monthly and daily resolution at 0.25° and will further improve as more measurements and better remote sensing products become available.

### Data availability:

The data table containing the training data used for this article along with an explanatory table are available online at: 10.5281/zenodo.7689032 (Vernooij, 2023). Model results are available upon request.

### Author contribution:

RV and GRvdW designed the study; RV, TE, JRS, CY, RB, JE, AE, NR, MW, TS, MG, MB, MC and CB conducted the field measurements; RV conducted the analyses on the samples; RV performed the random forest modelling and global analyses and wrote the manuscript with help from DvW and GRvdW.

**Competing interests:** The authors declare no competing interests.

**Acknowledgements**

This research was supported by the Netherlands organization for Scientific Research (NWO) (Vici scheme research programme, no. 016.160.324) and the Ammodo Science Award (2017) for Natural Sciences. The measurement campaigns in Botswana received funding from the International Savanna Fire Management
5  Initiative (ISFMI) and Australia's department of foreign affairs and trade while the field campaigns in Zambia in 2021 and 2022 were partially funded by the United Nations green climate fund. We owe great thanks for the contributions of countless individuals and institutions that provided the permissions, oversight, logistics and expertise needed to perform the field measurements in a safe and coordinated fashion. Among others this has been made possible thanks to 321 Fire, the Brazilian Instituto Chico Mendes de Conservação da Biodiversidade, South
10  African National Parks, the Botswana department of forestry and range resources, the Tsodilo community development trust, the Zambian department of forestry, the wildlife conservation society, the Administração Nacional das Áreas de Conservação in Mozambique, the Australian central land council and the Yanunijarra aboriginal corporation.

# Literature

Adzhar, R., Kelley, D., Dong, N., Torello Raventos, M., Veenendaal, E., Feldpausch, T., Philips, O., Lewis, S., Sonké, B., Taedoumg, H., Schwantes Marimon, B., Domingues, T., Arroyo, L., Djagbletey, G., Saiz, G. and Gerard, F.: Assessing MODIS Vegetation Continuous Fields tree cover product (collection 6): performance and applicability in tropical forests and savannas, Biogeosciences Discuss., (February), 1–20, 2021.

Akagi, S. K., Yokelson, R. J., Wiedinmyer, C., Alvarado, M. J., Reid, J. S., Karl, T., Crounse, J. D. and Wennberg, P. O.: Emission factors for open and domestic biomass burning for use in atmospheric models, Atmos. Chem. Phys., 11(9), 4039–4072, doi:10.5194/acp-11-4039-2011, 2011.

Andela, N., Kaiser, J. W., Van Der Werf, G. R. and Wooster, M. J.: New fire diurnal cycle characterizations to improve fire radiative energy assessments made from MODIS observations, Atmos. Chem. Phys., 15(15), 8831–8846, doi:10.5194/acp-15-8831-2015, 2015.

Anderson, M. C., Norman, J. M., Mecikalski, J. R., Otkin, J. A. and Kustas, W. P.: A climatological study of evapotranspiration and moisture stress across the continental United States based on thermal remote sensing: 1. Model formulation, J. Geophys. Res. Atmos., 112(10), 1–17, doi:10.1029/2006JD007506, 2007.

Andreae, M. O.: Emission of trace gases and aerosols from biomass burning – an updated assessment, Atmos. Chem. Phys., 8523–8546, 2019.

Andreae, M. O. and Merlet, P.: Emission of trace gases and aerosols from biomass burning, Biogeochemistry, 15(4), 955–966, doi:10.1029/2000GB001382, 2001.

Bertschi, I., Yokelson, R. J., Ward, D. E., Babbitt, R. E., Susott, R. A., Goode, J. G. and Hao, W. M.: Trace gas and particle emissions from fires in large diameter and belowground biomass fuels, J. Geophys. Res. Atmos., 108(13), doi:10.1029/2002jd002100, 2003.

Bullock, A.: Dambo hydrology in southern Africa-review and reassessment, J. Hydrol., 134(1–4), 373–396, doi:10.1016/0022-1694(92)90043-U, 1992.

Bustamante, M. M. C., Medina, E., Asner, G. P., Nardoto, G. B. and Garcia-Montiel, D. C.: Nitrogen cycling in tropical and temperate savannas, Biogeochemistry, 79(1–2), 209–237, doi:10.1007/s10533-006-9006-x, 2006.

Chen, L. W. A., Verburg, P., Shackelford, A., Zhu, D., Susfalk, R., Chow, J. C. and Watson, J. G.: Moisture effects on carbon and nitrogen emission from burning of wildland biomass, Atmos. Chem. Phys., 10(14), 6617–6625, doi:10.5194/acp-10-6617-2010, 2010.

Chen, Y., Hall, J., Wees, D. Van, Andela, N., Hantson, S., Giglio, L., Van, G. R., Werf, D., Morton, D. C. and Randerson, J. T.: Multi-decadal trends and variability in burned area from the 5th version of the Global Fire Emissions Database ( GFED5 ), , (May), 1–52, 2023.

Cofer, W. R., Levine, J. S., Winstead, E. L., Cahoon, D. R., Sebacher, D. I., Pinto, P. and Stocks, B. J.: Source compositions of trace gases released during African savanna fires, J. Geophys. Res., 101, 23,597-23,602, 1996.

DiMiceli, C., Carroll, M., Sohlberg, R., Kim, D., Kelly, M. and Townshend, J.: MOD44B MODIS/Terra Vegetation Continuous Fields Yearly L3 Global 250m SIN Grid V006 [Data set], NASA EOSDIS L. Process. DAAC, doi:10.5067/MODIS/MOD44B.006, 2015.

Eames, T., Russell-Smith, J., Yates, C., Edwards, A., Vernooij, R., Ribeiro, N., Steinbruch, F. and van der Werf, G. R.: Instantaneous pre-fire biomass and fuel load measurements from multi-spectral UAS mapping in southern African Savannas, Fire, 4(1), 1–19, doi:10.3390/fire4010002, 2021.

Eck, T. F., Holben, B. N., Reid, J. S., Mukelabai, M. M., Piketh, S. J., Torres, O., Jethva, H. T., Hyer, E. J., Ward, D. E., Dubovik, O., Sinyuk, A., Schafer, J. S., Giles, D. M., Sorokin, M., Smirnov, A. and Slutsker, I.: A seasonal trend of single scattering albedo in southern African biomass-burning particles : Implications for satellite products and estimates of emissions for the world ' s largest biomass-burning source, J. Geophys. Res. Atmos., 118, 6414–6432, doi:10.1002/jgrd.50500, 2013.

Field, R. D., Spessa, A. C., Aziz, N. A., Camia, A., Cantin, A., Carr, R., De Groot, W. J., Dowdy, A. J., Flannigan, M. D., Manomaiphiboon, K., Pappenberger, F., Tanpipat, V. and Wang, X.: Development of a Global Fire Weather Database, Nat. Hazards Earth Syst. Sci., 15(6), 1407–1423, doi:10.5194/nhess-15-1407-2015, 2015.

Friedl, M. and Sulla-Menashe, D.: MCD12Q1 MODIS/Terra+Aqua Land Cover Type Yearly L3 Global 500m SIN Grid V006 [Data set]., NASA EOSDIS L. Process. DAAC, doi:10.5067/MODIS/MCD12Q1.006, 2019.

Di Giuseppe, F., Remy, S., Wetterhall, F. and Pappenberger, F.: Improving CAMS biomass burning estimations by means of the Global ECMWF Fire Forecast system (GEFF)., 2016.

Gonçalves, R. V. S., Cardoso, J. C. F., Oliveira, P. E., Raymundo, D. and de Oliveira, D. C.: The role of topography, climate, soil and the surrounding matrix in the distribution of Veredas wetlands in central Brazil, Wetl. Ecol. Manag., 30(6), 1261–1279, doi:10.1007/s11273-022-09895-z, 2022.

De Groot, W. J.: Interpreting the canadian forest fire weather index (FWI) system, in Fourth Central Regional Fire Weather Committee Scientific and Technical Seminar, pp. 3–14, Canadian Forestry Service, Northern Forestry Centre, Edmonton, Alberta., 1987.

Hao, W. M., Scharffe, D., Lob, J. M. and Crutzen, P. J.: Emissions of N20 from the Burning of Biomass in an Experimental System, Geophys. Res. Lett., 18(6), 999–1002, 1991.

Hersbach, H., Bell, B., Berrisford, P., Hirahara, S., Horányi, A., Muñoz-Sabater, J., Nicolas, J., Peubey, C., Radu,
R., Schepers, D., Simmons, A., Soci, C., Abdalla, S., Abellan, X., Balsamo, G., Bechtold, P., Biavati, G., Bidlot, J., Bonavita, M., Chiara, G., Dahlgren, P., Dee, D., Diamantakis, M., Dragani, R., Flemming, J., Forbes, R., Fuentes, M., Geer, A., Haimberger, L., Healy, S., Hogan, R. J., Hólm, E., Janisková, M., Keeley, S., Laloyaux, P., Lopez, P., Lupu, C., Radnoti, G., Rosnay, P., Rozum, I., Vamborg, F., Villaume, S. and Thépaut, J.: The ERA5 global reanalysis, Q. J. R. Meteorol. Soc., 146(730), 1999–2049, doi:10.1002/qj.3803, 2020.

Hurst, D. F., Griffith, D. W. T., Carras, J. N., Williams, D. j. and Fraser, P. J.: Measurements of trace gases emitted by Australian savanna fires during the 1990 dry season, J. Atmos. Chem., 18(1), 33–56, doi:10.1007/BF00694373, 1994.

Kaiser, J. W., Heil, A., Andreae, M. O., Benedetti, A., Chubarova, N., Jones, L., Morcrette, J. J., Razinger, M., Schultz, M. G., Suttie, M. and Van Der Werf, G. R.: Biomass burning emissions estimated with a global fire
assimilation system based on observed fire radiative power, Biogeosciences, doi:10.5194/bg-9-527-2012, 2012.

Korontzi, S.: Seasonal patterns in biomass burning emissions from southern African vegetation fires for the year 2000, Glob. Chang. Biol., 11(10), 1680–1700, doi:10.1111/j.1365-2486.2005.001024.x, 2005.

Korontzi, S., Ward, D. E., Susott, R. A., Yokelson, R. J., Justice, C. O., Hobbs, P. V., Smithwick, E. A. H. and Hao, W. M.: Seasonal variation and ecosystem dependence of emission factors for selected trace gases and PM
2.5 for southern African savanna fires, J. Geophys. Res. Atmos., 108(D24), doi:10.1029/2003JD003730, 2003.

Laris, P.: On the problems and promises of savanna fire regime change, Nat. Commun., 12(1), 1–5, doi:10.1038/s41467-021-25141-1, 2021.

Laris, P., Koné, M., Dembélé, F., Yang, L. and Jacobs, R.: Methane gas emissions from savanna fires : What analysis of local burning regimes in a working West African landscape tell us, , (March), 1–20, 2021.

van Leeuwen, T. T. and van der Werf, G. R.: Spatial and temporal variability in the ratio of trace gases emitted from biomass burning, Atmos. Chem. Phys., 11(8), 3611–3629, doi:10.5194/acp-11-3611-2011, 2011.

Liu, S., Aiken, A. C., Arata, C., Dubey, M. K., Stockwell, C. E., Yokelson, R. J., Stone, E. A., Jayarathne, T., Robinson, A. L., DeMott, P. J. and Kreidenweis, S. M.: Aerosol single scattering albedo dependence on biomass combustion efficiency: Laboratory and field studies, Geophys. Res. Lett., 41(2), 742–748,
doi:10.1002/2013GL058392, 2014.

Loveland, T. R. and Belward, A. S.: The International Geosphere Biosphere Programme Data and Information System global land cover data set (DISCover), Acta Astronaut., 41(4–10), 681–689, doi:10.1016/S0094-5765(98)00050-2, 1997.

Meyer, C. P., Cook, G. D., Reisen, F., Smith, T. E. L., Tattaris, M., Russell-Smith, J., Maier, S. W., Yates, C. P.
and Wooster, M. J.: Direct measurements of the seasonality of emission factors from savanna fires in northern Australia, J. Geophys. Res. Atmos., 117(D20), doi:10.1029/2012JD017671, 2012.

Mu, M., Randerson, J. T., Van Der Werf, G. R., Giglio, L., Kasibhatla, P., Morton, D., Collatz, G. J., Defries, R. S., Hyer, E. J., Prins, E. M., Griffith, D. W. T., Wunch, D., Toon, G. C., Sherlock, V. and Wennberg, P. O.: Daily and 3-hourly variability in global fire emissions and consequences for atmospheric model predictions of carbon
monoxide, J. Geophys. Res. Atmos., 116(24), 1–19, doi:10.1029/2011JD016245, 2011.

Muñoz-Sabater, J., Dutra, E., Agustí-Panareda, A., Albergel, C., Arduini, G., Balsamo, G., Boussetta, S., Choulga, M., Harrigan, S., Hersbach, H., Martens, B., Miralles, D. G., Piles, M., Rodríguez-Fernández, N. J., Zsoter, E., Buontempo, C. and Thépaut, J. N.: ERA5-Land: A state-of-the-art global reanalysis dataset for land applications, Earth Syst. Sci. Data, 13(9), 4349–4383, doi:10.5194/essd-13-4349-2021, 2021.

Muzio, L. J. and Kramlich, J. C.: An artifact in the measurement of N2O from combustion sources, Geophys. Res. Lett., 15, 1369–1372, doi:https://doi.org/10.1029/GL015i012p01369, 1988.

Myneni, R., Knyazikhin, Y. and Park, T.: MCD15A2H MODIS/Terra+Aqua Leaf Area Index/FPAR 8-day L4 Global 500m SIN Grid V006 [Data set], , doi:10.5067/MODIS/MCD15A2H.006, 2015.

N'Dri, A. B., Soro, T. D., Gignoux, J., Dosso, K., Koné, M., N'Dri, J. K., Koné, N. A. and Barot, S.: Season affects fire behavior in annually burned humid savanna of West Africa, Fire Ecol., 14(2), 5, doi:10.1186/s42408-018-0005-9, 2018.

Pedregosa, F., Varoquaux, G., Gramfort, A., Michel, V., Thirion, B., Grisel, O., Blondel, M., Prettenhofer, P., Weiss, R., Dubourg, V. and Vanderplas, J.: Scikit-learn: Machine Learning in Python Fabian, J. Mach. Learn. Res., 12, 2825–2830, 2011.

Pokhrel, R. P., Wagner, N. L., Langridge, J. M., Lack, D. A., Jayarathne, T., Stone, E. A., Stockwell, C. E., Yokelson, R. J. and Murphy, S. M.: Parameterization of single-scattering albedo (SSA) and absorption Ångström exponent (AAE) with EC/OC for aerosol emissions from biomass burning, Atmos. Chem. Phys., 16(15), 9549–9561, doi:10.5194/acp-16-9549-2016, 2016.

Randerson, J. T., Chen, Y., Van Der Werf, G. R., Rogers, B. M. and Morton, D. C.: Global burned area and biomass burning emissions from small fires, J. Geophys. Res. Biogeosciences, 117(4), doi:10.1029/2012JG002128, 2012.

Roteta, E., Bastarrika, A., Franquesa, M. and Chuvieco, E.: Landsat and sentinel-2 based burned area mapping tools in google earth engine, Remote Sens., 13(4), 1–30, doi:10.3390/rs13040816, 2021.

Roy, D. P., Huang, H., Boschetti, L., Giglio, L., Yan, L., Zhang, H. H. and Li, Z.: Landsat-8 and Sentinel-2 burned area mapping - A combined sensor multi-temporal change detection approach, Remote Sens. Environ., 231(October 2018), 111254, doi:10.1016/j.rse.2019.111254, 2019.

Russell-smith, J., Yates, C., Vernooij, R., Eames, T., Werf, G. Van Der, Ribeiro, N., Edwards, A., Beatty, R., Lekoko, O., Mafoko, J., Monagle, C. and Johnston, S.: Opportunities and challenges for savanna burning emissions abatement in southern Africa, J. Environ. Manage., 288(March), 112414, doi:10.1016/j.jenvman.2021.112414, 2021.

Russell-Smith, J., Cook, G. D., Cooke, P. M., Edwards, A. C., Lendrum, M., Meyer, C. (Mick) and Whitehead, P. J.: Managing fire regimes in north Australian savannas: applying Aboriginal approaches to contemporary global problems, Front. Ecol. Environ., 11(s1), e55–e63, doi:10.1890/120251, 2013.

Schmidt, I. B., Moura, L. C., Ferreira, M. C., Eloy, L., Sampaio, A. B., Dias, P. A. and Berlinck, C. N.: Fire management in the Brazilian savanna: First steps and the way forward, J. Appl. Ecol., 55(5), 2094–2101, doi:10.1111/1365-2664.13118, 2018.

Selimovic, V., Yokelson, R. J., Warneke, C., Roberts, J. M., De Gouw, J., Reardon, J. and Griffith, D. W. T.: Aerosol optical properties and trace gas emissions by PAX and OP-FTIR for laboratory-simulated western US wildfires during FIREX, Atmos. Chem. Phys., 18(4), 2929–2948, doi:10.5194/acp-18-2929-2018, 2018.

Shea, R. W., Shea, B. W., Kauffman, J. B., Haskins, C. I. and Scholes, M. C.: in savanna ecosystems of South Africa and Zambia subjected of fire exclusion . intensity was only 1419 kW, , 101, 551–568, 1996.

Sinha, P., Hobbs, P. V., Yokelson, R. J., Blake, D. R., Gao, S. and Kirchstetter, T. W.: Emissions from miombo woodland and dambo grassland savanna fires, J. Geophys. Res. D Atmos., 109(11), doi:10.1029/2004JD004521, 2004.

Surawski, N. C., Sullivan, A. L., Meyer, C. P., Roxburgh, S. H. and Polglase, P. J.: Greenhouse gas emissions from laboratory-scale fires in wildland fuels depend on fire spread mode and phase of combustion, Atmos. Chem. Phys., 15(9), 5259–5273, doi:10.5194/acp-15-5259-2015, 2015.

Susott, R. A., Olbu, G., Baker, S. P., Ward, D. E., Kauffman, J. B. and Shea, R. W.: Carbon, hydrogen, nitrogen, and thermogravimetric analysis of tropical ecosystem biomass, in Biomass Burning and Global Change: Remote sensing, modeling and inventory Development and Biomass Burning in Africa, edited by J. S. Levine, pp. 350–360, MIT Press, Cambridge, Mass., 1996.

Tetens, O.: Uber einige meteorologische Begriffe, Zeitschrift fur Geophys., 6, 297–309, 1930.

Urbanski, S.: Forest Ecology and Management Wildland fire emissions , carbon , and climate : Emission factors, For. Ecol. Manage., 317, 51–60, doi:10.1016/j.foreco.2013.05.045, 2014.

Vermote, E.: MOD09A1 MODIS Surface Reflectance 8-Day L3 Global 500m SIN Grid V006, NASA EOSDIS L. Process. DAAC, doi:10.5067/MODIS/MOD09A1.006, 2015.

Vernooij, R.: Measurements of savanna landscap fire emission factors for CO2, CO, CH4 and N2O using a UAV-based sampling methodology, Zenodo [data set], doi:10.5281/zenodo.7689032, 2023.

Vernooij, R., Giongo, M., Assis Borges, M., Costa, M. M., Carolina Sena Barradas, A. and Van Der Werf, G. R.: Intraseasonal variability of greenhouse gas emission factors from biomass burning in the Brazilian Cerrado,

Biogeosciences, 18(4), 1375–1393, doi:10.5194/bg-18-1375-2021, 2021.

Vernooij, R., Winiger, P., Wooster, M., Strydom, T., Poulain, L., Dusek, U., Grosvenor, M., Roberts, G. J., Schutgens, N. and Werf, G. R. Van Der: A quadcopter unmanned aerial system (UAS)-based methodology for measuring biomass burning emission factors, Atmos. Meas. Tech., 15(July), 4271–4294, doi:10.5194/amt-15-4271-2022, 2022a.

Vernooij, R., Dusek, U., Popa, M. E., Yao, P., Shaikat, A., Qiu, C., Winiger, P., van der Veen, C., Eames, T. C., Ribeiro, N. and van der Werf, G. R.: Stable carbon isotopic composition of biomass burning emissions – implications for estimating the contribution of C3 and C4 plants, Atmos. Chem. Phys., 22(4), 2871–2890, doi:10.5194/acp-22-2871-2022, 2022b.

Vitolo, C., Di Giuseppe, F., Barnard, C., Coughlan, R., San-Miguel-Ayanz, J., Libertá, G. and Krzeminski, B.: ERA5-based global meteorological wildfire danger maps, Sci. Data, 7(1), 1–11, doi:10.1038/s41597-020-0554-z, 2020.

Van Wagner, C. E.: Development and structure of the Canadian forest fire weather index system. [online] Available from: http://scholar.google.com/scholar?hl=en&btnG=Search&q=intitle:Development+and+Structure+of+the+Canadian+Forest+Fire+Weather+Index+System#0, 1987.

Ward, D. E. and Radke, L. F.: Emissions Measurements from Vegetation Fires : A Comparative Evaluation of Methods and Results, in Fire in the Environment: The Ecological, Atmospheric, and Climatic Importance of Vegetation Fires, edited by P. J. . Crutzen and J. G. Goldammer, pp. 53–76, Chischester, England., 1993.

Ward, D. E., Susott, R. A., Kauffman, J. B., Babbitt, R. E., Cummings, D. L., Dias, B., Holben, B. N., Kaufman, Y. J., Rasmussen, R. A. and Setzer, A. W.: Smoke and fire characteristics for cerrado and deforestation burns in Brazil: BASE-B Experiment, J. Geophys. Res., 97(D13), 14601–14619, doi:10.1029/92JD01218, 1992.

van Wees, D. and van der Werf, G. R.: Modelling African biomass burning emissions and the effect of spatial resolution, Geosci. Model Dev. Discuss., 1–43, doi:10.5194/gmd-2019-116, 2019.

van der Werf, G. R., Randerson, J. T., Giglio, L., Van Leeuwen, T. T., Chen, Y., Rogers, B. M., Mu, M., Van Marle, M. J. E., Morton, D. C., Collatz, G. J., Yokelson, R. J. and Kasibhatla, P. S.: Global fire emissions estimates during 1997-2016, Earth Syst. Sci. Data, 9(2), 697–720, doi:10.5194/essd-9-697-2017, 2017.

Wiedinmyer, C., Kimura, Y., McDonald-Buller, E., Emmons, L. K., Buchholz, R. R., Tang, W., Seto, K., Joseph, M. B., Barsanti, K. C. and Yokelson, R.: The Fire Inventory from NCAR version 2.5: an updated global fire emissions model for climate and chemistry applications, Egusph. [preprint], 2019(February) [online] Available from: https://doi.org/10.5194/egusphere-2023-124, 2023.

Winter, F., Wartha, C. and Hofbauer, H.: The Relative Importance of Radicals on the N2O and NO Formation and Destruction Paths in a Quartz CFBC, J. Energy Resour. Technol., 121(2), 131–136, doi:10.1115/1.2795068, 1999.

Wooster, M. J., Freeborn, P. H., Archibald, S., Oppenheimer, C., Roberts, G. J., Smith, T. E. L., Govender, N., Burton, M. and Palumbo, I.: Field determination of biomass burning emission ratios and factors via open-path FTIR spectroscopy and fire radiative power assessment: Headfire, backfire and residual smouldering combustion in African savannahs, Atmos. Chem. Phys., 11(22), 11591–11615, doi:10.5194/acp-11-11591-2011, 2011.

Yokelson, R. J., Bertschi, I. T., Christian, T. J., Hobbs, P. V., Ward, D. E. and Hao, W. M.: Trace gas measurements in nascent, aged, and cloud-processed smoke from African savanna fires by airborne Fourier transform infrared spectroscopy (AFTIR), J. Geophys. Res. Atmos., 108(13), doi:10.1029/2002jd002322, 2003.

Yokelson, R. J., Burling, I. R., Urbanski, S. P., Atlas, E., Adachi, K., Buseck, P. R., Wiedinmyer, C., Akagi, S. K., Toohey, D. W. and Wold, C. E.: Trace gas and particle emissions from open biomass burning in Mexico, Atmos. Chem. Phys., 11(14), 6787–6808, doi:10.5194/acp-11-6787-2011, 2011.

## Tables

**Table 1: Measurement campaigns including the number of fires for which emission factors were measured as well as the number of corresponding fuel-transects.**

| Area | Timeframe | # Fires | # Fuel transects |
|---|---|---|---|
| Kruger National Park, South Africa | 29.08.2017 – 02.09.2017 | 3 | - |
| | 22.04.2018 – 28.04.2018 | 3 | - |
| | 21.08.2018 – 31.08.2018 | 8 | - |
| | 22.10.2018 – 26.10.2018 | 6 | - |
| Estação Ecológica Serra Geral do Tocantins, Brazil | 10.09.2017 – 20.09.2017 | 10 | - |
| | 15.06.2018 – 30.06.2018 | 11 | - |
| | 21.09.2018 – 12.10.2018 | 6 | - |
| North-west Ngamiland, Botswana | 21.05.2019 – 08.06.2019 | 5 | 39 |
| | 04.09.2019 – 15.09.2019 | 6 | 37 |
| Niassa special reserve Mozambique | 19.06.2019 – 09.07.2019 | 10 | 20 |
| | 05.10.2019 – 20.10.2019 | 11 | 24 |
| Kasane Extension Forest Reserve, Botswana | 12.10.2021 – 20.10.2021 | 2 | 42 |
| Bovu Forest Reserve, Zambia | 22.10.2021 – 26.10.2021 | 3 | 9 |
| Kafue national park, Zambia | 30.10.2021 – 12.11.2021 | 6 | 54 |
| | 15.06.2022 – 20.06.2022 | 5 | 24 |
| Lualaba Forest Reserve, Zambia | 21.06.2022 – 25.06.2022 | 5 | 60 |
| Tanami desert, Australia | 20.04.2022 – 28.04.2022 | 10 | 90 |
| | 12.08.2022 – 05.09.2022 | 6 | 24 |

**Table 2: Satellite reanalysis features assessed for the prediction of savanna biomass burning emission factors**

| | Parameter | Data source | Product reference | Spatial resolution | Temporal resolution | Feature range |
|---|---|---|---|---|---|---|
| **Vegetation parameters** | Fraction tree cover (FTC, %) | MODIS | MOD44BV006 (DiMiceli et al., 2015) | 500×500 meter | year$^{-1}$ | 0 – 53% |
| | Fraction bare soil cover (FBC, %) | MODIS | MOD44BV006 (DiMiceli et al., 2015) | 500×500 meter | year$^{-1}$ | 1 – 88% |
| | Time since the last fire (years) | MODIS | MCD64A1C6 (Giglio et al., 2018) | 500×500 meter | year$^{-1}$ | 1 – >10 years |
| | Normalized difference vegetation index (NDVI) before fire | MODIS | MOD09GAC6 (Vermote, 2015) | 500×500 meter | day$^{-1}$ | 0.02 – 0.79 |
| | Fraction of absorbed photosynthetically active radiation (FPAR) | MODIS | MCD15A2HC6 (Myneni et al., 2015) | 500×500 meter | 8 days$^{-1}$ | 0.09 – 0.75 |
| | Leaf area index (LAI) | MODIS | MCD15A2HC6 (Myneni et al., 2015) | 500×500 meter | 8 days$^{-1}$ | 2 – 30 |
| | Leaf area index (LAI) Low vegetation | Reanalysis | ERA5-Land (Muñoz-Sabater et al., 2021) | 0.1×0.1 degree | day$^{-1}$ | 0.5 – 2.0 |
| | Leaf area index (LAI) High vegetation | Reanalysis | ERA5-Land (Muñoz-Sabater et al., 2021) | 0.1×0.1 degree | day$^{-1}$ | 0.0 – 5.0 |
| | Mean annual rainfall (MAR)(mm) | Reanalysis | ERA5-Land (Muñoz-Sabater et al., 2021) | 0.1×0.1 degree | month$^{-1}$ | 200 – 1550 |
| **Seasonal parameters** | Rainfall in the last 12 months (mm) | Reanalysis | ERA5 (Hersbach et al., 2020) | 0.25×0.25 degree | month$^{-1}$ | 220 – 1550 |
| | Rainfall since the last fire (mm) | Reanalysis | ERA5 (Hersbach et al., 2020) | 0.25×0.25 degree | month$^{-1}$ | 220 – 11300 |
| | Percentage green vegetation (%) (Korontzi, 2005) | MODIS | MOD09GAC6 (Vermote, 2015) | 500×500 meter | day$^{-1}$ | 2 – 89 |
| | Soil moisture content (m$^3$ m$^{-3}$) in the top layer (0 - 7cm depth) | Reanalysis | ERA5-Land (Muñoz-Sabater et al., 2021) | 0.1×0.1 degree | hour$^{-1}$ | 0.01 – 0.43 |
| | Vapor pressure deficit (mbar) | Reanalysis | ERA5-Land (Muñoz-Sabater et al., 2021) | 0.1×0.1 degree | hour$^{-1}$ | 8 – 51 |
| | Evaporative stress index (index) | Reanalysis | ERA5-Land (Muñoz-Sabater et al., 2021) | 0.1×0.1 degree | hour$^{-1}$ | 0.02 – 0.73 |
| **Weather** | Temperature at 2m (°C) | Reanalysis | ERA5-Land (Muñoz-Sabater et al., 2021) | 0.1×0.1 degree | hour$^{-1}$ | 16 – 36 |
| | Windspeed (m sec$^{-1}$) | Reanalysis | ERA5-Land (Muñoz-Sabater et al., 2021) | 0.1×0.1 degree | hour$^{-1}$ | 0 – 11.2 |
| | Relative humidity (%) | Reanalysis | ERA5-Land (Muñoz-Sabater et al., 2021) | 0.1×0.1 degree | hour$^{-1}$ | 8 – 71 |
| | Canadian Fire Weather Index (FWI) | Reanalysis | CEMS EFFIS (Vitolo et al., 2020) | 0.25×0.25 degree | day$^{-1}$ | 10 – 102 |
| | Fine Fuel Drought Code (FFDC) | Reanalysis | CEMS EFFIS (Vitolo et al., 2020) | 0.25×0.25 degree | day$^{-1}$ | 81 – 99 |
| | Initial Spread Index (ISI) | Reanalysis | CEMS EFFIS (Vitolo et al., 2020) | 0.25×0.25 degree | day$^{-1}$ | 1.9 – 47.5 |
| **Fire intensity indices** | Build up index (BUI) | Reanalysis | CEMS EFFIS (Vitolo et al., 2020) | 0.25×0.25 degree | day$^{-1}$ | 64 – 624 |
| | Differential normalized difference vegetation index (dNDVI) | MODIS | MOD09GAC6 (Vermote, 2015) | 500×500 meter | day$^{-1}$ | -0.43 – 0.61 |
| | Differential normalized burn ratio (dNBR) | MODIS | MOD09GAC6 (Vermote, 2015) | 500×500 meter | day$^{-1}$ | -0.25 – 0.45 |

**Table 3: Consumption of RSC-prone fuels in the EDS and LDS for xeric open savannas measured in Botswana and Australia and Miombo woodlands measured in Mozambique and Zambia.**

| Field measurements before and after burning | Xeric savannas (500 - 750 mm year$^{-1}$ MAR) | | | | Mesic savannas (750 - 1500 mm year$^{-1}$ MAR) | | | |
|---|---|---|---|---|---|---|---|---|
| | Australian arid open woodland | | Kalahari open woodland | | Kafue woodland savanna | | Niassa woodland savanna | |
| | EDS | LDS | EDS | LDS | EDS | LDS | EDS | LDS |
| Fine fuel load (tonne ha$^{-1}$) | 5.1 | 6.7 | 3.1 | 3.4 | 2.8 | 5.9 | 6.3 | 5.4 |
| Grass percentage of total fine fuel (i.e. grass, litter and coarse) | 76% | 79% | 27% | 25% | 24% | 17% | 45% | 35% |
| Nitrogen to Carbon ratio[1] | 1.0% | 0.8% | 2.3% | 2.1% | - | 1.7% | 1.3% | 1.1% |
| Time since last fire (years) Based on MCD64A1C5 | 6.7 | 6.0 | 2.9 | 2.8 | 1.7 | 1.6 | 1.7 | 1.3 |
| WA Carbon content[1] | 45.2% | 44.6% | 49.1% | 47.8% | - | 46.5% | 43.5% | 46.2% |
|     Grass | 45.1% | 43.9% | 47.6% | 47.5% | - | 47.0% | 43.0% | 44.0% |
|     Litter | 45.2% | 47.0% | 50.1% | 48.0% | - | 46.7% | 43.2% | 47.2% |
|     Coarse woody debris | 48.1% | 48.0% | 48.2% | 47.7% | - | 44.7% | 47.2% | 47.8% |
|     Shrub stems[2] | - | 47.1% | 47.9% | - | - | 47.5% | - | 48.2% |
|     Shrub foliage[2] | - | 50.0% | 50.3% | - | | 51.6% | - | 50.7% |
| WA Nitrogen content | 0.45% | 0.37% | 1.11% | 1.00% | - | 0.81% | 0.55% | 0.52% |
|     Grass | 0.46% | 0.31% | 1.06% | 0.65% | - | 0.42% | 0.34% | 0.30% |
|     Litter | 0.43% | 0.65% | 1.22% | 1.17% | - | 0.92% | 0.73% | 0.65% |
|     Coarse woody debris | 0.33% | 0.48% | 0.89% | 0.69% | - | 0.61% | 0.42% | 0.48% |
|     Shrub stems[2] | - | 0.63% | 1.10% | - | - | 0.65% | - | 0.52% |
|     Shrub foliage[2] | - | 1.03% | 2.55% | - | | 2.02% | - | 1.13% |
| Relative humidity (air) | 18% | 10% | 13% | 6% | 22% | 17% | 24% | 19% |
| Fuel moisture content[1] | 15.6% | 8.6% | 20.3% | 8.7% | 16.5% | 6.5% | 16.6% | 8.8% |
| Fine fuel combusted | 93% | 97% | 69% | 75% | 58% | 77% | 60% | 71% |
| Coarse fuel combusted (∅ < 5cm) | 21% | 17% | 21% | 16% | 4% | 26% | 2% | 19% |
| Heavy fuels combusted (∅ > 5cm) | 76% | 32% | 3% | 35% | 0% | 16% | 2% | 8% |
| 0-50 Cm shrubs combusted: | | | | | | | | |
|     Leaves[2] | 72% | 86% | 50% | 60% | 17% | 79% | 20% | 71% |
|     Stems | 60% | 65% | 38% | 88% | 1% | 44% | 24% | 40% |
| 50-100 Cm shrubs combusted: | | | | | | | | |
|     Leaves[2] | 51% | 78% | 26% | 48% | 7% | 43% | 46% | 55% |
|     Stems[2] | 60% | 65% | 19% | 8% | 0% | 15% | 3% | 20% |
| 100-200 Cm shrubs combusted: | | | | | | | | |
|     Leaves | 26% | 69% | 22% | 31% | 0% | 47% | 20% | 35% |
|     Stems | 7% | 13% | 4% | 3% | 0% | 5% | 4% | 11% |
| >200 Cm shrubs combusted: | | | | | | | | |
|     Leaves | 33% | 36% | 10% | 16% | 0% | 10% | 7% | 43% |
|     Stems | 5% | 7% | 0% | 1% | 0% | 3% | 2% | 4% |
| Scorch height (m) | 2.0 m | 2.2 m | 0.4 m | 0.4 m | 0.3 m | 10.3 m | 0.5 m | 1.7 m |
| Char height (m) | 0.9 m | 1.1 m | 0.2 m | 0.3m | 0.2 m | 0.9 m | 0.4 m | 1.6 m |
| Patchiness (% burned) | 69% | 94% | 51% | 72% | 54% | 99% | 63% | 95% |

[1] weighted average over the consumed contribution of each individual fuel subclass.
[2] weighted average over the dominant shrub types found in the plots.

**Table 4: Emission factor averages for the global savanna**

| EF Specie | GFED4s | Andreae (2019) | Wiedinmeijer et al. (2023) | Sample data avg.[1] | Training data avg.[2] | Effective EF (Eq. 1)[3] |
|---|---|---|---|---|---|---|
| $CO_2$ | 1686 | 1660 | 1686 | 1637 | 1670 | 1685 |
| CO | 63 | 69 | 63 | 55 | 61 | 64 |
| $CH_4$ | 1.94 | 2.70 | 2.00 | 1.38 | 1.61 | 1.85 |
| $N_2O$ | 0.20 | 0.17 | | 0.12 | 0.12 | 0.16 |

[1]Averaged over the fires measured using the drone methodology (skewed towards xeric savannas)

[2]Averaged over the fires measured using the drone methodology and the included literature studies.

[3]Dynamic EFs weighted by the consumed biomass at time and location of fires as calculated using GFED4s.

**Table 4. Spearman correlation matrix for the field-measured-ecosystem attributes and the fire-averaged emission factors and MCE as well as the satellite products used in the study. Positive correlations are presented in blue while negative correlations are presented in red.**

| | MCE | CH₄ emission factor | CO emission factor | CO₂ emission factor | N₂O emission factor | Years since last burn (MODIS) | Fraction bare soil (%) (MODIS) | Fraction tree cover (%) (MODIS) | Vapor pressure deficit (mbar) (ERA5-land) | Relative humidity (%) (ERA5-land) | Evaporative stress index (ratio) (ERA5-land) | windspeed (m/s) (ERA5-land) | Temperature (°C) (ERA5-land) | Soil moisture (m³ m⁻³) (ERA5-land) | mean annual rainfall (mm) (ERA5-land) | last years rainfall (mm) (ERA5) | rainfall since last fire (mm) (ERA5) | pre-fire NDVI (MODIS) | dNDVI (MODIS) | dNBR (MODIS) | Pgreen (MODIS) | Fire weather index (ERA5) | Build up index (ERA5) | initial spread index (ERA5) | Fine fuel moisture code (ERA5) | FPAR (%) (MODIS) | LAI (MODIS) | LAI high vegetation (ERA5-land) | LAI low vegetation (ERA5-land) |
|---|---|---|---|---|---|---|---|---|---|---|---|---|---|---|---|---|---|---|---|---|---|---|---|---|---|---|---|---|---|
| Temperature (°C) before fire | 0.04 | 0.15 | -0.02 | 0.16 | 0.24 | -0.32 | -0.12 | -0.13 | 0.18 | 0.01 | 0.06 | 0.04 | 0.21 | -0.05 | -0.19 | -0.11 | -0.35 | -0.02 | 0.07 | 0.3 | -0.05 | 0.16 | 0.16 | 0.21 | 0.22 | -0.13 | -0.06 | -0.01 | -0.23 |
| Rel Humidity (%) before fire | -0.46 | 0.32 | 0.42 | -0.44 | 0.06 | -0.08 | -0.35 | 0.54 | -0.21 | 0.45 | 0.38 | -0.36 | 0.03 | 0.43 | 0.63 | 0.51 | 0.1 | 0.3 | 0.09 | -0.29 | -0.18 | -0.64 | -0.67 | -0.08 | -0.67 | 0.5 | 0.49 | 0.55 | 0.64 |
| Fuel moisture content (%) | 0.48 | 0.47 | 0.37 | -0.64 | -0.01 | 0.09 | 0.32 | -0.24 | -0.35 | 0.18 | -0.3 | 0.02 | -0.2 | -0.21 | -0.09 | -0.26 | 0.1 | 0.07 | 0.01 | -0.18 | 0.19 | -0.17 | -0.17 | -0.11 | -0.25 | 0.01 | -0.09 | -0 | -0.02 |
| Trees per hectacte | -0.49 | 0.47 | 0.42 | -0.37 | -0.19 | -0.24 | -0.75 | 0.64 | -0.05 | 0.33 | 0.75 | -0.19 | 0.11 | 0.57 | 0.55 | 0.6 | 0.09 | 0.27 | -0.04 | 0.04 | -0.38 | -0.43 | -0.43 | -0.06 | -0.41 | 0.52 | 0.52 | 0.4 | 0.57 |
| Average tree height (m) | -0.4 | 0.39 | 0.37 | -0.16 | 0.09 | -0.13 | -0.63 | 0.51 | 0.13 | 0.19 | 0.62 | -0.21 | 0.33 | 0.46 | 0.43 | 0.45 | 0.03 | 0.4 | 0.07 | -0.05 | -0.03 | -0.38 | -0.36 | 0.09 | -0.36 | 0.5 | 0.42 | 0.37 | 0.51 |
| % grass of total dry Fine fuel | 0.27 | -0.32 | -0.33 | -0.37 | -0.51 | 0.16 | 0.47 | -0.28 | -0.27 | 0 | -0.46 | 0.35 | -0.33 | -0.24 | -0.15 | -0.08 | 0.14 | 0.06 | 0.14 | -0.31 | 0.06 | 0.14 | 0.14 | -0.34 | 0.03 | -0.12 | -0.16 | -0.18 | -0.21 |
| Total fine fuel (tonne ha⁻¹) | -0.04 | 0.16 | -0.03 | -0.47 | -0.26 | -0.2 | 0.09 | -0.04 | -0.26 | 0.28 | -0.05 | 0.31 | -0.21 | 0.14 | -0.02 | 0.2 | -0.14 | -0.07 | -0.07 | 0.04 | -0.13 | 0.02 | 0.02 | -0.31 | -0.08 | 0.04 | 0.09 | 0.07 | -0.1 |
| Combustion completeness fine fuel (%) | 0.58 | -0.5 | -0.6 | -0.02 | -0.4 | 0.31 | 0.65 | -0.59 | 0.17 | -0.38 | -0.67 | 0.58 | -0.07 | -0.5 | -0.6 | -0.41 | 0.11 | -0.36 | -0.01 | -0.01 | 0.02 | 0.72 | 0.71 | 0.01 | 0.61 | -0.62 | -0.56 | -0.6 | -0.65 |
| Combustion completeness coarse fuel (%) | 0.17 | -0.08 | -0.14 | 0.38 | -0.01 | 0.07 | 0.01 | -0.16 | 0.25 | -0.15 | -0.13 | 0.03 | 0.18 | -0.25 | -0.29 | -0.27 | -0.01 | -0.34 | -0.11 | 0.18 | -0.14 | 0.38 | 0.36 | 0.34 | 0.39 | -0.32 | -0.24 | -0.15 | -0.33 |
| Combustion completeness heavy fuel (%) | 0.19 | -0.2 | -0.2 | 0.06 | -0 | 0.13 | 0.32 | -0.24 | 0.12 | -0.16 | -0.35 | 0.11 | 0.11 | -0.24 | -0.26 | -0.26 | 0.01 | -0.23 | 0.04 | 0.27 | -0.02 | 0.33 | 0.31 | 0.14 | 0.39 | -0.4 | -0.29 | -0.37 | -0.42 |
| Char height (cm) | 0.34 | -0.19 | -0.38 | -0.15 | -0.33 | 0.04 | 0.22 | -0.25 | 0.1 | -0.02 | -0.18 | 0.38 | 0.03 | -0.24 | -0.35 | -0.11 | 0 | -0.22 | -0.01 | 0.18 | -0.14 | 0.37 | 0.33 | -0.03 | 0.3 | -0.19 | -0.16 | -0.12 | -0.42 |
| Scorch height (cm) | 0.41 | -0.13 | -0.42 | 0.06 | -0.27 | -0.03 | 0.1 | -0.29 | 0.47 | -0.1 | -0.11 | 0.35 | 0.36 | -0.22 | -0.39 | -0.13 | -0.07 | -0.1 | 0.02 | -0.03 | -0.12 | 0.53 | 0.5 | 0.29 | 0.48 | -0.16 | -0.12 | -0.2 | -0.5 |
| Patchiness (Surface percentage burned) | 0.01 | 0.11 | -0.09 | -0.49 | -0.01 | -0.35 | -0.22 | 0.28 | -0.33 | 0.44 | 0.33 | -0.02 | -0.22 | 0.31 | 0.27 | 0.44 | -0.16 | 0.12 | 0.15 | -0.03 | -0.43 | -0.2 | -0.23 | -0.3 | -0.19 | 0.26 | 0.34 | 0.42 | 0.22 |
| Nitrogent to Carbon (ratio) | -0.16 | 0.18 | 0.25 | 0.61 | 0.65 | -0.15 | -0.4 | 0.18 | 0.29 | -0.1 | 0.36 | -0.45 | 0.39 | 0.18 | 0.08 | -0.08 | -0.18 | 0.18 | -0.07 | 0.17 | 0.24 | -0.15 | -0.13 | 0.45 | -0.01 | 0.19 | 0.16 | 0.02 | 0.17 |
| nitrogen content grass | 0.22 | -0.2 | -0.15 | 0.55 | 0.3 | 0.29 | 0.19 | -0.36 | 0.55 | -0.35 | -0.28 | -0.08 | 0.58 | -0.32 | -0.38 | -0.59 | 0.09 | 0.01 | -0.06 | 0.02 | 0.44 | 0.26 | 0.24 | 0.63 | 0.29 | -0.18 | -0.26 | -0.42 | -0.32 |
| carbon content grass | 0.1 | -0.06 | 0.02 | 0.77 | 0.47 | 0.09 | -0.13 | -0.1 | 0.61 | -0.4 | -0.01 | -0.18 | 0.54 | -0.08 | -0.2 | -0.36 | -0.09 | 0.08 | 0.06 | 0.12 | 0.34 | 0.26 | 0.26 | 0.67 | 0.37 | -0.15 | -0.14 | -0.34 | -0.13 |
| nitrogen litter grass | -0.15 | 0.18 | 0.24 | 0.51 | 0.66 | -0.1 | -0.36 | 0.15 | 0.24 | -0.22 | 0.39 | -0.43 | 0.22 | 0.26 | 0.1 | 0.08 | -0.18 | 0.26 | 0.02 | 0.15 | 0.37 | -0.11 | -0.08 | 0.24 | 0.09 | 0.22 | 0.21 | 0.05 | 0.22 |
| carbon content litter | 0.09 | -0.13 | 0.03 | 0.58 | 0.36 | -0.09 | 0.11 | -0.09 | -0.05 | -0.3 | -0.12 | -0.12 | -0.16 | -0.08 | -0.07 | -0.12 | -0.23 | -0.25 | -0.06 | 0.25 | 0.11 | 0.14 | 0.16 | 0.11 | 0.22 | -0.31 | -0.22 | -0.12 | -0.01 |
| average Carbon content | -0.08 | 0.09 | 0.21 | 0.75 | 0.44 | -0.04 | -0.16 | 0.03 | 0.38 | -0.26 | 0.05 | -0.2 | 0.36 | 0.01 | -0.03 | -0.2 | -0.15 | -0.15 | -0.13 | 0.3 | 0.14 | 0.1 | 0.1 | 0.51 | 0.2 | -0.15 | -0.11 | -0.15 | 0 |
| average Nitrogen content | -0.18 | 0.2 | 0.27 | 0.57 | 0.63 | -0.12 | -0.38 | 0.17 | 0.38 | -0.12 | 0.32 | -0.42 | 0.48 | 0.17 | 0.09 | -0.09 | -0.16 | 0.12 | -0.05 | 0.21 | 0.2 | -0.11 | -0.1 | 0.53 | 0.03 | 0.15 | 0.15 | 0.03 | 0.16 |

## Figures

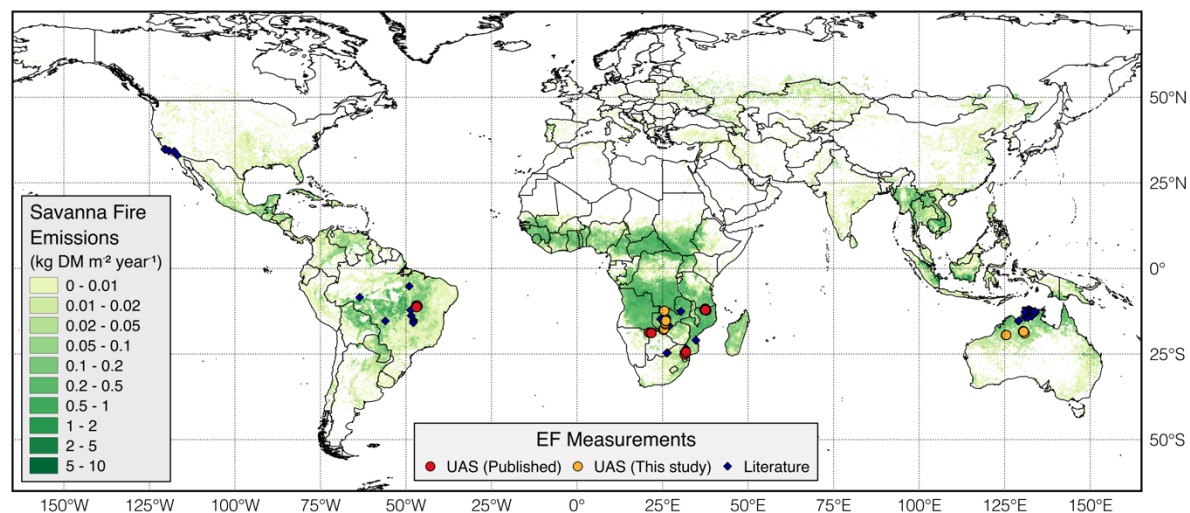

**Figure 1. Overview of sampling locations used for the analysis. The previously published (red) and new (orange) UAS measurements as well as the locations of the included literature studies on savanna fire emission factors listed in Andreae, 2019 (blue). The green shaded area shows the distribution of savanna and grassland fires over the 2002-2016 period according to GFED4s.**

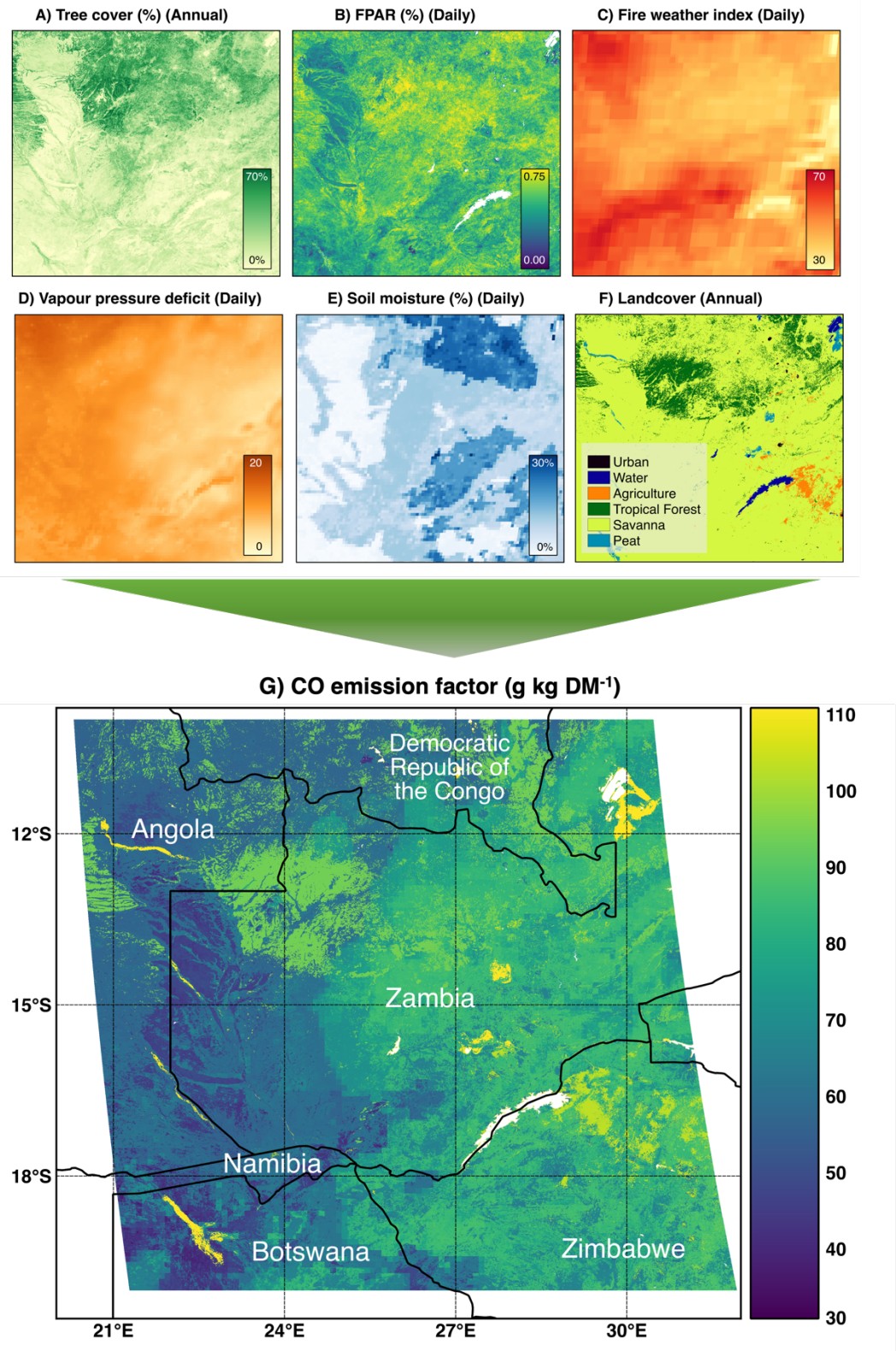

**Figure 2: Estimation of the CO EF at 500-meter resolution for MODIS tile "h20v10" on June 1st, 2019 (g), using a random forest regression based on (a) fractional tree cover (FTC), (b) fraction of absorbed photosynthetically active radiation (FPAR), (c) the fire weather index (FWI), (d) vapour pressure deficit (VPD) and (e) soil moisture. For grid cells containing other biomes than savanna (f), GFED4s static EFs for the respective biome were imposed replacing the savanna EFs. Sources of the individual features are listed in Table 2.**

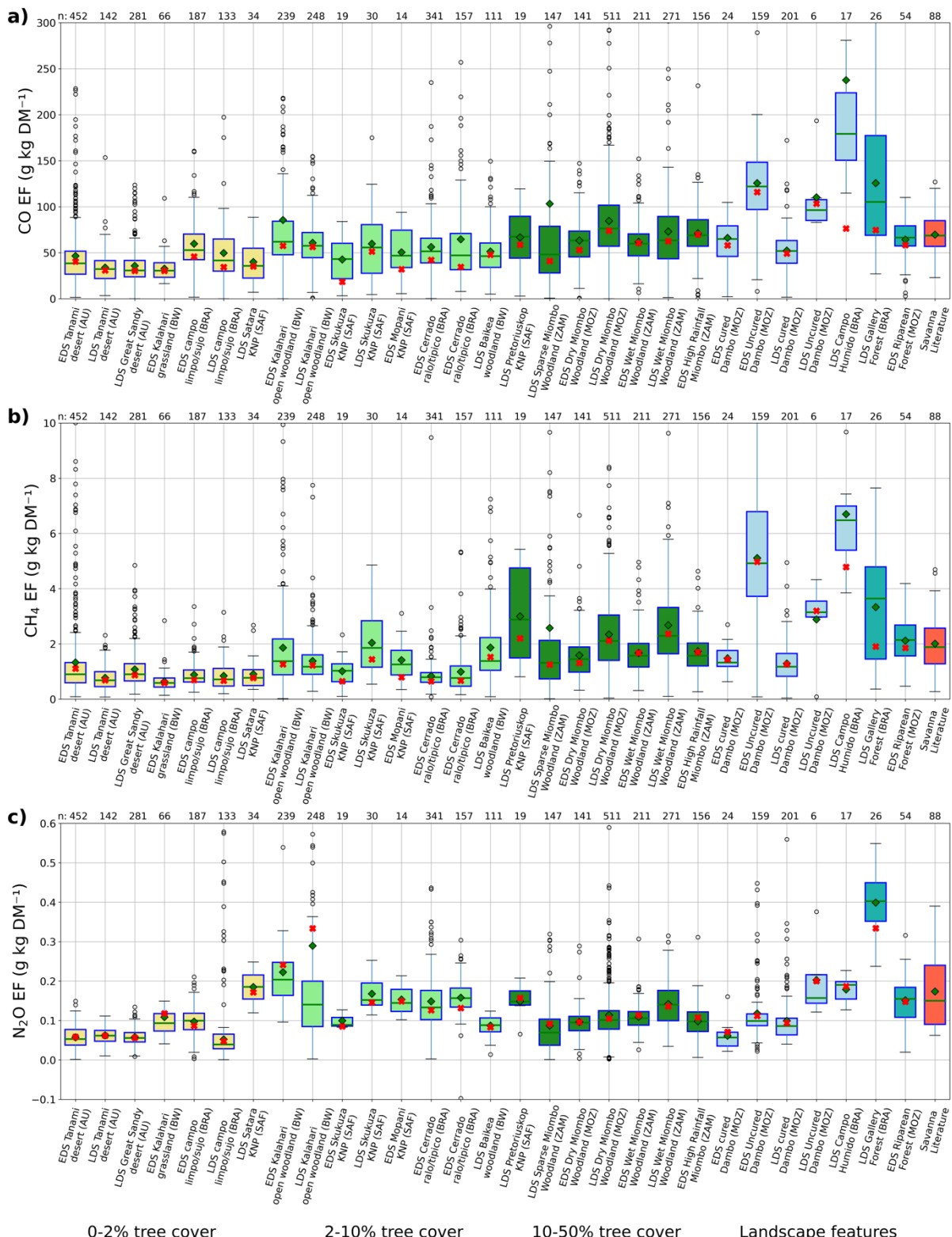

**Figure 3: EFs (g kg DM$^{-1}$) measured in the sampled vegetation types during the EDS and LDS as well as the EFs from savanna measurements listed in savanna literature based on the Andreae (2019) compilation. The green diamond represents the arithmetic mean, and the red cross represents the EMR-weighted average value. The colours correspond to the savanna subclasses on the bottom of the figure. Table 1 lists the timeframes of the individual field campaigns while Table A1 in the appendix provides a broad floristic description of the dominant vegetation types.**

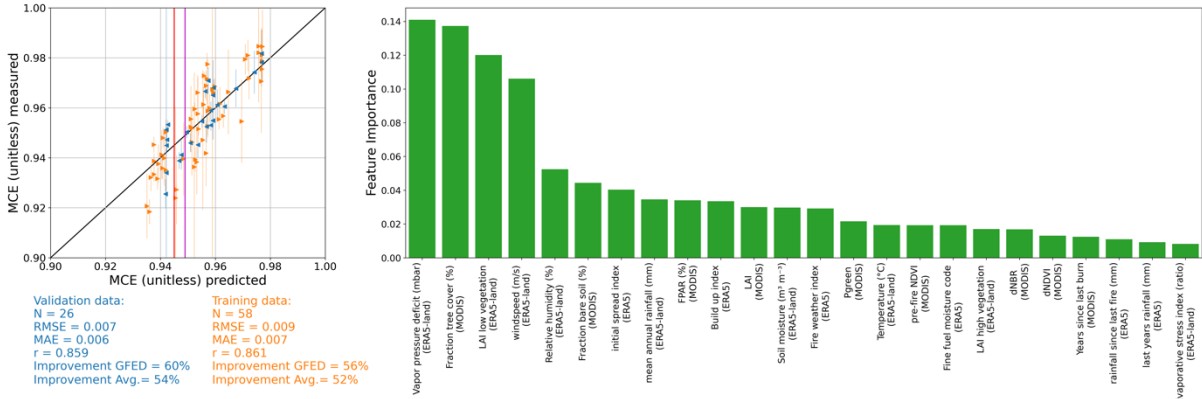

**Figure 4. Left: Correlation of the predicted and measured fire-integrated weighted average MCE for the training (orange) and validation (blue) datasets. The vertical blue and orange lines represent the standard error of the mean within the respective fire. The red vertical line is the static MCE derived from the EFs used in GFED4s. The 'improvement' refers to the reduced mean absolute error compared to prediction using this static GFED4 (red line) MCE and compared to the average of the input data (magenta line). Right: The remote sensing and reanalysis datasets used by the model and the feature importance (an indication of how strong each feature is used to differentiate the data) of the respective features.**

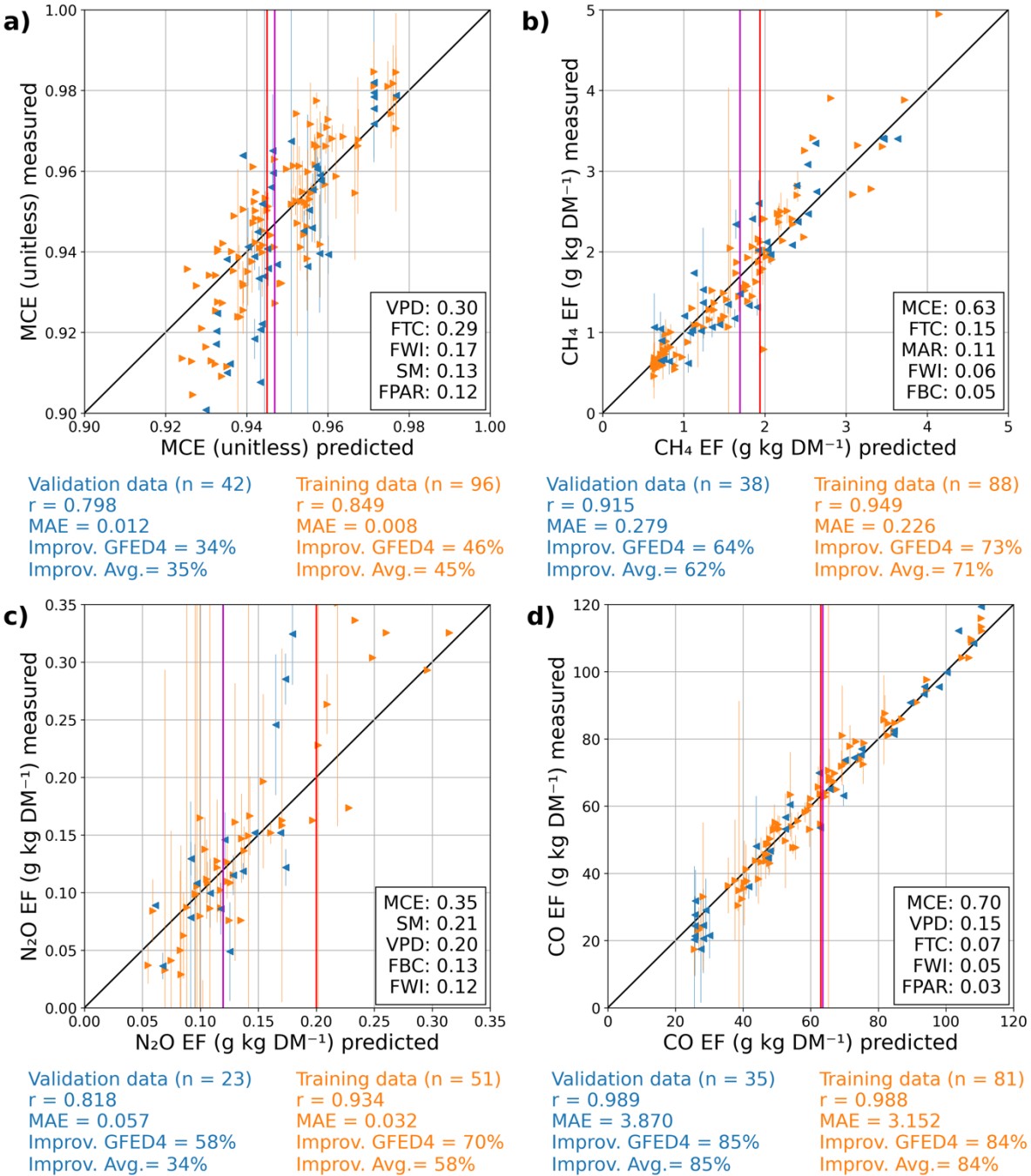

**Figure 5. Pearson correlation of the predicted and measured fire-integrated WA MCE (a), CH₄ EF (b) N₂O EF (c), and CO EF (d) for the training (orange) and validation (blue) datasets using a limited set of features. The boxes in the bottom right of the panels list the remote sensing and reanalysis datasets used by the model and the feature importance (an indication of how strong each feature is used to differentiate the data). The red line represents the static biome-average used in GFED4s while the magenta line represents the average of the training and validation data. 'improv. GFED' refers to the reduced mean absolute error compared to the static average used by GFED4s and the 'improv. Avg.' refers to the reduced mean absolute error compared to the static average of the input data.**

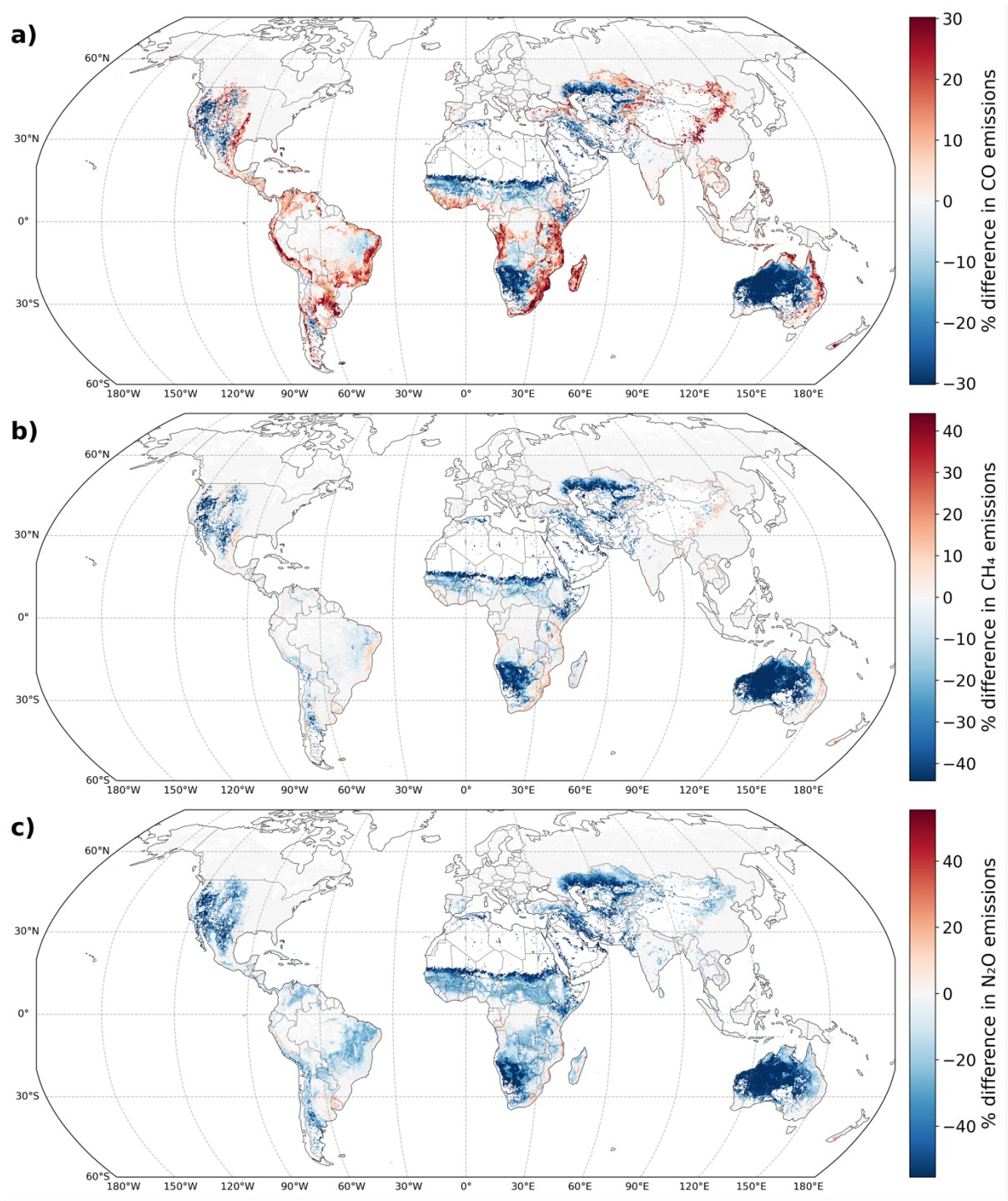

**Figure 6: Difference in savanna and grassland fire emissions for CO (a), CH₄ (b) and N₂O (c) between emission computation using dynamic EFs versus static biome reference EFs (dynamic minus static), calculated using GFED4s for the 2002-2016 period.**

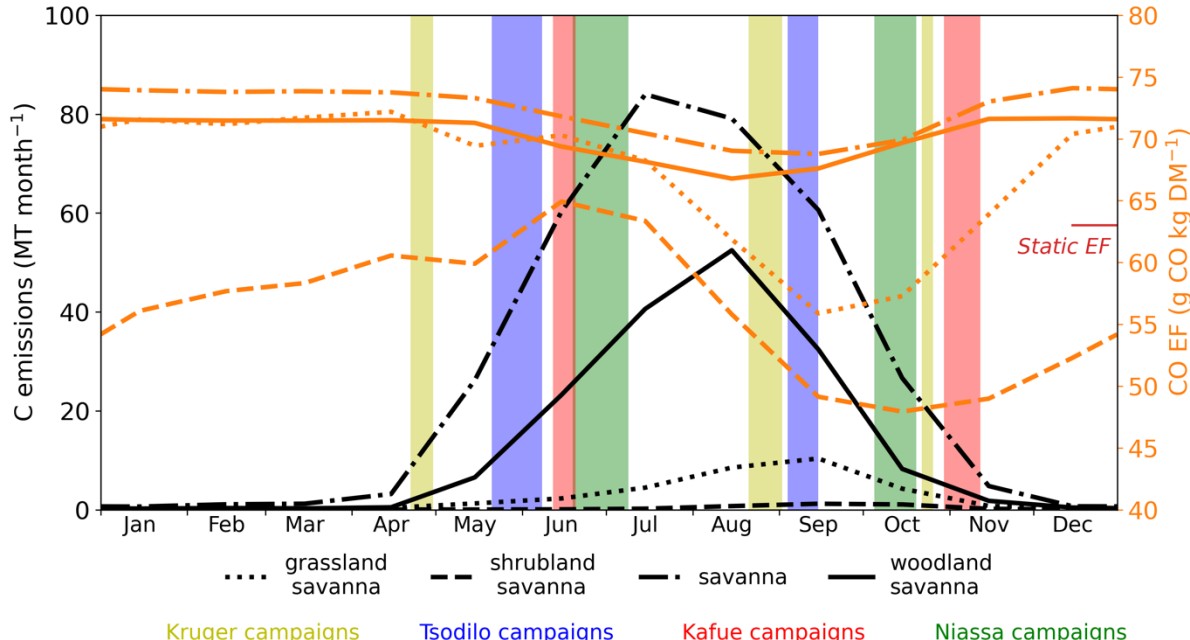

**Figure 7. Seasonality of fire carbon emissions (black) and the computed CO EF (orange) for different savanna subclasses in southern hemisphere Africa, averaged over the 2002-2016 period. The savanna classes are based on the International Geosphere-Biosphere Program (IGBP) classification (Loveland and Belward, 1997). The shaded areas**
5    **represent the timing of our measurements in southern hemisphere African savannas, indicating that especially our LDS campaigns may not be representative for the bulk of the fires. The red horizontal bar on the right represents the static EF used for savannas by GFED4s.**

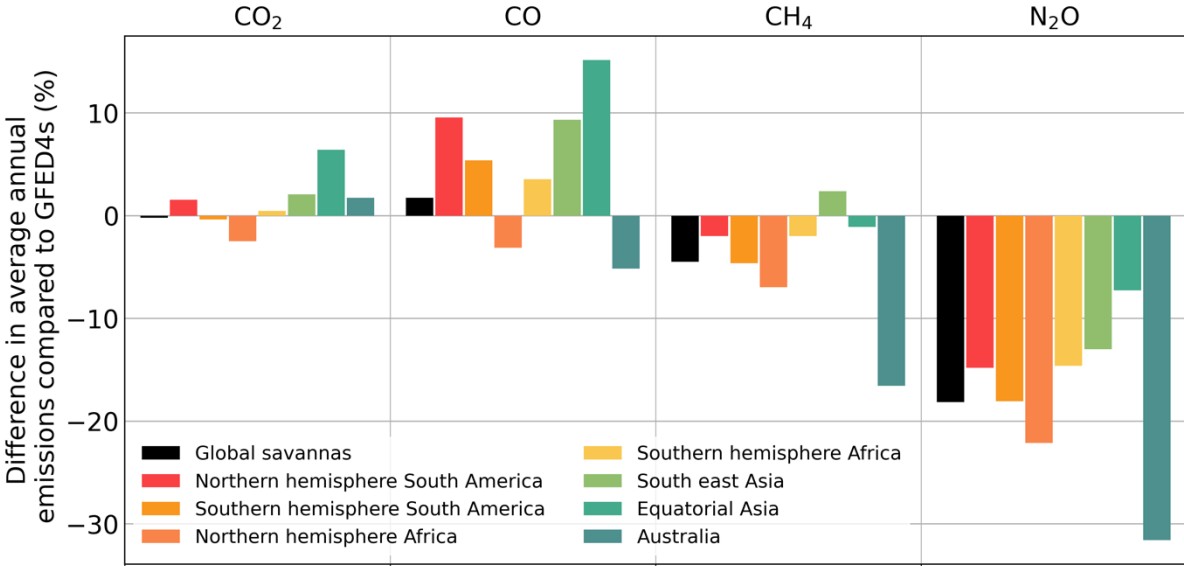

**Figure 8: Relative difference in the landscape fire emissions of CO₂, CO, CH₄ and N₂O for the 2002-2016 period when**
10    **using dynamic EFs versus static EFs using GFED4s (dynamic minus static) over the different savanna-rich GFED regions. Note that many of these regions encompass both xeric and mesic savannas with contrasting patterns that balance each other out. On a regional scale differences may therefore be much larger.**

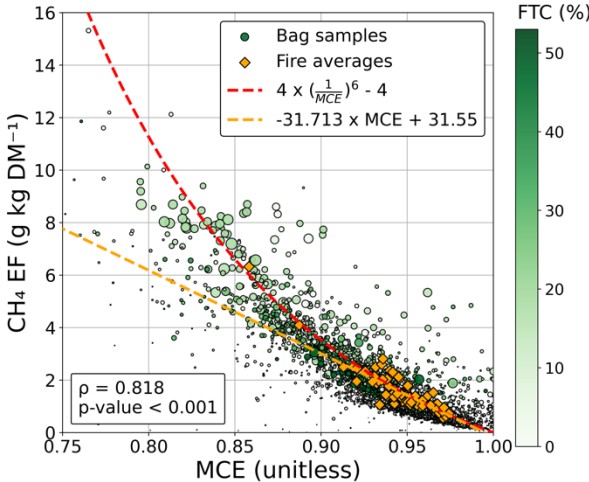

**Figure 9. The non-linear regression between the CH₄ EF and the MCE for the individual bag samples (green circles) and the fire averaged values (orange Diamonds). In the box on the bottom left, ρ refers to Spearman's rank correlation coefficient for the bag samples.**

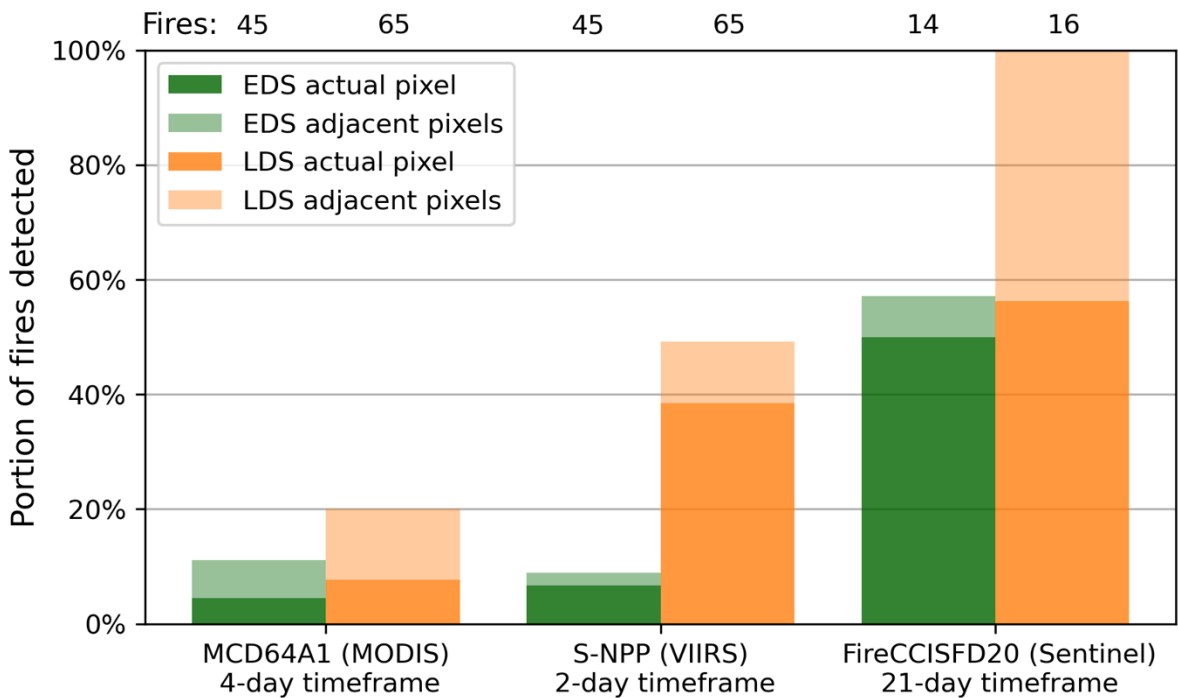

**Figure 10. Detection rate of the fires measured using the UAS-methodology by different satellite algorithms in the EDS (green) and LDS (orange). The darker area represents the cases where a fire was observed in the actual pixel within the listed timeframe. The lighter areas represent fires that were not detected in the same pixel as the samples but were detected in adjacent pixels. Timeframes are listed below the product labels. For the VIIRS detections the distance limits between the detection point and closest sample of the fire were 1km for the darker shaded area and 3.5 km for the lighter shaded area.**

## Appendix

**Table A1: Floristic and geomorphological description of the different vegetation types measured in this study.**

| Vegetation type (Fig. 3) | Vegetation description[1,2] | Satellite value range in the plots |
|---|---|---|
| Dambo grasslands Niassa Special reserve, Mozambique | Landscape feature limited to more humid and fertile places, containing seasonally inundated grassland savanna dominated by perennial tussock grasses e.g. beard grass (*Andropogon* (**PE**)) and thatching grass (*Hyparrhenia (A)*) with sparse Bushwillow (*Combretum* (**D**)) trees and on clayey swales with highly variably water tables based on geomorphology and soil type (Mbanze et al. 2019). | MAR: 1000–1100mm FTC: 10 – 20% FBC: 10 – 30% |
| Dry Miombo woodlands Niassa Special reserve, Mozambique | Dry Miombo Woodland dominated by (5-15m) Semi-deciduous Miombo (*Brachystegia (SD)*) and Mnondo (*Julbernardia (D)*) trees on sandy soils (Ribeiro et al., 2013, 2008). | MAR: 850 – 1100mm FTC: 15 – 30% FBC: 10 – 25% |
| Wet Miombo woodlands, Kafue National Park, Zambia | Savanna open forest dominated by (5-15m) *Brachystegia (SD))*, *Julbernardia (D)*, and *Isoberlinia (D)* trees on sandy soils. | MAR: 850 – 1300mm FTC: 10 – 35% FBC: 0 – 10% |
| Sparse Miombo Woodlands, Bovu Forest reserve, Zambia | Savanna open woodland containing perennial tussock grasses e.g. digitgrass (*Digitaria (PE))* and Tangleheads (*Heteropogon (A)*) with (5-15m) *Combretum (D)*, *Albizia (D)* and *Diospyros (EG)* trees on sandy soils. | MAR: 800 – 900mm FTC: 5 – 15% FBC: 0 – 15% |
| Baikea woodland, Kasane Extension Forest Reserve, Botswana | Open woodland savanna dominated by tussock perennial grasses e.g. digitgrass (*Digitaria eriantha (PE)*) and sickle grass (*Pogonarthria squarrosa(PE)*) with scattered (5–15m) African teak (*Baikiaea plurijuga (D)*) and silver cluster-leaf (*Terminalia sericea (D)*) trees on sandy soils. | MAR: 700 – 800mm FTC: 5 – 10% FBC: 5 – 20% |
| Satara experimental burn plots, Kruger National Park, South Africa | Grassland savanna dominated by perennial tussock grasses e.g. Sabi grass (*Urochloa mosambicensis (PE)*) and digitgrass (*Digitaria eriantha (PE)*) with scattered tall (10–15m) Marula (*Sclerocarya birrea (D)*) and knobthorn Acacia (*Acacia nigrescens (D)*) trees on clay soils overlying basalt plains (Venter and Govender, 2012). | MAR: 400 – 550mm FTC: 0 – 5% FBC: 10 – 30% |
| Skukuza experimental burn plots, Kruger National Park, South Africa | Savanna woodland dominated by dense Bushwillow (*Combretum collinum (D)/ Combretum zeyheri (D)*) trees on hydromorphic or duplex soils containing granite outcrops (Venter and Govender, 2012). | MAR: 500 – 600mm FTC: 3 – 10% FBC: 25 – 30% |
| Mopani experimental burn plots, Kruger National Park, South Africa | Savanna shrubland dominated by dense low (1–4m) mopane (*Colophospermum mopane (D)*) shrubs on flat or slightly sloping clay soils. (Venter and Govender, 2012). | MAR: 300 – 450mm FTC: 0 – 10% FBC: 30 – 50% |
| Pretoriuskop experimental burn plots, Kruger National Park, South Africa | Open forest savanna dominated by dense tall (10–15m) clusterleaf (*Terminalia sericea (D)*) and (5-10m) Sicklebush (*Dichrostachys cinerea (SD)*) trees on sandy soils. (Venter and Govender, 2012). | MAR: 800 – 900mm FTC: 0 – 20% FBC: 5 – 15% |
| Mata galleria, EESGT, Brazil | Riparian forest lining rivers dominated by palm trees e.g. *Mauritia flexuosa* with an undergrowth of perennial grasses e.g. bahiagrass (*paspalum veredense (PE)*) and *Abolboda* | MAR: 1400–1500mm FTC: 20 – 50% FBC: 20 – 25% |

| | | |
|---|---|---|
| | *poarchon (PE)* on gleysols that remain very humid for most of the year. | |
| Campo humido, Estação Ecológica Serra Geral do Tocantins, Brazil | Seasonally inundated grasslands dominated by perennial grasses e.g. bahiagrass (*paspalum veredense (PE)*) and carpet grass *(axonopus canescens (PE))* with sparse palm trees (*Mauritia flexuosa*) on gleysols that remain humid for most of the year. | MAR: 1400–1500mm<br>FTC: 5 – 10%<br>FBC: 20 – 25% |
| Campo limpo/ sujo, Estação Ecológica Serra Geral do Tocantins,  Brazil | Grassland savannas dominated by perennial tussock grasses e.g.  carpet grass (*Axonopus (PE)*, bluestems (*Schizachyrium (PE)* and Crinkleawn grass (*Trachypogon (PE)*) on sandy soils. | MAR: 1300–1500mm<br>FTC: 0 – 5%<br>FBC:10 – 50% |
| Cerrado ralo/ Cerrado tipico, Estação Ecológica Serra Geral do Tocantins, Brazil | Open woodland savanna dominated by perennial tussock grasses e.g. carpet (*Axonopus (PE)*, bluestems (*Schizachyrium (PE)* and Crinkleawn grass (*Trachypogon (PE)*) with sparse overgrowth of *pigeonwood (Hirtella ciliate (SD))*, *earringwood (Rourea induta (SD))* trees on deep sandy soils. | MAR: 1300–1500mm<br>FTC: 0 – 10%<br>FBC:10 – 60% |
| Kalahari open woodland, NW Ngamiland, Botswana | Open woodland savanna dominated by tussock perennial grasses e.g. digitgrass (*Digitaria eriantha (PE)*) and sickle grass (*Pogonarthria squarrosa (PE)*) with scattered (5–15m) African teak (*Baikiaea plurijuga (D)*) and silver cluster-leaf (*Terminalia sericea (D)*) trees on sandy hills. | MAR: 650 – 750mm<br>FTC: 0 – 5%<br>FBC: 20 – 35% |
| Kalahari grassland, NW Ngamiland, Botswana | Open grassland savanna dominated by tussock perennial e.g. *Stipagrostis uniplumis (PE)* and *Eragrostis rigidior (PE)* on clay soils. | MAR: 700 – 750mm<br>FTC: 0 – 2%<br>FBC: 25 – 30% |
| Great sandy desert, Ngurrara country, Western Australia | Grasslands dominated by spinifex hummocks (*Triodia (PE)*) interspersed with open (5–10m) semi-evergreen *Eucalypt (SE)* woodlands and *Acacia (D)* shrubs on lateritic swales and red sand dunes. | MAR: 400 – 450mm<br>FTC: 0 – 1%<br>FBC: 65 – 90% |
| Tanami desert, Warlpiri country, Northern Territory, Australia | Hummock-grass (*Triodia spinifex (PE)*) dominated grasslands interspersed with open (5–10m) semi-evergreen *Eucalypt (SE)* woodlands and *Acacia (D)* shrubs on sand plains. | MAR: 500 – 600mm<br>FTC: 1 – 3%<br>FBC: 50 – 85% |

[1] life cycle of the dominant grass species; **PE**: perennial > 2 years; **AN**: Annual grasses
[2] Deciduousness of the dominant trees; **D**: Deciduous, **SD**: Semi-deciduous, **EG**: Evergreen

**Table A2. Spearman correlation matrix for the field measurements and the globally available satellite products. Positive correlations are presented in blue while negative correlations are presented in red.**

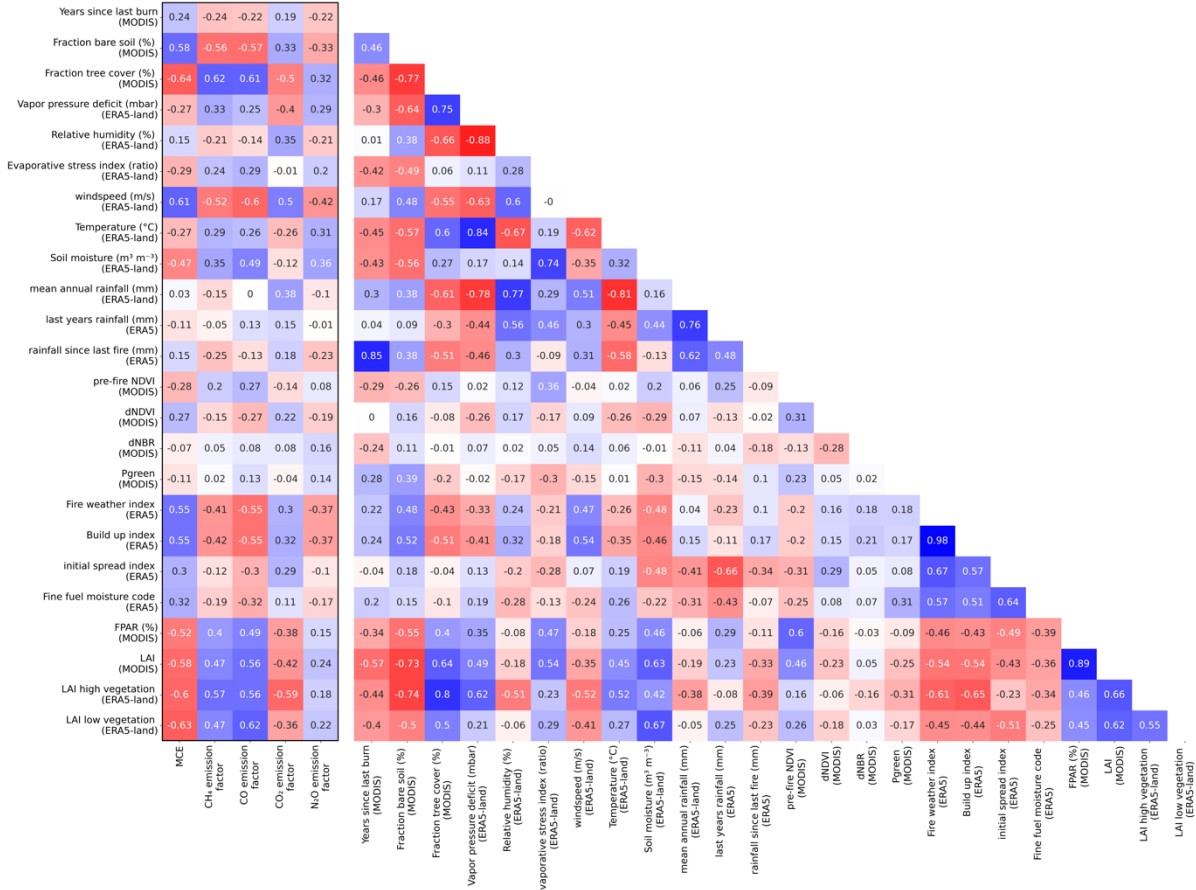

