# Peer review of "Dynamic savanna burning emission factors based on satellite data using a machine learning approach"

_EGUsphere, 2023_

## Referee Comment (RC1)

Dynamic savanna fire EFs by Vernooij et al

Review by Bob Yokelson

As described in a separate group of papers, this team has recently completed a very seasonally and geographically extensive series of measurements of savanna-fire, fire-average MCE and emission factors (EFs) for CO2, CO, CH4, and N2O; along with appropriate related factors like weather, fuel moisture, etc. In this paper, the authors focus mainly on:

1) How well their field-measured environmental factors can be used to calculate their field-measured EFs.

2) How widely-available remote sensing products (aka "features") correlate to their field-measured environmental factors.

3) How well the remote sensing products can be used to predict the field-measured EFs.

4) Using the remote-sensing proxies to calculate a new global emissions database that is sensitive to local spatial and temporal variability in the environment.

Not surprisingly, given the coefficient of variation (COV) (aka relative standard deviation (RSD)) of the EFs in both the literature and the author's work, local variability in emissions was found to be high and the authors work provides a way forward for those wanting to account for variability at a finer scale than is currently common. Also interesting, the +/- variability tends to cancel in their total global emissions, which is encouraging for researchers focused on larger-scale averages. This is important, creative, work that should be published. I have some minor suggestions that I think could strengthen the paper as summarized next.

1/ A few more sentences describing the sampled fires and data reduction would be helpful. I glanced at the previous publications and did not quickly find all the common or potentially useful details. For instance:

a/ Were the fires all prescribed?

b/ How big were they?

c/ Were they detected from space as hot-spots or burned areas?

d/ Were they all lit the same way? (In Brazil we noted that fires were often lit on opposing sides and the flame-fronts burned together. Fires were sometimes lit at night after wind died down.)

e/ What, in a nutshell, was the sampling strategy?

f/ Were RSC samples collected when relevant?

g/ How were the data processed into emission ratios (ERs) and EFs? To clarify last question, Yokelson et al., 1999 compared the impact of processing grab samples into ERs and EFs with several different justifiable approaches. Without proving one approach was best, they found only small differences among approaches. Similarly, regarding the authors work, I don't plan to

critique their approach, but it's useful for posterity to specify the approach used (see below on RSC for more).

2/ The paper would be easier to comprehend the first time thru with slightly more plain language and consistent terminology in describing the statistical analysis.

3/ The discussion on possible future applications is nice. Perhaps one other addition would be to identify which environmental variables might be available in timely enough fashion and have enough predictive power to improve air quality forecasts. I.e. could current or forecast temperatures from the global weather services help predict how fires will burn in near real time?

I also have a few other more focused general comments.

**Overview of value:**

Somewhat related to #3 above, can the computational burden be specified of using the author's full-scale approach or partial implementation? How much easier and how relatively accurate is simply using EDS and LDS EFs?

What is the error in the satellite proxies and how does propagated error in the dynamic EF compare to the impact of switching to dynamic EF?

During a recent field campaign, we found that one of the global vegetation products mapped a pine forest and an alpine wilderness area to savanna and agriculture respectively. Simple added info would be useful such as: do all the author's savanna fires show up as being in a savanna in the remote-sensing products? There is also considerable difficulty/uncertainty in field-measured fuel consumption, etc. Easier than adding many columns for uncertainties would be at least generic uncertainties in the table explaining the data set. The error bars in the figures do look generous to the author's credit. Again, it might be worth stating how the local variability compares to full, propagated uncertainty?

CH4 is exceptionally dependent on MCE, but not all important emissions are as seen in Yokelson et al. (2003) and other work including Andreae 2019.

Yokelson, R.J., I.T. Bertschi, T.J. Christian, P.V. Hobbs, D.E. Ward, and W.M. Hao, Trace gas measurements in nascent, aged, and cloud processed smoke from African savanna fires by airborne Fourier transform infrared spectroscopy (AFTIR), J. Geophys. Res., 108, 8478, doi:10.1029/2002JD002322, 2003.

**Questions on data:**

It seems the Excel spreadsheet is giving time as local time?

The spreadsheet seems not to include background samples. ERs and background values can be derived from slopes and intercepts, respectively. By subtraction of the "_em" column from the "_abs" column, it appears there was a fixed background for each fire. These backgrounds are interesting in themselves. For instance, one fire had a background of 0.17 ppm CO, which is pretty low compared to the 1-5 ppm CO background that can occur during regional smoke episodes during peak fire season. As we also see by FTIR (but don't report), there were negative

N2O emissions and EF at times. How were these negative emissions handled in further data processing?

There are a number of non-physical values in the spreadsheet easily found by plotting the columns in a line chart. E.g., rows 2209-2211, 2353, and especially 2382 and 3116. These data were presumably not used in the training or validation and might be removed?

The letter and number convention for the sample names, does it have any significance that should be explained?

Why are EF calculated for the cals?

Why is no date/time given for the cals?

Why are the cals not all the same or nearly the same? Were there different calibration mixtures or does the scatter reflect the precision?

I was surprised that field-measured temperature had poor correlation with the satellite temperature in Table 4. Then I noticed in the spreadsheet that the temperatures measured on the drone correlate with CO2. In general, the temperature, RH, and VPD seem to be measured in the convection column at times where they would reflect the heat and water production of the fire, rather than an ambient air value that would influence fire behavior. If this is the case, I suggest replacing sample-specific values from the drone with one best ambient value per fire and (if not already done) seeing how that correlates with measured EF and remote-sensing products. Or did the authors use pre-fire met data measured differently or on the drone during the pre-fire cal and that data is available somewhere else?

For example. Picking one fire randomly, EDS19_3 on a June afternoon in Mozambique, one notices that Tsat is close to the climatological average high for June in Maputo (26 C), but is well below the lowest Tdrone (33.57 C). Is that a shade versus sun-exposed thing? Was there a Tdrone during a cal or background that is more appropriate? Further, VPDsat is only close to VPDdrone at minimum Tdrone suggesting combustion products make VPDdrone not representative of ambient VPD unless a VPDdrone measured in background air was actually used? Likewise the RH comparison reveals differences.

In the LGR N2O-CO instrument, the N2O data needs to be corrected for CO and the correction only works up until 5ppm CO. This is because at high CO values, the CO line broadens enough to interfere with the N2O line. In general, the strongest N2O band is overlapped by water, CO2, and CO (and other gases). The CO values in the author's spreadsheet are in the 100s. The manufacturer of the author's N2O instrument (AERIS) product literature claims to use an interference-free, but unspecified, alternative spectral region and have an upper limit of 500 ppm for some unspecified molecule (probably CO?). Kudos to the authors for not using LGR for N2O, but I am curious if the authors have any evidence against or for CO interference in their N2O data? I am not assuming issues exist, but if they can be ruled out, it would be worth mentioning as N2O is an important, but undersampled fire emission.

**Variability of emissions during fire:**

Emissions can vary significantly fire to fire, but also during a fire and there are both lofted (at a range of velocities) and unlofted emissions. Atmospheric chemists tend to focus on chemical completeness and pack a plane full of many instruments and target vigorously lofted emissions at the peak of fire season. This approach is not sensitive to the un-lofted emissions from residual smoldering (aka "post frontal") combustion (RSC, Bertschi et al., 2003). In the author's approach, they were ground-based, but had a drone that could presumably sample both unlofted and lofted emissions, though not the vertical velocity of the emissions. They also measured fuel consumption of the fine fuels (whose emissions are mostly lofted) and the heavy fuels (more prone to RSC). This is a very powerful sampling strategy (for a limited selection of gases) to be commended. The challenges for representative sampling of fires are outlined elsewhere (Bertschi et al., 2003; Akagi et al., 2013) and especially with the timeline given for access by different sampling platforms to the same fire in Akagi et al. (2014). Thus, it would be helpful if this paper gave a brief narrative of when and where samples were collected with respect to the course of the fire and how the data were processed to get fire-averaged EFs. I did not find this info in the previous papers so just a summary here would be great.

Bertschi, I.T., R.J. Yokelson, J. G. Goode, D.E. Ward, R.E. Babbitt, R. A. Susott, and W.M. Hao, Trace gas and particle emissions from fires in large diameter and belowground biomass fuels, J. Geophys. Res., 108, 8472, doi:10.1029/2002JD002100, 2003.

Akagi, S. K., Yokelson, R. J., Burling, I. R., Meinardi, S., Simpson, I., Blake, D. R., McMeeking, G. R., Sullivan, A., Lee, T., Kreidenweis, S., Urbanski, S., Reardon, J., Griffith, D. W. T., Johnson, T. J., and Weise, D. R.: Measurements of reactive trace gases and variable O3 formation rates in some South Carolina biomass burning plumes, Atmos. Chem. Phys., 13, 1141-1165, doi:10.5194/acp-13-1141-2013, 2013.

Akagi, S. K., Burling, I. R., Mendoza, A., Johnson, T. J., Cameron, M., Griffith, D. W. T., Paton-Walsh, C., Weise, D. R., Reardon, J., and Yokelson, R. J.: Field measurements of trace gases emitted by prescribed fires in southeastern US pine forests using an open-path FTIR system, Atmos. Chem. Phys., 14, 199-215, doi:10.5194/acp-14-199-2014, 2014.

**Line by line**

I expand on, or overlap with, these overview comments in my detailed line-by-line remarks. The format is: page number/line number(s)

1/21 change to: "… the breakup of the constituents of the fuel …"? Elements cannot be broken down via chemistry.

1/22 " … other things …"

1/27 "collected" > "made"

1/28 delete "EF", add "EFs" after "N2O". Also "also" before "measured."

1/31 awesome data set! "85 savanna fires", delete "known" change "listed" to "provided"

2/1 Not 100% sure what is meant here. It almost reads like the biome average EF is 60-85% off on average. I think you mean e.g. if a measured fire had an EF 10% below the biome average EF, the satellite-based recalculation of the EF would be ~6-8.5% below the biome average?

2/2 "total global savanna fire emissions estimates"

2/3 change in CO2 totals? (expect small)

2/2-3 It's amazing that the global totals based on average biome EFs were within 1.8 to 18% of global totals using dynamic EFs. The difference is much smaller than the uncertainty in almost any other thing. However, it should be clear what biome average EFs are employed here. Probably the old literature average? Also, is good agreement seen every year or just for the 14 year total?

Ultimately, the paper could compare the old literature average EFs to the evolved literature average EFs that include the author's new data, and the average EFs based on just the authors new work. I.e. how much impact does this study have on averages? Finally, in addition to predicting measured EF better, it would be interesting to know if the use of dynamic EFs also better predicts downwind impacts, but that might be another paper.

2/5-6 Did not the authors observe that CO and CH4 EFs decreased with drying in xeric grasslands, but increased with drying in mesic woody savannas? Also "… annual average savanna fire …"

2/7 Are there just reductions? There is good agreement on totals so there should also be localized increases. In general, from the 1-sigma standard deviation in literature EFs we expect +/- 40% variation in EFs fire-to fire 1-sigma.

2/12 Throughout paper it should be "emissions inventories"

2/15 60% of net emissions? Could deforestation and peat be more important in the C-cycle if minimal regrowth?

2/26-28 There are many direct field measurements and they quantify overall variability, but previously we could not account for the total variability with quantitative contributions from very many specific factors. Previous studies targeted the average and variability, but not the causes of variability.

2/30 change "closed" to "open"? At least in Akagi et al there was a split between dry forest and woody savanna at 60% canopy coverage.

3/4-5 How about ", a series of savanna burning experiments measuring EFs using unmanned aerial systems (UAS) has resulted in a large amount of new data …"

3/6-7 How about "… the variability in over 4500 individual bag-measured EFs of CO2, CO, CH4 and N2O covering 129 fires."

3/25-26 Could other real-time data besides that from satellites be useful?

3/31 "savanna fires" and no comma after "fires". It wasn't immediately obvious that the cited references contained much detail about the fires, sampling strategy, and data reduction. Maybe I missed it in a supplement?

3/32 "… at altitudes between …"

3/36 Maybe I missed it somewhere, but worth repeating here as no page limit. Include averaging/plotting schemes, RSC sampling? Fire descriptions? How met data collected, etc. As detailed in overview

4/10 "sometimes prevented"?

4/15 The fire dates and coordinates are always "known" but not always "provided" in the paper, though perhaps available from authors. This illustrates value of describing fires in some detail.

4/21 collected > measured

4/23 The pre-fire met data mentioned here, where is it? The spreadsheet has non-useful met data collected in the fire convection column.

4/30 Okay this provides some of details requested above.

4/35 One naturally wonders here if the authors field environmental data can be used for insight into the accuracy of the global satellite products and were their fires detected by the satellite products GFED4s uses?

4/36 built

5/7 Impressive set of products. Is it easy to explain why no VIIRS or geostationary? Not available as long? Useful going forward?

5/8 Were all the samples of a fire usually in the same feature pixel?

5,/15 Is it easy to explain why not using historic NDVI range?

5, 25-26, TRMM useful for rainfall?

5/30-31 risk or behavior or both? Are any ideas in the "hot dry windy index" useful as predictors here?

5/34-35 Is the daily cycle of fine fuel moisture captured? Was FFMC compared to the author's field-measured fine-fuel moisture data?

6/6 How does spatial resolution of the fire severity proxies (dNDVI etc.) compare to the size of fires? If the fire is smaller, then is the signal diluted? Would a small severe fire look like a larger less severe fire? Did the authors expect better correlation of scorch and char height with the severity proxies?

6/17 modes or models?

6/18 all in-situ or 70% as on line 20?

6/19-20 What is "a measurement with a missing value of an included feature"? Do you mean you did not use EF measurements if even a single associated satellite product out of the whole set was missing?

6/21-22 What does "resampled using ten-fold cross validation while allowing sample replacement (i.e., bootstrap method)" mean? Can a simple plain language explanation be added?

6/22-23 Explain that "hyper parameter" refers to the most influential parameters?

6/28-30 This is hard to follow. How would an EF require a resolution and how would that be computed? Overlap is within or between features? Do you mean some fires were bigger than or occupied more than one grid cell in the original feature (note we have slipped into calling remote-sensing proxies "features" for short), so you averaged, or extrapolated, or built a new grid for each fire such that the fire was centered in a single grid cell? Sometimes a few extra words can help a lot!

6/32 How can an EF have a temporal resolution? Are the EFs referred to fire-average or sample-specific? Is the daily cycle in RH and fine fuel moisture considered?

6/40 "… savanna fire emissions …" Were the dynamic EFs calculated using global products and RF? Change "dry matter emissions" to "dry matter consumption" also at 7/8.

7/15-16 How were samples with negative N2O emissions treated when calculating fire-average N2O emissions?

7/19-20 Clarify this is the Andreae 2019 average and not the average of the 85 measurements used from other groups? Otherwise, how do you get locations for study-average or vegetation average emissions (unless one fire in study)?

7/23 substantial variability in fire-averages or samples?

7/24-25 The higher CO and CH4 EFs in woody savanna is supported in previous literature at least once, e.g. Sinha et al., (2004).

Sinha, P., P.V. Hobbs, R.J. Yokelson, D.R. Blake, S. Gao, and T.W. Kirchstetter, Emissions from miombo woodland and dambo grassland savanna fires, J. Geophys. Res., 109, D11305, doi:10.1029/2004JD004521, 2004.

FWIW, the Miombo fire was included in the tropical dry forest category in Akagi et al, but it was also a small part of a savanna fire study-average used in the savanna category.

7/25 Taking this to mean the authors study-averages were lower than previous literature averages.

7/29 by "seasonally inundated grasslands" do you mean aka dambos?

8/2-3 Any benefit to comparing the authors fuel measurements to similar measurements by Shea et al (1996) and Hoffa et al (1999) and others?

Shea, R. W., Shea, B. W., Kauffman, J. B., Ward, D. E., Haskins, C. I., and Scholes, M. C.: Fuel biomass and combustion factors associated with fires in savanna ecosystems of South Africa and Zambia, J. Geophys Res., 101(D19), 23551–23568, 1996.

8/4 What is meant by "corresponding mixtures of fuel age"? In Table 3, why was a higher percent of the heavy fuels consumed in the EDS in Australia, unlike elsewhere; maybe lit more aggressively?

8/10-13 I'm pretty sure that increased RSC and increased CO and CH4 EFs in the LDS in wooded savannas is already in the literature, but haven't found the reference. Maybe Hoffa or Korontzi?

8/16-28 good summary.

8/32 For Table 4, clarify which field-measured met data were compared to satellite met data, preferably NOT drone data in fire-processed air! However, Table 4 seems to specify that T and RH from the drone were used, which could be okay if NOT when drone was above the fire, but instead in ambient (background) air. Then again, currently, it's odd that the satellite temperature and drone temperature are weakly positively correlated at 0.18 while satellite temperature is most strongly correlated with field measured nitrogen content in the grass (perhaps a seasonal coincidence?).

8/31-33 This text and Table 4 could be clarified with slightly more precise and consistent terminology. I think Tab 4 shows how the *field measured-ecosystem attributes* correlate with the field-measured MCE and EFs and also how the *field-measured ecosystem attributes* correlate with the satellite products, but NOT how satellite products correlate with field-measured EF or how anything correlates with model-calculated EF? At this point in the paper, evidently, calculated EFs vs measured EFs and the sensitivity of calculated EFs will be discussed elsewhere.

8/40 strongest > most strongly

9/4-5 I'm taking this to mean that 70% of field-measured EF were used with "features" to train the RF model and the RF model then used features to predict EF for the other 30% of field measured EF (out of sample means fires not in training set) and performed well in terms of r-squared. Could give the slope too? Is the training set randomly selected or varied run to run?

9/5-7 Is there a simple way to connect feature importance and the concept of hyper parameters? Is "impurity decrease" essentially a fraction of total variability?

9/8 The red line in Fig 4 is useful for comparing the range of EF to the old literature average. But later in paper, the effect of dynamic EF should perhaps be compared to the biome average based just on the field data used by the authors, which could be shown with a second vertical line. Then recalculate MAE and improvement %. Currently, the comparison is "apples to oranges" in that "improvement" is based on a difference resulting partly from incorporating new data and partly from a change in approach.

9/9 replace "data" with "features"

9/10 change "recalculate the MCE" to "predict the field-measured MCEs of the fires in the validation set"

9/11 "static" > "old literature"

9/13-20 This is a nice exploration of simplifying the RF approach. Can the authors explain why VPD is the most important feature in the small subset of features, despite having a low rank in the full set of features? Any estimate of reduced computational burden?

9/24-25. Does this mean you ran the RF model once to get MCE and then used the MCE as a new feature in a re-run of the RF model?

9/28 130 or 129? Including fires with negative N2O emissions?

9/35-38 It would be interesting to see the study-average EFs vs the former literature average EFs and then also what the new literature averages are including this study, all in a little 3x4 table.

10/2 How common are mixed biome grid cells? Percentage of total? Is the most common type of mix with tropical dry forest? Is there a percent tree cover or canopy closure that defines the boundary between what the authors consider savanna and something else?

10/11-12 What is "annual effective EF"? An annual global savanna-fire average EF for each compound? This is also saying the year to year variation in global average EFs is small?

10/14-15 averaged over what time and space? I.e. the daily average over all areas occupied by the indicated vegetation class? Fig 7 doesn't seem to show much or any EFCO increase in woody savanna as the fire season progresses? Does this figure clash with previous text? What is "typical savanna"?

10/30-34 Interesting, shows the RF model may have value to at least partially correct sampling bias in a field campaign!

11/2 Just to be clear, the N is in the foliage of the trees, not the wood itself

11/15 "CO-"?

11/24-26 Did Hoffa and Korontzi predict higher MCE in LDS?

11/30 The Eck trend in SSA is averaging over all sub-Saharan Africa AERONET sites?

11/32 References that support an increase in SSA as MCE decreases include Liu et al and Pokhrel et al and probably many others

Liu, S., Aiken, A. C., Arata, C., Dubey, M. K., Stockwell, C. E., Yokelson, R. J., Stone, E. A., Jayarathne, T., Robinson, A. L., DeMott, P. J., and Kreidenweis, S. M.: Aerosol single scattering albedo dependence on biomass combustion efficiency: Laboratory and field studies, Geophys. Res. Lett., 41, 742–748, doi:10.1002/2013GL058392, 2014.

Pokhrel, R. P., Wagner, N. L., Langridge, J. M., Lack, D. A., Jayarathne, T., Stone, E. A., Stockwell, C. E., Yokelson, R. J., and Murphy, S. M.: Parameterization of single-scattering

albedo (SSA) and absorption Ångström exponent (AAE) with EC / OC for aerosol emissions from biomass burning, Atmos. Chem. Phys., 16, 9549-9561, doi:10.5194/acp-16-9549-2016, 2016.11/37 linear or non-linear?

11/40-12/1 Not sure about the interpretation here. Does CH4/CO vary with MCE? CO is not technically independent of MCE since MCE has CO in its definition.

12/2-3 interesting

12/5-9 This discussion could be misleading in a subtle way. I think the effect seen here is probably because the other studies compared to are plotting the fire-average EFCH4 versus the fire-average MCE, while the authors are plotting EFCH4 vs MCE for "snapshot grab samples" that could include samples during flaming that may have much higher MCE than the fire-average MCE for typical useful real-world fires. We've seen this often over the years. To illustrate we can revisit the comparison to the Selimovic et al lab fire study. If one plots the instantaneous EFCH4 vs instantaneous MCE for these typical lab fires you often get "curvature" at high MCE values during "pure flaming" and other effects. The ERCH4 vs MCE can also be non-linear at high MCE or have interesting other interesting patterns with time. The plots show this for the 1-s data from randomly selected Fire #74 on the NOAA FIREX-Firelab archive (https://esrl.noaa.gov/csd/groups/csd7/measurements/2016firex/FireLab/DataDownload/). Fire #74 is one of the fires in the linear plot of fire-integrated EFCH4 vs MCE in Selimovic et al. (2018). Interesting topic but variability during a fire is a level of detail large-scale models can't cope with yet. Thus, in providing guidance for large-scale models it may be best to stick to fire-average data.

[Figure]

12/11-22 Since Brazil has a lower elevation than Zambia it may have higher temperatures and more evaporation than Zambia making it more xeric at the same total rainfall? In addition, African sites may have keystone grazing species that encourage forest cover by reducing grass fuels and thus fire intensity and tree mortality?

12/25-26 Think you mean "This is the first study to quantify the spatial distribution of GHG EFs over the entire savanna biome by using both field measurements from a variety of savanna ecosystems and their relation to global data mainly from satellites". I.e. the field measurements

have gaps as explained in the following lines, but by connecting the measurements to features you have a new way to get a useful global savanna estimate!

13/11 The idea of a gross underestimate here is worrisome. How well do the authors think GFED4s accounts for fires too small to show up in their burned area product? Worth mentioning here?

14/7 change "propose" to "developed"?

14/9 "modelled"> "model-produced". Also "significant fire-specific improvements"

14/10 Here I think it's important to preserve the idea that you have not concluded the biome averages have large errors, just that fire to fire variability is large and is better accounted for by using a more sophisticated model. Also + and – local errors tend to cancel. It worries me that someone reading quickly may think you mean that global CO and CO2 emissions from savanna fires are off by ~80%.

14/21 Remind reader that the N2O decrease is a combined effect of new data and the dynamic approach.

14/31 I did not check the zenodo link. If it is different from spreadsheet, I could check it by request.

Fig 1. Andreae 2019 is not indicated in yellow as caption suggests?

Fig 7. Why do "typical savanna" fire emissions peak earlier than all the subtypes?

---

## Author Comment (AC1)

Response of the authors to comments by Paul Laris on the manuscript: "Dynamic savanna burning emission factors based on satellite data using a machine learning approach"

Roland Vernooij (corresponding author) on behalf of the authors:

We sincerely thank Paul Laris for taking the time and effort to read and comment on our manuscript, and the detailed and constructive comments both on this platform and in earlier conversations, which helped to improve the quality of this paper. Please find below our point-to-point response to the review. The revised text and updated figures are included in the updated manuscript. A separate 'track-changes' document is included to highlight the changes we made to the manuscript.

| Detailed comments | Author's response, reasoning and comments |
|---|---|
| Can you clarify that these fires were all **"head" fires** as opposed to backfires? And, if so, can you comment on why the following dimensions are adequate? We question wether 10m is wide enough for head fires to fully develop. This width is fine for backfires. Also, if only head fires were examined, can you justify given that many fires are purposefully set as backfires in Africa. Headfires have long been used in research on African fires, yet research finds more backfires are set. | While it is indeed correct that most of the measurements have been taken during 'headfires' and 'sideward propagating' fires we also measured backfires. We tried to obtain measurements proportionately to the area burned by the different types within these fires. However, we did also conduct measurements where we tried to distinguish the different fire propagation directions, which in a changing wind regime was much more challenging than we previously anticipated. Individual 35-second bag samples more often than not contained smoke from multiple changing wind directions. In agreement with the findings by Wooster et al. (2011) and Laris et al. (2021) we found "back" fire samples to have slightly higher combustion completeness compared to "head" fire samples. A possible explanation being that slower lofting smoke from the residual smouldering does not mix with the flaming emissions in these measurements, like it does in head fires. There was no significant difference between head and sideward propagating fires.

The early dry season fires were all lit by land managers under the guidance of prescribed burning experts. This meant that head fires were only used if the conditions allowed them (which was most often the case), to prevent runaway fires. Although these experts deemed the measured EDS fires representative of prescribed fires, you |

are correct that pure backfires may burn more efficiently.

To clarify this in the text, we have added the following in (P3 L32): "Fires were lit with the aim of being representative of early dry season (EDS, often prescribed) fires and late dry season (LDS) non-prescribed fires. Although some backfires were sampled during the initial phase of the fires, the majority of samples were obtained from the faster 'head' fires, which consumed most of the biomass. Fire sizes generally ranged between 2 to 10 hectares based on UAS drone imagery described by Eames et al. (2021), with exceptions of some fires that would not light and conversely, some fires that burned several hundred hectares. In the EDS, fire size was primarily limited by environmental conditions and fires ceased burning as humidity increased overnight whereas in the LDS, fire size was confined by low-fuel areas like burn scars, roads and prepared fire breaks. Particularly in the LDS, this means a limited fire size does not necessarily indicate limited fire intensity. Emissions were sampled at altitudes between 5–50 m depending on flame height for a duration of 35 seconds, resulting in 0.7 litres per gas sample. On average, we took 35 samples per fire. The sampling methodology involved taking samples from a fire passing a certain point –while correcting for wind direction and severity– until no more visual smoke passed the drone anymore. From earlier work (Vernooij et al., 2022a), where we compared the average of these measurements to results using continuous measurements taken at a mast, we have some confidence in the fidelity of this approach."

Also, in the discussion (P14 L19) we added: "The samples were predominantly collected over "head" fires, which in the measured fires typically represented most of the burned area. A common approach for prescribed fires is burning against the wind

| | (backburning), to minimise both the impact on vegetation and risk of spread. In accordance with Wooster et al. (2011) and Laris et al. (2021), we found higher MCE in samples from backfires, which indicates these types of fires may emit less $CH_4$ and CO. Another possible explanation is that slower lofting RSC smoke does not mix with the flaming combustion emissions in these measurements, like it does in "head" fires." |
|---|---|
| You do not appear to have published local or ground data on weather conditions. While T and H can be collected from regional weather stations, **wind speed is critical to determining fire intensity and will influence MCE as well.** Do you have wind data, it would seem critical for accurate fire intensity and MCE results. | Unfortunately, we have only started to log windspeed (from a Kestrel 5500FW Fire Weather Meter) in the very last campaigns. We agree that windspeed is most likely a more significant predictor that the models suggest based on the ERA5-land data. Note that although WS is not often seen as a major predictor, the FWI which contains WS is. While it would be very interesting to verify the windspeed from ERA5-land with the on-site windspeed, more accurate on-site windspeed measurements could not be used for the spatiotemporal extrapolation, and therefore would not improve the model. |
| I wonder about this comment: "

The grasslands with the highest EFs (found in high-rainfall savanna Dambos) were **"uncharacteristically green for the time of the season"** given that many fires are set to "green" grasses in African savannas, especially the perennials (See Le Page who documented this back in 2010 as well as many West African case studies). | The vegetation we refer to with this comment was highly limited in its spatial extent to relatively deep and clayey Dambos with widths often smaller than a 500-meter MODIS pixel. Since the water availability and grass curing state in these areas is highly dependent on soil type and geomorphology, these characteristics are poorly captured by the much coarser seasonal features (e.g. soil moisture and VPD). The Dambos where we measured the highest EFs (also in the LDS) had just fallen dry and were still very green, whereas other Dambos close by had already fully cured and showed very low EFs. By this statement we mean that because of the dominant role of soil type and geomorphology, the EFs measured in those Dambos were a poor indicator of the seasonal cycle in other grasslands. |

| | |
|---|---|
| | We added the following text to the discussion (P14 L36): "Although burning grasslands under green conditions releases more $CH_4$ and CO, there are valid reasons to do so. For example to remove moribund grass that remains after the dry season with minimal damage to the grass sward (Nieman et al., 2021; Le Page et al., 2010). In its current form, the model may not always pick up on those landscape features." |
| I think Laris et al found very similar results to: "The strongest predictors for the MCE and the CO and CH4 EF were the **tree density in the plots, the grass to litter ratio, the combustion completeness** and the **moisture** content of the consumed fuel. It might be useful to compare and to consider the hypothesis that burning of green leaves on shrubs and trees vs. dried leaves on the ground may explain why EF CH4 is not linerally related to MCE. This reasoning may also explain the following finding, "For CO and CH4, the dominant effect is a spatial redistribution with higher CO and CH4 EFs in mesic, high-tree cover savannas and lower EFs in xeric savannas compared to previous estimates. The Higher CH4 EF in mesic may well be a function of leaf burning. This is logical given the findings from Senegal research by Barker finding burning trees emitted smoke with the highest methane EF.

This needs further explanation: "Although CO and CH4 followed the same spatial pattern, we found that MCE affected the CH4 which resulted in lower CH4 to MCE ratios in open **(lower tree density?) savannas…. Do you mean higher CH4/MCE in wooded savannas as compared to grass-dominated ones? What is "open"? Clarify.** Again, see works in Mali and Senegal which agree with this finding. | We indeed find higher $CH_4$/MCE ratios in tree-dominated savannas compared to grass-dominated fires.

In our previous work on isotopes, we found $CH_4$ EFs to be more $^{13}C$ depleted compared to CO emissions when burning wooden logs. This may indicate $CH_4$ is more RSC driven than CO and possibly stronger dominated by the pyrolysis of lignin rather than cellulose and hemicellulose.

In P12 L37 we added the following text: "In accordance with previous studies (e.g. Korontzi et al., 2003b; van Leeuwen and van der Werf, 2011; Barker et al., 2020), we found steeper $CH_4$ EF to MCE regression slopes in woodlands compared to grasslands. Our data indicated a positive correlation of the $CH_4$ EF to MCE slope with the FTC based on MOD44Bv006. The MCE is a simplified form of the combustion efficiency and only calculated using CO and $CO_2$ emissions. Being less oxidized than CO (which is still common in flaming combustion), $CH_4$ emissions have a stronger dependency on the actual combustion efficiency ($CO_2$ divided by all carbon emissions). While most studies describe the relationship between the $CH_4$ EF and the MCE as being linear (Korontzi et al., 2003; van Leeuwen and van der Werf, 2011; Selimovic et al., 2018; Yokelson et al., 2003), we found that for individual bag samples it was better described using a nonlinear function (Fig. 9), in line with findings by Meyer et al. (2012) for Australian savanna measurements. Figure 9 represents |

| | individual bag measurements rather than fire averages (for which the spread in MCE is much lower). Laboratory experiments described by Selimovic et al. (2018) showed that the $CH_4$ to CO ratio is strongly dependent on flaming or smouldering phases if the fire. Individual bag samples –which often hold emission from a single phase– therefore show much more variation compared to fire averages. Stable carbon isotopes also point to $CH_4$ emissions being more depleted in heavy carbon ($^{13}C$) compared to CO in both mixed (C3 and C4) and single-fuel-type experiments, indicating a stronger dominance of RSC and the pyrolysis of lignin in its total emissions (Vernooij et al. 2022b). This explains both why studies that are skewed towards either smouldering or flaming phase emissions find different $CH_4$ EF to MCE slopes using linear regressions and why this slope varies with FTC.”

 With “Open savannas” we indeed meant lower tree density. To avoid this confusion, we changed the text to: ‘savannas with lower tree density’ |
|---|---|
| **Not sure I agree with this logic:** "Contrary to previous research which indicated that dryer conditions in the LDS would lead to higher-MCE fires late-LDS conditions (Fig. 3). In part, this may be because our measurement campaigns missed the peak-season fires when the fires may be hotter..."  Winds are the critical factor here.  When do they peak in areas studied.  High winds (especially if fires studied are head fires) result in higher intensity regardless of fuel moisture. Laris also found lower MCE in LDS due to leaf litter in the fuel load and lighter winds with much higher winds in MDS for the region studied. Note that these factors are key reasons why binary (LDS/EDS) is problematic for determining emissions. | While we did not include windspeed in the field measurements and therefore in the intermediate explanatory field drivers. However, we agree that it is a very influential driver of fire behavior. In the future we will include windspeed measurements on the ground. Although this means we currently cannot correlate reliable measurements of the actual windspeed during the fire with satellite derived proxies, we do include windspeed proxies in the model.

 We added the following text (P12 L24): “Contrary to previous research which indicated that dryer conditions in the LDS would lead to higher-MCE fires in both grasslands and savanna woodlands (Korontzi, 2005), we found lower MCE in these regions under late-LDS conditions (Fig. 3). This may be because our |

| | measurement campaigns missed the peak-season fires when the fires may be hotter due to stronger winds (Laris et al., 2021; N'Dri et al., 2018)." |
|---|---|
| | We acknowledge that the binary (LDS/EDS) classification is in many ways flawed, as you rightfully point out in your earlier work. With this study, we hope to work towards getting rid of the EDS and LDS classes altogether when it comes to savanna EFs. |
| | In the introduction (P2 L37) we state: "EFs used for the accreditation of such projects currently assume a dichotomy of early- and late dry season averages, determined by a cut-off date. However, as discussed by Laris (2021), the fuel and meteorological conditions thought to drive EFs vary more gradually over the season and are subjected to substantial inter-annual and spatial variability. Incorporating spatiotemporal variability in inventories makes emission inventories more dynamic and better equipped for assessing seasonal fluctuations." |
| Again, see research in Mali and Senegal which support this finding: In accordance with previous studies (e.g. Korontzi et al., 2003b; van Leeuwen and van der Werf, 2011), we found steeper CH4 EF to MCE regression slopes in woodlands compared to grasslands.

Comments

Figure 3. What is **"typical savanna"** there is no such thing.

Also, use more specific terminology, what is "open"? | These classes serve to indicate that the prevalence of trees was a useful feature for clustering the EFs. In Figure 3, we removed the classes and replaced those with rough FTC bands (0-2%, 2-10% and 10-50%) |
| **This and other data rely on 500 x 500 MODIS is this relevant given efforts to burn patchy EDS fires which operate at a hectare level scale? Can you justify using 500m data for the following?** For fire severity proxies we used the differential | That is indeed an issue and could be one of the main reasons why the models did not pick out any of these features as strong indicators of the fire. Although not mentioned in the list of features, we also used Landsat retrievals for the |

| | |
|---|---|
| Normalized Burn Ratio (dNBR) and 5 the differential Normalized Difference Vegetation Index (dNDVI) retrieved before and after the fire. These were based on the MODIS surface spectral reflectance, corrected for atmospheric conditions (MOD09GAV6; Vermote) | abovementioned spectral indices. While the spatial resolution is better, it goes at the cost longer intervals between cloud-free scenes and just as with MODIS data our model did not find these features were important.

In their current form these models were developed with the application of global modelling in mind. This means that using high resolution (e.g. Landsat and Sentinel) data becomes computationally heavy. Although it could be possible to retrieve the training data at higher resolution and subsequently use courser products (e.g. MODIS) for the spatiotemporal extrapolation, using different data for training and final usage is risky as tree-based models use absolute split values. Therefore, the consistency of these datasets would have to be proven for the entire savanna biome first.

We added the following text in the discussion (P15 L4): "Fire intensity proxies (dNDVI and dNBR from MODIS) were poor predictors for the EFs. A potential explanation is that these features were at times heavily diluted, as many of the measured fires only affected part of the pixel. Similar misrepresentation errors can be expected for the NDVI before the fire, FPAR and the Pgreen. Particularly in the LDS, we were often limited to areas that were enclosed by recent fire scars (0-2 years) or other non-flammable boundaries. Although these areas were sizable (several hectares) many of the retrievals in these pixels may poorly represent the burned vegetation. Along with inconsistent retrievals related to cloud cover, this may be an important reason why these features were deemed poor predictors by the models while seen as strong predictors in previous research (Korontzi et al., 2004). Higher resolution features may increase the representativeness of the pixels for the actual burned vegetation." |

---

## Author Comment (AC2)

**Response of the authors to comments by Bob Yokelson on the manuscript: "Dynamic savanna burning emission factors based on satellite data using a machine learning approach"**

Roland Vernooij (corresponding author) on behalf of the authors:

We sincerely appreciate the considerable time and effort spent in assessing our manuscript, and the detailed and constructive comments which helped to improve the quality of this paper. Please find below our point-to-point response to the review. The revised text and updated figures are included in the updated manuscript. A separate 'track-changes' document is included to highlight the changes to the manuscript. Additional explanatory figures which we refer to in the answers are added to the bottom of this document.

**Reviewer 2, Bob Yokelson**

| General comments | Author's response, reasoning and comments |
|---|---|
| 1/ A few more sentences describing the sampled fires and data reduction would be helpful. I glanced at the previous publications and did not quickly find all the common or potentially useful details.

For instance:
a/ Were the fires all prescribed?
b/ How big were they?
c/ Were they detected from space as hot-spots or burned areas?
d/ Were they all lit the same way? (In Brazil we noted that fires were often lit on opposing sides and the flame-fronts burned together. Fires were sometimes lit at night after wind died down.)
e/ What, in a nutshell, was the sampling strategy?
f/ Were RSC samples collected when relevant? | We added the following lines to the methodology (P3 L32): "Fires were lit with the aim of being representative of EDS (often prescribed) fires and LDS non-prescribed fires. Although some backfires were sampled during the initial phase of the fires, the majority of samples were obtained from the faster 'head' fires, which consumed most of the biomass. Fire sizes generally ranged between 2 to 10 hectares based on UAS drone imagery described by Eames et al. (2021), with exceptions of some fires that would not light and conversely, some fires that burned several hundred hectares. In the EDS, fire size was primarily limited by environmental conditions and fires ceased burning as humidity increased overnight whereas in the LDS, fire size was confined by low-fuel areas like burn scars, roads and prepared fire breaks. Particularly in the LDS, this means a limited fire size does not necessarily indicate limited fire intensity. Emissions were sampled at altitudes between 5–50 m depending on flame height for a duration of 35 seconds, resulting in 0.7 litres per gas sample. On average, we took 35 samples per fire. The sampling methodology involved taking samples from a fire passing a certain point –while correcting for wind direction and severity– until no more visual smoke passed the |

drone anymore. From earlier work (Vernooij et al., 2022a), where we compared the average of these measurements to results using continuous measurements taken at a mast, we have some confidence in the fidelity of this approach. "

Regarding point c, we added the following text to the discussion (P15 L19): "Of the UAS-measured fires in this study only 5 of the 45 EDS fires (11%) and 13 of the 65 LDS fires (20%) were registered by MCD64A1 as burned area (while also accepting adjacent pixels and a 4-day time lag) and only 4 of the 45 EDS fires (9%) and 32 of the 65 LDS fires (49%) were registered by VIIRS S-NPP as thermal anomalies (with the center point of the hotspot (including a 1-day time lag) being within a 3.5 km radius of the sample). Depending on the spatiotemporal nature of these omissions, this may affect some of the results in this study concerning the effects of the EF dynamics on total emissions. Chen et al. (2023) indicate that in the savannas, disproportionately more burned area is added in higher tree-cover areas when using higher resolution satellite imagery. Giving more significance to these areas would mean our savanna-wide effective EFs of CO, $CH_4$ and $N_2O$ would increase, The LandSat and Sentinel based burned area product from (Roteta et al., 2021) performed much better and registered 8 of our 14 EDS fires (57%) and all of our 16 LDS fires (100%) in Botswana and Mozambique in 2019 (while also accepting adjacent pixels and up to a 21-day time lag). Due to the fewer overpasses the temporal allocation of this product is less precise with an average time lag of 5.5 days. Figure 10 shows the portion of our EDS and LDS fires that were detected by various satellite algorithms."

Figure 10 is also included at the bottom of this document.

| | |
|---|---|
| How were the data processed into emission ratios (ERs) and EFs? To clarify last question, Yokelson et al., 1999 compared the impact of processing grab samples into ERs and EFs with several different justifiable approaches. Without proving one approach was best, they found only small differences among approaches. Similarly, regarding the authors work, I don't plan to critique their approach, but it's useful for posterity to specify the approach used (see below on RSC for more). | The excess mixing ratios (EMR, sample minus background concentrations) of the GHG and aerosols were converted to EFs using the carbon mass balance method (Yokelson et al., 1999):

$$EF_i = F_c \times \frac{MW_i}{AM_c} \times \frac{C_i}{C_{total}}$$

where $EF_i$ is the emission factor of species $i$ (usually reported in g kg⁻¹) and $Fc$ is the fractional carbon content of the fuel by weight (estimated at 50% following Akagi et al., 2011). $MW_i$ is the molecular weight of species $i$ which is divided by the atomic mass of carbon, $AM_c$. $C_i$ is the moles of carbon per mole of species $i$ multiplied by the EMR of species $i$. $C_{total}$ is the total number of moles of emitted carbon in all carbonaceous species. Because we did not measure the non-methane hydrocarbons and the chemical composition of carbonaceous particulates, the NMHC and the carbon content of the particulates were estimated based on literature values in order to estimate $C_{total}$; The total amount of carbon in non-methane hydrocarbons was estimated to be 3.5 times the ER($CH_4$/$CO_2$) based on common ratios for savanna fires (Andreae, 2019; Yokelson et al., 2011, 2013). For the bag and mast measurements, we used the PM to CO ratio based on AM520 and CRDS measurements, with carbon accounting for 68% of the PM-mass (Reid et al., 2005a). Overall, the carbon in PM and NMHC constitute respectively 0.5−2% and 0.4−3% of the total emitted carbon. Therefore, the uncertainty from the effect this assumption on the EFs of gaseous species is limited. On average, the PM to CO ratio in our measurements was $0.0946 \pm 0.0218$ which corresponds well with the $0.0969 \pm 0.0403$ average for savanna fires (Andreae, 2019). |
| The paper would be easier to comprehend the first time thru with slightly more plain language and consistent terminology in describing the statistical analysis. | We have added some clarifications to the text (particularly section 2.2.2) where we explain some of the terminology. These specific clarifications will be further discussed in the answers to the detailed comments below. |

| | |
|---|---|
| The discussion on possible future applications is nice. Perhaps one other addition would be to identify which environmental variables might be available in timely enough fashion and have enough predictive power to improve air quality forecasts. I.e. could current or forecast temperatures from the global weather services help predict how fires will burn in near real time? | Supplementing the datasets on which the models are trained with alternative data always comes with uncertainty and consistency should be checked. However, we believe substituting for instance ERA5-land temperature with ERA5 temperature to achieve more NRT, or even T predictions from CMIP projections can be useful.

We added the following statement (P16 L14): "The models are currently trained using meteorological features obtained from ERA5-Land (Muñoz-Sabater et al., 2021) which is available from 1950 to present and has a 2- to 3-month delay. When interested in longer time periods or for near-real-time (NRT) applications these features may be substituted with ERA5 (Hersbach et al., 2020) which is available from 1940 to present with a shorter latency period of 5 days, or even CMIP climate projections. Although supplementing the datasets on which the models are trained with alternative data always comes with additional uncertainty, we found meteorological parameters obtained from ERA5-Land to be in close accordance with ERA5, indicating the two may also be substituted. This means that the EFs computed using the methodology outlined in this paper can also be used to improve NRT biomass burning emission estimates like those from CAMS-GFAS (Andela et al., 2015; Di Giuseppe et al., 2016)." |
| Somewhat related to #3 above, can the computational burden be specified of using the author's full-scale approach or partial implementation? How much easier and how relatively accurate is simply using EDS and LDS EFs? | We found that a binary (EDS vs LDS EF) approach is not justified given the gradual changes over time we observed. To make sure data users are not burdened with an overload of information we will provide NetCDF files with daily savanna EFs for various species as well as MCE which will be part of the Global Fire Emissions Database version 5 we are currently working on. |
| What is the error in the satellite proxies and how does propagated error in the dynamic EF compare to the impact of switching to dynamic EF? | The reviewer brings up a valid point; satellite proxies carry uncertainty and we do not account for this when building our models. We cannot provide a definitive answer but would like to note two things. |

| | First, we warn against substituting data sources to avoid biases in these to start playing an important role. Right now, if there is a bias in a dataset this will not matter. Clearly this does not count for misinterpretation or uncertainty in general. Second, we note that this issue is common for large-scale modelling approaches and to some degree it is difficult to properly account for given that the uncertainty of the large-scale datasets is uncertain. From our perspective, we feel the errors are small enough to be sure about the key findings such as lower EFs in dry regions and higher in wetter regions, but they clearly matter. We have inserted a statement on this (P15 L32): "The meteorological parameters obtained from the ERA5-Land dataset carry uncertainty. This uncertainty becomes higher when going back further in time due to a decrease in validation data. To what extent uncertainty propagates to the EF predictions varies depends mostly on whether there is a bias that was also present in the training data or misinterpretation or uncertainty in general. As this model is trained using specific datasets, these datasets should not be replaced by other sources without evaluating the consistency of that source with the training data. FTC and FBC, based on MOD44Bv006 were found to be strong predictors of BB EFs. However, intercomparison with Tropical Biomes in Transition (TROBIT) field sites in African, Brazilian and Australian savannas has shown that this product consistently underestimates canopy cover in tropical savannas by between 9 to 15% (Adzhar et al., 2021). Products based on higher-resolution satellite retrievals (e.g. LandSat and Sentinel) have the potential to further enhance the spatial resolution of the EF estimates to include small landscape features and thus become more representative. Although all satellite data comes with some uncertainty, we feel the errors are small enough to be sure about the key findings such as lower EFs in dry regions and higher in wetter regions, but they clearly matter." |
|---|---|

| | |
|---|---|
| During a recent field campaign, we found that one of the global vegetation products mapped a pine forest and an alpine wilderness area to savanna and agriculture respectively. Simple added info would be useful such as: do all the author's savanna fires show up as being in a savanna in the remote-sensing products? | We agree that there is much ambiguity on what constitutes "savanna". We also found that some of the fires we lit in protected areas got marked as "croplands" in the IGBP classification of the MCD12Q1 product. These classes are listed in column BC "MOD_vegtype" of the Excel sheet for each sample. When aggregating to larger pixels, the land use classification was based on the dominant type in the 0.25° grid cell, meaning some of the nuance is lost.

An important side note –too often forgotten when upscaling to global models– is that most samples (whether they are EFs or fuel loads or combustion completeness), are obtained in protected and relatively undisturbed areas. However, most of the area classified as savanna is not.

We added the following statement (P10 L33): "Using the IGBP classification, our samples were classified as "Woody savannas" (24%), "Savannas" (42%), "Open shrubland" (21%), "Grassland" (4%), "Cropland/Natural vegetation mosaic" (6%) and "Croplands" (1%). The latter two classes are misclassifications and were all situated in protected areas with no crops. These classes are listed in the accompanied dataset (Vernooij, 2023)." |
| There is also considerable difficulty/uncertainty in field-measured fuel consumption, etc. Easier than adding many columns for uncertainties would be at least generic uncertainties in the table explaining the data set. The error bars in the figures do look generous to the author's credit. Again, it might be worth stating how the local variability compares to full, propagated uncertainty? | Just as with the comment above, we agree but do not have a fully satisfying way forward. Some of these issues also play a role when building other components of GFED and in the end often an aggregated expert-judgement uncertainty estimate is used. |
| CH4 is exceptionally dependent on MCE, but not all important emissions are as seen in Yokelson et al. (2003) and other work including Andreae 2019. | That is correct, in this paper we only present the results for the emission species that were directly measured. However, as some species may be scaled using MCE as you show in Yokelson et al. (2003), we will |

| | |
|---|---|
| | also add MCE to the downloadable data files in the future. |
| It seems the Excel spreadsheet is giving time as local time | This is true. We have added "(local time)" to the column name |
| The spreadsheet seems not to include background samples. ERs and background values can be derived from slopes and intercepts, respectively. By subtraction of the "_em" column from the "_abs" column, it appears there was a fixed background for each fire. These backgrounds are interesting in themselves. For instance, one fire had a background of 0.17 ppm CO, which is pretty low compared to the 1-5 ppm CO background that can occur during regional smoke episodes during peak fire season. | This is correct. Shortly before lighting the fire, four background samples were taken at 15m. The average mixing ratio of these samples is then taken as the background for all the samples in that fire.

In the revised Excel table (provided in the zenodo file) we have included the background values pre-fire in a separate sheet.

Particularly for $CO_2$ and $N_2O$ (mostly due to the low signal) they fluctuate significantly compared to the excess mixing ratios in the samples. |
| As we also see by FTIR (but don't report), there were negative N2O emissions and EF at times. How were these negative emissions handled in further data processing? | We found that the Aeris Pico analyzer was less accurate at low concentrations due to temperature and pressure stabilization issues which are now addressed in the "Ultra" model which was not yet used in our work but will be in future work. In Vernooij et al. (2021)'s Figure 11 we show this issue is mainly important at low carbon EMRs where we find both high and low $N_2O$ EF extremities. As mentioned in P8 L7, we excluded samples which contained less than 10 moles of total carbon emissions for the calculation of the WA $N_2O$ EF as we deemed these samples too uncertain.

It is very interesting that you also find negative $N_2O$ emissions in your FTIR measurements. Besides measurement error, could $N_2O$ consumption in flaming combustion (Winter et al., 1999) be a cause? In our work we ignored this and assumed it is mostly a measurement error which cancels out when taking multiple samples. |
| There are a number of non-physical values in the spreadsheet easily found by plotting the columns in a line chart. E.g., rows 2209-2211, 2353, and especially 2382 and 3116. These data were presumably not used in the training or validation and might be removed? | Indeed, samples with negative emissions for $CO_2$, CO or $CH_4$ were omitted from the training and validation data for the further analysis. We have now deleted them from the spreadsheet to avoid confusion. |

| | |
|---|---|
| The letter and number convention for the sample names, does it have any significance that should be explained? | These codes refer to flight and sample numbers of the individual bags. Although we used them to allocate times and coordinates and use comments, they do not have any further role in the analyses.

In the "dataset explanatory table" provided in the zenodo link, we added a description of the letter and number convention of the sample and fire names. Since they combine with notes, photos, lab results, etc., they are mainly helpful for us if someone has questions regarding certain data. |
| Why are EF calculated for the cals? | This is indeed an error in the script. Since the calibration samples are filtered out for the statistical analysis, these EFs do not affect the models. In the new version we have removed EFs for the Cals. |
| Why is no date/time given for the cals? | The date and time in the sheet refer to the date and time of sampling which are logged by the sampling unit on the drone. Since the calibration gas bags are manually filled from a canister on the ground, sampling date and time are not logged. They were filled before starting the analyses around sunset on the same day of the fire. |
| Why are the cals not all the same or nearly the same? Were there different calibration mixtures or does the scatter reflect the precision? | There was indeed more scatter in the calibration samples than the measurement precision (provided by the manufacturers) indicates. To mimic the measurement method, we have first filled bags with the calibration gas and then fed them into the analyzer rather than straight from the canister. Uncertainties may thus relate to both the measurement precision and the sampling.

Average calibration values ($\pm$ std) measured in the field were:
$CO_2$: 4732 $\pm$ 128
$CO$: 102 $\pm$ 7.3
$CH_4$: 15.1 $\pm$ 0.36
$N_2O$: 1.14 $\pm$ 0.047

We have added the following to the discussion (P13 L17): "The difference in the mean calibration value compared to the calibration gasses was -4.75% for $CO_2$, - |

| | 1.32% for CO, -3.97% for $CH_4$ and -1.28% for $N_2O$. Although the measurements were linearly correlated using the calibration bags for the individual fires, the standard deviations between the calibration samples were 2.58% for $CO_2$, 7.06% for CO, 2.32% for $CH_4$ and 4.04% for $N_2O$, indicating larger measurement uncertainties than reported by the manufacturers, which possibly arises from the bag methodology." |
|---|---|
| I was surprised that field-measured temperature had poor correlation with the satellite temperature in Table 4. Then I noticed in the spreadsheet that the temperatures measured on the drone correlate with CO2. In general, the temperature, RH, and VPD seem to be measured in the convection column at times where they would reflect the heat and water production of the fire, rather than an ambient air value that would influence fire behavior. If this is the case, I suggest replacing sample-specific values from the drone with one best ambient value per fire and (if not already done) seeing how that correlates with measured EF and remote-sensing products. Or did the authors use pre-fire met data measured differently or on the drone during the pre-fire cal and that data is available somewhere else?

For example. Picking one fire randomly, EDS19_3 on a June afternoon in Mozambique, one notices that Tsat is close to the climatological average high for June in Maputo (26 C), but is well below the lowest Tdrone (33.57 C). Is that a shade versus sun-exposed thing? Was there a Tdrone during a cal or background that is more appropriate? Further, VPDsat is only close to VPDdrone at minimum Tdrone suggesting combustion products make VPDdrone not representative of ambient VPD unless a VPDdrone measured in background air was actually used? Likewise the RH comparison reveals differences. | That is correct, the values listed in this column were logged using a temperature sensor on the drone (a safety feature) and are in no way representative of the general conditions without fire. Although we at some point reprogrammed it to also log T and Rh after changing batteries this often occurred in still hot burn scars and we found these values were also not helpful.

The only thing these values represent is the conditions under which the sample was collected. These values were not included as predictor features in the models. To avoid confusion, we will remove the columns from the data sheets. |

| In the LGR N2O-CO instrument, the N2O data needs to be corrected for CO and the correction only works up until 5ppm CO. This is because at high CO values, the CO line broadens enough to interfere with the N2O line. In general, the strongest N2O band is overlapped by water, CO2, and CO (and other gases). The CO values in the author's spreadsheet are in the 100s. The manufacturer of the author's N2O instrument (AERIS) product literature claims to use an interference-free, but unspecified, alternative spectral region and have an upper limit of 500 ppm for some unspecified molecule (probably CO?). Kudos to the authors for not using LGR for N2O, but I am curious if the authors have any evidence against or for CO interference in their N2O data? I am not assuming issues exist, but if they can be ruled out, it would be worth mentioning as N2O is an important, but undersampled fire emission | To help address this concern, we contacted Dr. Jerome Thiebaud from AERIS, who explained it this way:

The following statement may be true at the LGR wavelength in the near infrared, but not at the Aeris wavelength in the middle infrared:
"This is because at high CO values, the CO line broadens enough to interfere with the $N_2O$ line. In general, the strongest $N_2O$ band is overlapped by water, $CO_2$, and CO (and other gases)."

The Aeris gas analyzer operates at low pressure to minimize spectral congestion and near a wavelength of 4.5 microns where $N_2O$ absorption lines free of any interference (including from water, $CO_2$, and CO) can be measured in typical atmospheric gas mixtures.

We did not find any evidence of interference. But unfortunately, we do not have access to calibration gases with known $N_2O$ mixing ratio and varying CO mixing ratios to test this. |

In the line-by-line comments, the table below only includes the comments that required some additional explanation or answer. In all other cases, we took over the reviewers' suggestions which are revised accordingly in the 'Track changes' document.

| Reviewer 2, Bob Yokelson line by-line comments | Author's response, reasoning and comments |
|---|---|
| 2/1 Not 100% sure what is meant here. It almost reads like the biome average EF is 60-85% off on average. I think you mean e.g. if a measured fire had an EF 10% below the biome average EF, the satellite-based recalculation of the EF would be ~6-8.5% below the biome average? | Not entirely, what is meant is that in your example the absolute error would be 1.5 - 4% (below or above average) instead of 10%. We have changed the sentence to (P1 L38): "RF models using satellite observations performed well for the prediction of EF variability in the measured fires with out-of-sample correlation coefficients between 0.80 and 0.99, reducing the error between measured and modelled EFs by 60–85% compared to using the static biome average." |
| 2/3 change in CO2 totals? (expect small) | The difference in $CO_2$ emissions over the entire timeframe was -0.2% compared to |

| | the static EF average. We added this to the text (P2 L3). |
|---|---|
| 2/2-3 It's amazing that the global totals based on average biome EFs were within 1.8 to 18% of global totals using dynamic EFs. The difference is much smaller than the uncertainty in almost any other thing. However, it should be clear what biome average EFs are employed here. Probably the old literature average? Also, is good agreement seen every year or just for the 14 year total? | This was indeed surprising to us, particularly since our measurement averages (overrepresented by xeric regions) suggested much larger deviations.

The 'static average' we compare with are the GFED4s EFs which are not updated with current literature. However, for these species and savannas, these are similar to those proposed for FINN 2.5 (Wiedinmeijer et al. (2023) preprint).

When comparing to the EFs suggested by Andreae (2019), the differences would be larger.  We added the following text (P11 L7): "Both our measurements and the savanna biome averages in literature compilations (e.g. Akagi et al., 2011; Andreae, 2019) are subject to sampling bias when representing global savannas. A disproportionate number of field studies are clustered around reactively accessible locations with a well-developed research infrastructure, whereas other fire-prone areas lack direct field measurements. Rather than comparing the average of our savanna measurements to the literature averages, we computed the dynamic EFs globally using the RF model and subsequently calculated the emissions for the entire savanna biome. We then divided these annual emissions by the consumed biomass from GFED4s to get the annual consumed-biomass-weighted-average EFs, which we will further refer to as the "effective" EFs. Over the 2002-2016 period, the effective EFs over the savanna biome were $1685 \pm 5$ for $CO_2$, $64.3 \pm 0.6$ for $CO$, $1.9 \pm 0.0$ for $CH_4$ and $0.16 \pm 0.00$ for $N_2O$, with the number in the parentheses indicating the interannual standard deviation. In Table 4, we compare the effective average EFs over the 2002-2016 period calculated by our model to the static average EFs for savanna and grassland vegetation used by GFED4s and those suggested by Andreae (2019) and Wiedinmyer et al. (2023). Table 4 also lists the average EFs of the UAS measured fires |

| | and the average EFs of all included fires (including literature studies). Except for $N_2O$, the differences between the effective EFs compared to more recently updated static EFs from Andreae (2019) were larger (+1.3% for $CO_2$, -7.1% CO, -31.4% $CH_4$ and -3.7%) than the differences compared to static EFs from GFED4s. |
|---|---|
| Ultimately, the paper could compare the old literature average EFs to the evolved literature average EFs that include the author's new data, and the average EFs based on just the authors new work. I.e. how much impact does this study have on averages? Finally, in addition to predicting measured EF better, it would be interesting to know if the use of dynamic EFs also better predicts downwind impacts, but that might be another paper. | Many thanks for the great suggestions. As mentioned in the paper, the averages of our own measurements deviated more from the previous static averages than the 'effective EFs' listed above. This is mainly because a disproportionate number of our measurements were done in Xeric savannas. We feel this sampling bias makes it unwise to add our samples to the biome average without weighing (i.e. the effective savanna EF). Given the comments above about uncertainty we are also more careful now in stating the the biome-average values is different.

Upon request these effective EF averages can also be calculated for individual regions or timeframes (e.g. EDS vs LDS). We hope that our work will encourage researchers to step away from using average values.

These emission factors have been used in a paper which compares bottom-up and top-down (TROPOMI) data which provides encouraging results (Van der Velde et al., in preparation) and will become part of GFED5 |
| 2/5-6 Did not the authors observe that CO and CH4 EFs decreased with drying in xeric grasslands, but increased with drying in mesic woody savannas? Also "… annual average savanna fire …" | That is indeed the case, we changed the sentence to (P2 L5): "Over the course of the fire season, drying resulted in gradually lower EFs of these species. Relatively speaking, the trend was stronger in open savannas than in woodlands where towards the end of the fire season they increased again." |

| | |
|---|---|
| 2/7 Are there just reductions? There is good agreement on totals so there should also be localized increases. In general, from the 1-sigma standard deviation in literature EFs we expect +/- 40% variation in EFs fire-to-fire 1-sigma. | Indeed, the models also predicted increases. Since average $CH_4$ and particularly $N_2O$ EFs were lower, the largest localized deviations were reductions.

We changed the text to (P2 L7): "Contrary to the minor impact on annual average savanna fire emissions, the model predicts localized deviations from static averages of the EFs of CO, $CH_4$ and $N_2O$ exceeding 60% under seasonal conditions." |
| 2/15 60% of net emissions? Could deforestation and peat be more important in the C-cycle if minimal regrowth? | We clarified the text (P2 L15): "They estimate that, due to their high burning frequency, savannas account for roughly 60% of the gross (i.e. not considering regrowth) global carbon emissions from biomass burning (BB)." |
| 2/26-28 There are many direct field measurements and they quantify overall variability, but previously we could not account for the total variability with quantitative contributions from very many specific factors. Previous studies targeted the average and variability, but not the causes of variability. | We changed the text to (P2 L26): "Although there are many direct field measurements and they quantify overall variability (as summarized in for example Akagi et al., 2011 and Andreae, 2019), to date we cannot quantify how specific factors such as moisture content impact EFs (van Leeuwen and van der Werf, 2011)." |
| 3/25-26 Could other real-time data besides that from satellites be useful? | Our aim, for the implementation in global emission inventories, was to have a global coverage over a considerable timespan (at least the MODIS era). Any dataset with a record long enough to train models and NRT availability can be useful.

Supplementing the datasets on which the models are trained with alternative data should be done carefully to avoid biases. However, substituting for instance ERA5-land temperature with ERA5 temperature to achieve NRT capacity, or even T predictions from CMIP5 projections as you suggested can be useful for NRT applications. |
| 4/23 The pre-fire met data mentioned here, where is it? The spreadsheet has non-useful met data collected in the fire convection column. | That is correct, the pre-fire conditions were logged in a similar fashion but using the background measurements before the fire was lit. Although we started logging the windspeed, temperature and relative humidity using a Kestrel fire weather sensor, that was only true for the very last |

| | experiments and not useful to analyze the full record.

In the revised Excel sheet, we added the background data including the relative humidity and temperature when available. |
|---|---|
| 4/35 One naturally wonders here if the authors field environmental data can be used for insight into the accuracy of the global satellite products and were their fires detected by the satellite products GFED4s uses? | As previously stated, we added the following text (P15 L17): "Of the UAS-measured fires in this study only 5 of the 45 EDS fires (11%) and 13 of the 65 LDS fires (20%) were registered by MCD64A1 as burned area (including adjacent pixels and a 4-day time lag) and only 4 of the 45 EDS fires (9%) and 32 of the 65 LDS fires (49%) were registered by VIIRS S-NPP as thermal anomalies (with the center point of the hotspot (including a 1-day time lag) being within a 3.5 km radius of the sample). Depending on the spatiotemporal nature of these omissions, this may affect some of the results in this study concerning the effects of the EF dynamics on total emissions. Chen et al. (2023) indicate that in the savannas, disproportionately more burned area is added in higher tree-cover areas when using higher resolution satellite imagery. Giving more weight to these areas would mean our savanna-wide effective EFs of $CO$, $CH_4$ and $N_2O$ would increase. The Sentinel-2 based burned area product from Roteta et al. (2021) performed much better and registered 8 of our 14 EDS fires (57%) and all of our 16 LDS fires (100%) in Botswana and Mozambique in 2019 (including adjacent pixels and up to a 21-day time lag). Due to the fewer overpasses the temporal allocation of this product is less precise with an average time lag of 5.5 days. Figure 10 shows the portion of our EDS and LDS fires that were detected by various satellite algorithms." |
| 5/7 Impressive set of products. Is it easy to explain why no VIIRS or geostationary? Not available as long? Useful going forward? | Our aim was to have a global coverage for the implementation in global emission inventories (using a uniform approach) while covering at least the MODIS era to look at global trends. Therefore, we did not consider geostationary satellites at this stage. |

| | However, since all our own measurements are from the VIIRS era the models can be trained using VIIRS data as well going forward. As only 4 of the 45 EDS fires (9%) and 32 of the 65 LDS fires (49%) were registered by VIIRS S-NPP as thermal anomalies, including VIIRS as a feature would therefore result in a lot of missing values which then have to be removed from the training data. We added a sentence to the text (P5 L15): "We used remote sensing products based on retrievals and reanalysis data with sufficient spatial and temporal coverage, primarily using products based on the Moderate Resolution Imaging Spectroradiometer (MODIS). This meant that at this stage, we did not include data from VIIRS or geostationary satellites." |
|---|---|
| 5/8 Were all the samples of a fire usually in the same feature pixel? | For the courser features like ERA5-land this was the case, although feature values may differ between samples based on their timestamp. For the MODIS derived features the samples of the individual fires covered $1 - 13$ pixels with an average of 2.5 pixels per fire. |
| 5,/15 Is it easy to explain why not using historic NDVI range? | We are not sure whether we fully understand the question, but we focused on Pgreen. Pgreen is the NDVI before the fire, relative to the NDVI range of the pixel throughout the year. As further explained later, the reason this did show us as a strong indicator may be the pixel misrepresentation of the actual burned vegetation. |
| 5, 25-26, TRMM useful for rainfall? | Since TRMM was in operation from 1997 to 2015 and our measurements are done between 2017 and 2022, TRMM rainfall cannot be used to train our models.

We have experimented using IMERG data for rainfall but decided to use ERA-land as we were more interested in consistency for broader patterns than highly accurate readings. |
| 5/30-31 risk or behavior or both? Are any ideas in the "hot dry windy index" useful as predictors here? | Many thanks for the suggestion. We were not familiar with this product and will surely include it in updates. The individual parameters that go into the DHW (VPD and windspeed) were included in training the |

| | models and were (not surprisingly) found to be strong predictors. |
|---|---|
| 5/34-35 Is the daily cycle of fine fuel moisture captured? Was FFMC compared to the author's field-measured fine-fuel moisture data? | No, although we did include the diurnal cycle of VPD, ESI, T, WS and RH. However, FFMC was obtained from EFFIS CEMS, at a daily resolution.

The FFMC compared very poorly to our measured weighted average fine fuel moisture content with a Pearson correlation coefficient of -0.36. This may explain why in itself (not as part of FWI), the FFMC was never assigned as one of the main EF predictors by the models. |
| 6/6 How does spatial resolution of the fire severity proxies (dNDVI etc.) compare to the size of fires? If the fire is smaller, then is the signal diluted? Would a small severe fire look like a larger less severe fire? Did the authors expect better correlation of scorch and char height with the severity proxies? | Mismatch of the burned vegetation and the pixel retrieval is indeed an issue for these features.

We added the following text to the discussion (P15 L2): "Fire intensity proxies (dNDVI and dNBR from MODIS) were poor predictors for the EFs. A potential explanation is that these features were at times heavily diluted, as many of the measured fires only affected part of the pixel. Similar misrepresentation errors can be expected for the NDVI before the fire, FPAR and the Pgreen. Particularly in the LDS, we were often limited to areas that were enclosed by recent fire scars (0-2 years) or other non-flammable boundaries. Although these areas were sizable (several hectares) many of the retrievals in these pixels may poorly represent the burned vegetation. Along with inconsistent retrievals related to cloud cover, this may be an important reason why these features were deemed poor predictors by the models while seen as strong predictors in previous research (Korontzi et al., 2004). Higher resolution features may increase the representativeness of the pixels for the actual burned vegetation." |
| 6/19-20 What is "a measurement with a missing value of an included feature"? Do you mean you did not use EF measurements if even a single associated satellite product out of the whole set was missing? | That is indeed what we meant. The models cannot deal with missing values. We decided to drop those measurements rather than using average feature values as this could distort the relations. This was only an issue when including the full set of features |

| | to decide the most important predictors. For training the eventual models we used a subset of five features. This had both the benefit of reducing the requirements for data availability and computational demands. These features did not have missing values. |
|---|---|
| 6/21-22 What does "resampled using ten-fold cross validation while allowing sample replacement (i.e., bootstrap method)" mean? Can a simple plain language explanation be added? | Ten-fold cross-validation is a technique used to evaluate the performance of a random forest model by splitting the training data into multiple subsets or "folds". The entire dataset is divided into ten equal-sized parts or folds. The random forest model is trained and evaluated 10 times. In each iteration, one fold is used as the validation set, and the remaining 9 folds are used as the training set.

By using ten-fold cross-validation, we can get a more robust estimation of how well the random forest model performs on unseen data. It helps to reduce the bias that may arise from using a single train-test split and provides a better understanding of the model's generalization capabilities.

Random forests are ensemble models that combine multiple decision trees to make predictions. The bootstrap method starts by randomly sampling the original dataset with replacement. This means that for each sample in the dataset, there is an equal chance of it being selected more than once or not selected at all in the bootstrap sample. This also helps to create an ensemble of diverse decision trees and contributes to the model's robustness and generalization capabilities.

We changed the text to (P6 L31): "We removed measurements with missing values for any of the included features. The remaining data was divided into training (70%) and validation data (30%), and the training data was resampled using ten-fold cross validation. This means that the training dataset is divided into ten equal-sized parts or folds. The random forest model is trained and evaluated 10 times. In each iteration, one fold is used as the |

| | "temporary validation" set (different from the 30% which is not included in the training data), and the remaining nine folds are used as the training set. The folds are created while allowing sample replacement (i.e., bootstrap method), meaning that for each sample in the dataset, there is an equal chance of it being selected more than once or not selected at all." |
|---|---|
| 6/22-23 Explain that "hyper parameter" refers to the most influential parameters? | Hyperparameters refer to the settings or configurations that determine how the random forest algorithm operates. These parameters are not learned from the data but are predefined by the user before training the model. These are for instance the number of trees, tree depth and number of features per split. These features depend on the amount and variability in the data and are used to avoid overfitting.

We changed the text to (P6 L39): "The hyper parameters (model configurations like number of trees, minimum samples per leaf, maximum features, etc.) were tuned using the scikitlearn "GridsearchCV" algorithm (Pedregosa et al., 2011)." |
| 6/28-30 This is hard to follow. How would an EF require a resolution and how would that be computed? Overlap is within or between features? Do you mean some fires were bigger than or occupied more than one grid cell in the original feature (note we have slipped into calling remote-sensing proxies "features" for short), so you averaged, or extrapolated, or built a new grid for each fire such that the fire was centered in a single grid cell? Sometimes a few extra words can help a lot! | We rewrote this section to make it easier to follow (P7 L6): "To assess the impact of EF dynamics on emission estimates, and study global spatiotemporal patterns, we developed gridded EF layers that can easily be incorporated into existing emission inventories. The remote-sensing proxies ("features") were resampled to the required spatial resolution by simply averaging the values of the relevant gridcells. For example, to compute the 0.25° fraction tree cover feature, we averaged the fraction tree cover of all 500-meter pixels classified as savanna or grassland." |
| 6/32 How can an EF have a temporal resolution? Are the EFs referred to fire-average or sample-specific? Is the daily cycle in RH and fine fuel moisture considered? | This refers to the gridded product. We clarified the text to (P7 L14): "The temporal resolution of the computed gridded EFs in the example of Fig. 2 is daily, in which the day-to-day EF dynamics are being driven by daily variations in VPD, FPAR, FWI and soil moisture." |

| | The BA data we used to calculate global emissions using GFED4 is daily. Therefore, it did not make sense to calculate our EFs at higher temporal resolution.

However, in calculating the daily EFs, we did consider the daily cycle for RH, VPD and T. As we state later in the text (P7 L17): "For features with a typical diurnal pattern, we therefore weighed the hourly meteorological data by the average diurnal fire profile in the respective month for the grid cell. This diurnal fire profile was based on the three-hourly fractions of daily emissions obtained from GFED4.1s, which is based on the timing of active fire detections from both MODIS and geostationary satellites (Mu et al., 2011; van der Werf et al., 2017)."

This means rather than taking the average daily average, the daily averages were weighed by when fires in the grid cell typically occur at that time of the year. |
|---|---|
| 6/40 "… savanna fire emissions …" Were the dynamic EFs calculated using global products and RF? | Correct, we changed the text to (P7 L21): "To study the impact of EF dynamics in savannas, we calculated monthly global savanna emissions by multiplying the dynamic EFs computed by our models with dry matter consumption from GFED4s (Randerson et al., 2012; van der Werf et al., 2017) at 0.25° spatial resolution, for the 2002-2016 period (the period for which MCD64AC5 as used in GFED4s was available)." |
| 7/15-16 How were samples with negative N2O emissions treated when calculating fire-average N2O emissions? | For samples with a total increase in carbon below 10 moles, the increase was calculated as:

$$\Delta C = \int \frac{EMR_{CO_2}}{MM_{CO_2}} + \frac{EMR_{CO}}{MM_{CO}} + \frac{EMR_{CH_4}}{MM_{CH_4}}$$

We did this to avoid the (assumed) measurement error found in very low signal samples following Vernooij et al. (2021).

Because these samples had low EMRs and the fire-averages are calculated over the cumulative EMR in all the bags, their |

| | omission did not significantly affect the fire-averaged $N_2O$ emissions. |
|---|---|
| 7/19-20 Clarify this is the Andreae 2019 average and not the average of the 85 measurements used from other groups? Otherwise, how do you get locations for study-average or vegetation average emissions (unless one fire in study)? | That is correct. We changed the text to (P8 L2): "The relatively small range in the boxplot describing previous savanna literature (Fig. 3, red box based on studies listed by Andreae (2019)) may be attributed to the fact that most studies report either fire-averages, vegetation type averages or even study averages, whereas the other boxplots based on our measurements show the variability observed between individual samples." |
| 7/23 substantial variability in fire-averages or samples? | The boxplots represent the variability in the individual samples. To some extent, this variability also translated to fire averages but that is not shown here. To prevent confusion, we changed the text to (P8 L8): "We observed substantial variability within EF bag samples from different savanna ecosystems." |
| 7/24-25 The higher CO and CH4 EFs in woody savanna is supported in previous literature at least once, e.g. Sinha et al., (2004).

FWIW, the Miombo fire was included in the tropical dry forest category in Akagi et al, but it was also a small part of a savanna fire study-average used in the savanna category. | Thanks for pointing this out. The higher CO and CH4 EFs were indeed in line with previous literature and expectations. We added the reference to the discussion. |
| 7/25 Taking this to mean the authors study-averages were lower than previous literature averages. | Correct, that is the average of all our measured fires. It should be noted that the aim of our campaigns was to cover spatiotemporal variability rather than getting a representative average of all fires in the savanna biome. The biome-average is not the same as our sampling average because we oversampled xeric regions. The relatively low EFs we measured are therefore to be expected. The issue with sampling bias is also true when taking study averages like Andreae (2019). |
| 7/29 by "seasonally inundated grasslands" do you mean aka dambos? | Correct, we changed the text to (P8 L14): "In humid areas like dambos (seasonally inundated grasslands) and riverine forests, …." |
| 8/2-3 Any benefit to comparing the authors fuel measurements to similar measurements | The fuel measurements are definitely very interesting in itself and will be further |

by Shea et al (1996) and Hoffa et al (1999) and others?

Shea, R. W., Shea, B. W., Kauffman, J. B., Ward, D. E., Haskins, C. I., and Scholes, M. C.: Fuel biomass and combustion factors associated with fires in savanna ecosystems of South Africa and Zambia, J. Geophys Res., 101(D19), 23551–23568, 1996.

studied, and used in different applications (e.g. to inform DGVMs and emission inventories). In this paper, however, the goal was not to look at fuel in detail but rather to use those measurements to explain patterns in EFs and EF-satellite correlations, so we only discuss them briefly.

We added the following text (P13 L34): "Measurements of fuel loads were higher than previous measurements from African savannas described by Shea et al. (1996). They found average fine fuel loads (litter and grass) of 3.8 tonne ha$^{-1}$ in moist Miombo woodland. In semiarid Miombo woodland they found 3.1 tonne ha$^{-1}$, In comparison we found 5.6 tonne ha$^{-1}$ in Mozambican Miombo woodland and 5.6 tonne ha$^{-1}$ in Zambian Miombo woodland. The percentage of grasses in these fuels was similar; Shea et al. (1996) reported 24% in moist Miombo woodland and 18% in semi-arid Miombo woodland whereas we found 37% in Mozambican and 18% in Zambian Miombo woodlands. The combustion completeness of these fuels was slightly lower in our fires at 50-80% versus 80-92% reported by (Shea et al., 1996), albeit that the lower values in this range occurred in the EDS. Combustion completeness of shrub leaves and course woody debris were in the same range. For dambo grasslands our fuel loads were also much higher at 6.2 (±2.16) tonne ha$^{-1}$ of which 99% grass versus 3.1 tonne ha$^{-1}$ from Shea et al. (1996). Although these differences are large, they may be attributed to the significant natural variability in productivity and decay related to water availability, fire frequency, and termite and grazing activities in these natural landscapes."

8/4 What is meant by "corresponding mixtures of fuel age"? In Table 3, why was a higher percent of the heavy fuels consumed in the EDS in Australia, unlike elsewhere; maybe lit more aggressively?

By "the columns do not necessarily represent corresponding mixtures of fuel age" we mean that for some vegetation types or season, we may have more measurements of older fuels than for others. This may affect things such as litter load,

| | nitrogen content, etc. regardless of the seasonal effects. We changed the text to (P8 L32): "For some characteristics (e.g., the total fuel load), it is important to note that the measurements in the different vegetation types do not necessarily represent identical mixtures of fuel age. The higher fuel loads open savannas in Australian compared to Botswana, may be partially attributed to the longer fuel build-up."

We also added the average time since the last fire to Table 3.

Fires were lit in similar fashion in the different vegetation types, including those listed in Table 3. In the Australian sites, grasses were very dominant and abundant and heavy fuels were scarce. The sample size of heavy fuels being very low may also explain why this deviated so much from the other areas. |
|---|---|
| 8/10-13 I'm pretty sure that increased RSC and increased CO and CH4 EFs in the LDS in wooded savannas is already in the literature but haven't found the reference. Maybe Hoffa or Korontzi? | The measurements described by Hoffa et al. (1999) are all performed between June 5th and August 6th and therefore miss the period we refer to. Korontzi (2005) does indeed predict a slight increase (recovery) in CO and CH4 EFs from September to October (Fig. 11) for both woodlands and grasslands. This increase, however, is very small compared to the overall pattern and EFs are still much lower compared to EDS values. Contrarily, we found EFs that were higher in LDS woodland fires compared to EDS fires. |
| 8/32 For Table 4, clarify which field-measured met data were compared to satellite met data, preferably NOT drone data in fire-processed air! However, Table 4 seems to specify that T and RH from the drone were used, which could be okay if NOT when drone was above the fire, but instead in ambient (background) air. Then again, currently, it's odd that the satellite temperature and drone temperature are weakly positively correlated at 0.18 while satellite temperature is most strongly correlated with field measured nitrogen content in the grass (perhaps a seasonal coincidence?). | The drone data during the fire are indeed not indicative of environmental conditions, but rather sampling conditions. In Table 4, we replaced the temperature and relative humidity with the values taken while making background measurements.

Background measurements were obtained before the fire which can be several hours earlier than the latest samples. As both T and Rh are strongly diurnal, these values may not always represent the environmental conditions during the fire. With respective spearman correlation coefficients of 0.21 and 0.45 for T and Rh compared to their |

| | ERA5-Land counterparts, correlation was slightly higher but still not great. |
|---|---|
| 8/31-33 This text and Table 4 could be clarified with slightly more precise and consistent terminology. I think Tab 4 shows how the *field measured-ecosystem attributes* correlate with the field-measured MCE and EFs and also how the *field-measured ecosystem attributes* correlate with the satellite products, but NOT how satellite products correlate with field-measured EF or how anything correlates with model-calculated EF? At this point in the paper, evidently, calculated EFs vs measured EFs and the sensitivity of calculated EFs will be discussed elsewhere. | We changed the title of Table 4 to: "Table 4. Spearman correlation matrix for the field-measured-ecosystem attributes and the fire-averaged emission factors and MCE as well as the satellite products used in the study. Positive correlations are presented in blue while negative correlations are presented in red."

Also, we added Table A2 to the appendix (Table 1, in this document below) which shows how satellite products correlate with field-measured EF or how anything correlates with model-calculated EF. |
| 9/4-5 I'm taking this to mean that 70% of field-measured EF were used with "features" to train the RF model and the RF model then used features to predict EF for the other 30% of field measured EF (out of sample means fires not in training set) and performed well in terms of r-squared. Could give the slope too? Is the training set randomly selected or varied run to run? | Correct. The out of sample performance refers to the comparison of the 30% validation data against the modelled EFs based on the validation data features. These data were not included in training the model.

The train-test split was randomly selected although the "random state" was then fixed. Rather than to optimize the results, this is done to make sure the models can be reproduced. |
| 9/5-7 Is there a simple way to connect feature importance and the concept of hyper parameters? Is "impurity decrease" essentially a fraction of total variability? | The feature importance is calculated as the total reduction in the node impurity that a feature contributes to when it is used for splitting in all the individual decision trees. This impurity is calculated as the probability of misclassifying a randomly chosen data point within that node. The feature importance provides an overall measure of how much each feature contributes to the predictive power of the entire RF model.

Hyperparameters are unrelated to the features used as predictors or the feature importance but refer to the settings or configurations that determine how the random forest algorithm operates. |
| 9/8 The red line in Fig 4 is useful for comparing the range of EF to the old literature average. But later in paper, the effect of dynamic EF should perhaps be | You are correct. The effect is both the mismatch of our (xeric dominated) dataset to the savanna average and the effect of dynamic versus static. Moreover, in |

| | |
|---|---|
| compared to the biome average based just on the field data used by the authors, which could be shown with a second vertical line. Then recalculate MAE and improvement %. Currently, the comparison is "apples to oranges" in that "improvement" is based on a difference resulting partly from incorporating new data and partly from a change in approach. | particular for $N_2O$, the older "static average" represented by the red vertical line is not up to date. Andreae (2019), which includes more recent studies, reports a savanna $N_2O$ EF of 0.17 which would reduce these mismatches.

In Fig 4 and 5, we have added a separate magenta vertical line representing the average of the input data. |
| 9/13-20 This is a nice exploration of simplifying the RF approach. Can the authors explain why VPD is the most important feature in the small subset of features, despite having a low rank in the full set of features? Any estimate of reduced computational burden? | The VPD is strongly seasonal and correlates strongly to other features from the full set of features like temperature, relative humidity, soil moisture, and evaporative stress index. This means that similar decision trees can split the data similarly following any of these features, so in a way they are competing. This reduces the impurity reductions (and thus feature score) of those features.

The smaller feature subset has several advantages, including reduced computational burden, less dependencies on underlying datasets, easier to make NRT data, and no data losses due to missing values. |
| 9/24-25. Does this mean you ran the RF model once to get MCE and then used the MCE as a new feature in a re-run of the RF model? | That is correct. Thanks to your work we know that MCE is strongly correlated to the EFs of particularly $CO_2$, CO and $CH_4$. By first computing the MCE and then offering that as a feature, we can isolate the effect of MCE from other effects making it more informative. Also, we found that doing it this way improved the overall predictive performance of the models. |
| 9/35-38 It would be interesting to see the study-average EFs vs the former literature average EFs and then also what the new literature averages are including this study, all in a little 3x4 table. | Previous studies have often used the average of all the available measurements as the savanna average EF. However, in selecting our field campaigns, we were interested in capturing variability and dynamics, rather than determining a representative savanna average EF. Many of our measurements target les fire prone conditions and, for instance, from a representability perspective oversample the EDS and xeric savannas.

For those interested in a single number for savannas, we would suggest taking the |

| | effective EFs rather than the average of the samples. These values represent the average modelled EFs weighted by the consumed dry matter from GFED4s. This means they only include EFs at the time and location that the savanna fires occurred (according to MCD64A1), which eliminates the sampling bias in global measurements.

As requested, we added Table 4. We also added the following text (P11 L17): "In Table 4, we compare the effective average EFs over the 2002-2016 period calculated by our model to the static average EFs for savanna and grassland vegetation used by GFED4s and those suggested by Andreae (2019) and Wiedinmyer et al. (2023). Table 4 also lists the average EFs of the UAS measured fires and the average EFs of all included fires (including literature studies). Except for $N_2O$, the differences between the effective EFs compared to more recently updated static EFs from Andreae (2019) were larger (+1.3% for $CO_2$, -7.1% CO, -31.4% $CH_4$ and -3.7%) than the differences compared to static EFs from GFED4s." |
|---|---|
| 10/2 How common are mixed biome grid cells? Percentage of total? Is the most common type of mix with tropical dry forest? Is there a percent tree cover or canopy closure that defines the boundary between what the authors consider savanna and something else? | This depends on the resolution desired for the model. In this study, we aggregated the data to 0.25-degree grid cells meaning mixed grid cells were quite common. For the biome classification, we used the biome classification from GFED4s, which is based on the annual International Geosphere-Biosphere Programme (IGBP) classification and obtained from MCD12Q1, in which classes 5-10 make up our "savannas and grasslands". This means we did not have a "tropical dry forest" class. The IGBP classification uses a FTC cut-off of 60% to distinguish the "woody savanna" and "forest" classes. |
| 10/11-12 What is "annual effective EF"? An annual global savanna-fire average EF for each compound? This is also saying the year to year variation in global average EFs is small? | That is correct, the "annual effective EF" was calculated by multiplying all the GFED4s biomass consumption by the dynamic EFs at the time and location of burning, and then dividing these annually integrated annual emissions by the integrated annual biomass consumption. |

| | This way we get a savanna EF weighted by the time and places that burned. |
|---|---|
| | $Eff. \; EF_x \; (year) =$ |
| | $\dfrac{\Sigma_{days} \, \Sigma_{grid \; cells} \; (BC \, (day, grid \; cell) \times EF_x \, (day, grid \; cell)}{\Sigma_{days} \, \Sigma_{grid \; cells} \; BC \, (day, grid \; cell)}$ |
| | (1) |
| | We clarified the text (P11 L11): "Rather than comparing the average of our savanna measurements to the literature averages, we computed the dynamic EFs globally using the RF model and subsequently calculated the emissions for the entire savanna biome. We then divided these annual emissions by the consumed biomass from GFED4s to get the annual consumed-biomass-weighted-average EFs, which we will further refer to as the "effective" EFs. Over the 2002-2016 period, the effective EFs over the savanna biome were $1685 \pm 5$ for $CO_2$, $64.3 \pm 0.6$ for $CO$, $1.9 \pm 0.0$ for $CH_4$ and $0.16 \pm 0.00$ for $N_2O$, with the number in the parentheses indicating the interannual standard deviation." |
| 10/14-15 averaged over what time and space? I.e. the daily average over all areas occupied by the indicated vegetation class? Fig 7 doesn't seem to show much or any EFCO increase in woody savanna as the fire season progresses? Does this figure clash with previous text? What is "typical savanna"? | The graph contains monthly CO EFs averaged over the 2002-2016 timeframe, for all the areas occupied by the indicated vegetation class. The vegetation classes are based on the IGBP classes. We have changed "Typical savanna" to "Savanna" (referring to tropical regions with Tree cover 10-30% (canopy >2m). |
| | We agree that in the graph, the upward trend is not as evident for savanna and woody savanna as the measurements seem to indicate. Although we did focus on southern hemisphere Africa in this graph, there are still some effects of temporal mismatched between east and west and north and south that may dilute these patterns. Also, as you mentioned earlier, these classifications are not always correct. |

| | |
|---|---|
| 10/30-34 Interesting, shows the RF model may have value to at least partially correct sampling bias in a field campaign! | Exactly! |
| 11/2 Just to be clear, the N is in the foliage of the trees, not the wood itself | We changed the text to (P11 L39): "In line with Susott et al. (1996) and Ward et al. (1992) we found that woody vegetation has higher nitrogen content contained in the foliage (Table 3), causing higher $N_2O$ emissions from tree dominated areas." |
| 11/24-26 Did Hoffa and Korontzi predict higher MCE in LDS? | That is indeed incorrect. We changed the reference to Korontzi (2005), which is a temporal extrapolation through Pgreen (also assessed in this study) based on measurements from Hoffa. |
| 11/30 The Eck trend in SSA is averaging over all sub-Saharan Africa AERONET sites? | It used three sites (in Etosha, Namibia), Kruger national park, South Africa, and Mongu, Zambia) which are discussed separately. While all sites show an increasing SSA trend over the dry season, the trend is strongest in Mongu where the signal is probably the most dominated by fires. |
| 11/32 References that support an increase in SSA as MCE decreases include Liu et al and Pokhrel et al and probably many others | Thanks, we have added the references to the text. |
| 11/40-12/1 Not sure about the interpretation here. Does CH4/CO vary with MCE? CO is not technically independent of MCE since MCE has CO in its definition. | Although the main point here is that this relation varies with FTC, you are correct. MCE and CO are linear. Therefore, the fact that the $CH_4$ EF/MCE ratio varies with MCE also means that the $CH_4$ EF/CO EF ratio varies with MCE. |

| 12/5-9 This discussion could be misleading in a subtle way. I think the effect seen here is probably because the other studies compared to are plotting the fire-average EFCH4 versus the fire-average MCE, while the authors are plotting EFCH4 vs MCE for "snapshot grab samples" that could include samples during flaming that may have much higher MCE than the fire-average MCE for typical useful real-world fires. We've seen this often over the years. To illustrate we can revisit the comparison to the Selimovic et al lab fire study. If one plots the instantaneous EFCH4 vs instantaneous MCE for these typical lab fires you often get "curvature" at high MCE values during "pure flaming" and other effects. The ERCH4 vs MCE can also be non-linear at high MCE or have interesting other interesting patterns with time. The plots show this for the 1-s data from randomly selected Fire #74 on the NOAA FIREX-Firelab archive (https://esrl.noaa.gov/csd/groups/csd7/measurements/2016firex/FireLab/DataDownload/). Fire #74 is one of the fires in the linear plot of fire-integrated EFCH4 vs MCE in Selimovic et al. (2018). Interesting topic but variability during a fire is a level of detail large-scale models can't cope with yet. Thus, in providing guidance for large-scale models it may be best to stick to fire-average data. | Many thanks for this clarification. We agree that this effect is much smaller in fire averages due to the limited range in MCE and behaves linearly. In Figure 1 (below this table) we have added the fire-averages and linear regression based on those averages. It shows a similar pattern for fires with exceptionally low MCE. Your graph indicated the eventual fire-average $CH_4/CO$ ratio (and thus the $CH_4$ EF/MCE ratio) is dependent on the ratio between smouldering and flaming combustion in the fire, which may be expected to correlate with FTC. Therefore, while the pattern is certainly more pertinent in individual bag samples, it may also hold for fire-averages.

 We feel the main point of this text, that studies that disproportionately target smouldering or flaming emissions would reach different linear $CH_4$ EF/MCE slopes, is still true and confirmed by the graph.

 We therefore changed the text to (P12 L37): "In accordance with previous studies (e.g. Korontzi et al., 2003b; van Leeuwen and van der Werf, 2011; Barker et al., 2020), we found steeper $CH_4$ EF to MCE regression slopes in woodlands compared to grasslands. Our data indicated a positive correlation of the $CH_4$ EF to MCE slope with the FTC based on MOD44Bv006. The MCE is a simplified form of the combustion efficiency and only calculated using CO and $CO_2$ emissions. Being less oxidized than CO (which is still common in flaming combustion), $CH_4$ emissions have a stronger dependency on the actual combustion efficiency ($CO_2$ divided by all carbon emissions). While most studies describe the relationship between the $CH_4$ EF and the MCE as being linear (Korontzi et al., 2003; van Leeuwen and van der Werf, 2011; Selimovic et al., 2018; Yokelson et al., 2003), we found that for individual bag samples it was better described using a nonlinear function (Fig. 9), in line with findings by Meyer et al. (2012) for Australian savanna measurements. Figure 9 represents |

| | individual bag measurements rather than fire averages (for which the spread in MCE is much lower). Laboratory experiments described by Selimovic et al. (2018) showed that the $CH_4$ to CO ratio is strongly dependent on flaming or smouldering phases if the fire. Individual bag samples –which often hold emission from a single phase– therefore show much more variation compared to fire averages. Stable carbon isotopes also point to $CH_4$ emissions being more depleted in heavy carbon ($^{13}C$) compared to CO in both mixed (C3 and C4) and single-fuel-type experiments, indicating a stronger dominance of RSC and the pyrolysis of lignin in its total emissions (Vernooij et al. 2022b). This explains both why studies that are skewed towards either smouldering or flaming phase emissions find different $CH_4$ EF to MCE slopes using linear regressions and why this slope varies with FTC. " |
|---|---|
| 12/25-26 Think you mean "This is the first study to quantify the spatial distribution of GHG EFs over the entire savanna biome by using both field measurements from a variety of savanna ecosystems and their relation to global data mainly from satellites". I.e. the field measurements have gaps as explained in the following lines, but by connecting the measurements to features you have a new way to get a useful global savanna estimate! | As suggested, we have changed the text to (P14 L9): "This is the first study to quantify the spatial distribution of GHG EFs over the entire savanna biome using field measurements from a variety of savanna ecosystems and their relation to global data mainly from satellites." |
| 13/11 The idea of a gross underestimate here is worrisome. How well do the authors think GFED4s accounts for fires too small to show up in their burned area product? Worth mentioning here? | The 'gross underestimate' is compared to the GFED4s burned area used in this study. To clarify this we changed the sentence to (P15 L16): "New high-resolution burned area products, however, indicate that these global products, including the GFED4s data used for global emission analyses in this study, grossly underestimate burned area due to omission of small fires (Chen et al., 2023; Roteta et al., 2021; Roy et al., 2019). This also refers to a significant portion of our measured fires. Of the UAS-measured fires in this study only 5 of the 45 EDS fires (11%) and 13 of the 65 LDS fires (20%) were registered by MCD64A1 as burned area (including adjacent pixels and a 4-day time lag) and only 4 of the 45 EDS fires |

| | |
|---|---|
| | (9%) and 32 of the 65 LDS fires (49%) were registered by VIIRS S-NPP as thermal anomalies (with the center point of the hotspot (including a 1-day time lag) being within a 3.5 km radius of the sample). Depending on the spatiotemporal nature of these omissions, this may affect some of the results in this study concerning the effects of the EF dynamics on total emissions. Chen et al. (2023) indicate that in the savannas, disproportionately more burned area is added in higher tree-cover areas when using higher resolution satellite imagery. Giving more weight to these areas would mean our savanna-wide effective EFs of CO, $CH_4$ and $N_2O$ would increase." |
| 14/10 Here I think it's important to preserve the idea that you have not concluded the biome averages have large errors, just that fire to fire variability is large and is better accounted for by using a more sophisticated model. Also + and − local errors tend to cancel. It worries me that someone reading quickly may think you mean that global CO and CO2 emissions from savanna fires are off by ~80%. | We agree that particularly compared to errors in other model aspects like BA and fuel load these errors are limited. We changed the text to (P17 L7): "The model-produced data resulted in significant fire-specific improvements compared to static biome-averaged EFs, reducing the mean absolute error in the modelled versus measured predictions by 63% for $CH_4$, 57% for $N_2O$, 81% for CO and 79% for $CO_2$. Except for $N_2O$ EFs, our study does not indicate that savanna averages have large errors, but rather that fire to fire variability is large and is better accounted for by using a more sophisticated model." |
| 14/31 I did not check the zenodo link. If it is different from spreadsheet, I could check it by request. | The data is indeed the same as the spreadsheet provided. |
| Fig 7. Why do "typical savanna" fire emissions peak earlier than all the subtypes? | This may be an artifact of the spatial distribution of the different savanna classes. In general, but particularly for woody savannas, there is a trend in the SHAF region with western areas burning sooner in the year than eastern savannas. In figure 2 (below) you can see that the frequently burning "savanna" class areas are more situated in the western part of the region.

Another possible explanation would be more fire suppression in shrublands and grasslands. |

**Table 4: Emission factor averages of this**

| EF Specie | GFED4s | Andreae (2019) | Wiedinmeijer et al. (2023) | Sample data avg.[1] | Training data avg.[2] | Effective EF (Eq. 1)[3] |
|---|---|---|---|---|---|---|
| $CO_2$ | 1686 | 1660 | 1686 | 1637 | 1670 | 1685 |
| CO | 63 | 69 | 63 | 55 | 61 | 64 |
| $CH_4$ | 1.94 | 2.70 | 2.00 | 1.38 | 1.61 | 1.85 |
| $N_2O$ | 0.20 | 0.17 | | 0.12 | 0.12 | 0.16 |

[1]Average over the fires measured using the drone methodology (skewed towards xeric savannas)
[2]Average over the fires measured using the drone methodology and the included literature studies.
[3]Dynamic EFs weighted by the consumed biomass at time and location of fires as calculated using GFED4s.

**Table A2. Spearman correlation matrix for the field measurements and the globally available satellite products. Positive correlations are presented in blue while negative correlations are presented in red.**

[Figure]

[Figure]

**Figure 1. The non-linear regression between the CH₄ EF and the MCE for the individual bag samples. In the box on the bottom left, ρ refers to Spearman's rank correlation coefficient measured in the bag samples. The orange linear regression line is the linear regression of fire-averages.**

[Figure]

**Figure 2. Distribution of the IGBP landcover classes used in figure 7 of the main text.**

[Figure]

**Figure 10. Detection of the fires measured using the UAS-methodology by different satellite algorithms in the EDS (green) and LDS (orange). The darker area represents the cases where a fire was observed in the actual pixel within the listed timeframe. The lighter areas represent fires that were not detected in the same pixel as the samples but were detected in adjacent pixels. Timeframes are listed below the product labels. For the VIIRS detections the distance limits between the detection point and closest sample of the fire were 1km for the darker shaded area and 3.5 km for the lighter shaded area.**

---

## Author Comment (AC3)

**Response of the authors to comments by reviewer 2 on the manuscript: "Dynamic savanna burning emission factors based on satellite data using a machine learning approach"**

**General comments:**

This study collected a large dataset of savanna emission factors (EFs), including over 4500 EF bag measurements of $CO_2$, CO, $CH_4$ and $N_2O$ during 129 individual fires from 2017 to 2022. Based on this in-situ observations, the authors identified the drivers of EF variability and implemented this variability into global models through dynamic EFs. The optimized machine learning reduced the error in EF estimates by 60-85% compared to static biome averages. They also found seasonal drying resulted in a decrease of the EFs with the fire season progressing, with a stronger trend in open savannas than woodlands. Overall, this is an important study to understand the variability and mechanisms for biome-specific carbon emissions, particularly at the spatial scales. The generated global EF products can be used to better estimate fire-induced greenhouse gas emissions. However, I do have some concerns on the methodology parts, which may need to be addressed before publication.

**Roland Vernooij (corresponding author) on behalf of the authors:**

We sincerely thank the reviewer for the time and effort in assessing our manuscript, and the constructive comments which helped to improve the quality of this paper. Please find below our point-to-point response to the review. The revised text and updated figures are included in the updated manuscript. A separate 'track-changes' document is included to highlight the changes to the manuscript. Tables and figures referred to in the answers are added at the bottom of this document.

| Reviewer 2, detailed comments | Author's response, reasoning and comments |
|---|---|
| 1). Biomass burning EFs are highly dynamic both at the spatial and temporal scales for a given fire. For example, EFs may differ a lot as the fire spreads across different vegetation covers and terrain/moisture gradients at the local scale. How well did the collected EF bag measurements represent the total or averaged EFs for each selected fire? Is there a consistent spatial-temporal framework to integrate the concrete EF measurements to reflect the total EFs for all involved fires? Such processing details need to be provided for better understanding the uncertainty of "in-situ" measurement itself. | You raise a valid point which we can only address empirically. In Vernooij et al. (2022), we have described comparisons on EFs measured using a measurement mast and the UAS method (Figs 2 and 3). In this comparison we found that a limited amount of bag samples (8-12) resembled the fire-averaged EFs of the mast relatively well considering the spatiotemporal heterogeneity. The strategy was therefore to take 8-12 samples at a location until visible smoke from both flaming and smouldering had passed the drone and continue to the next location. The fire-average EF of the individual fires is calculated by adding up the EMRs from all the individual bags and calculating the EFs based in that sum. This means samples with high EMRs have a stronger effect on the fire-average EF than low-EMR samples. |

To make sure the features are representative of the vegetation consumed in the fire, we tried to target larger homogenous areas. Rather than marking the fire with features from the entire fire scar, the features are assigned to the samples based on location and timestamp of the sample. The average of all the sample features is then assigned to the fire-WA EF, meaning we only assign data from when and where we sampled.

It might be possible to better quantify the spatial and temporal variability within a fire when continuous measurements can be done while being airborne. Currently, however, the equipment is too heavy to be carried by the drone but this may change in the future and at that stage the reviewer's question may be addressed better.

We added the following text (P3 L32): "Fires were lit with the aim of being representative of early dry season (EDS, often prescribed) fires and late dry season (LDS) non-prescribed fires. Although some backfires were sampled during the initial phase of the fires, the majority of samples were obtained from the faster 'head' fires, which consumed most of the biomass. Fire sizes generally ranged between 2 to 10 hectares based on UAS drone imagery described by Eames et al. (2021), with exceptions of some fires that would not light and conversely, some fires that burned several hundred hectares. In the EDS, fire size was primarily limited by environmental conditions and fires ceased burning as humidity increased overnight whereas in the LDS, fire size was confined by low-fuel areas like burn scars, roads and prepared fire breaks. Particularly in the LDS, this means a limited fire size does not necessarily indicate limited fire intensity. Emissions were sampled at altitudes between 5–50 m depending on flame height for a duration of 35 seconds, resulting in 0.7 litres per gas sample. On average, we took 35 samples per fire. The sampling methodology involved taking samples from

| | a fire passing a certain point –while correcting for wind direction and severity– until no more visual smoke passed the drone anymore. From earlier work (Vernooij et al., 2022a), where we compared the average of these measurements to results using continuous measurements taken at a mast, we have some confidence in the fidelity of this approach." |
|---|---|
| 2). How did the fire induced EFs match the possible drivers at the spatial scale? Are they overlayed by the actual size of each fire, or just at the grid size of 0.25 degree? The latter may introduce large uncertainties. | For the training data we assigned the features to the fire using the highest resolution available. For instance, the fractional tree cover (FTC) would be assigned based on the 500 x 500-meter pixel value (or the weighted average of several of these pixels). For features with a strong diurnal pattern (e.g. VPD, RH or temperature) we took the hourly data, and interpolated this to assign the feature value at the minute of sampling. However, the spatial resolution of these datasets is typically relatively coarse (0.1°) introducing uncertainty.

In our global analysis we indeed averaged out feature data over the 0.25° grid cell (filtering out non-savanna vegetation), and subsequently computed the EFs. This was done to match the spatial resolution of GFED4 and analyze global patterns. When looking at smaller regions, using the native resolution of the features (e.g. Figure 2) may reduce these uncertainties but we expect our data to be used mostly within coarse-scale applications. |
| 3). The authors tested a series of machine learning methods and concluded that random forest performed best. Such a part may need data support. Past experiences suggest that the gradient boost MLs such as lightGBM and Xgboost tend to be better than random forest. | We appreciate the advice; We did not use the tools you mention but will include this in future work to test whether the results improve further.

When we started this research we have used a suite of approaches (using the scy-kit learn "GradientBoostingRegressor()" function and GridSearchCV hyperparameter tuning) and actually found that the RFs performed slightly better than GBMs, although the difference was very |

| | small. In comparison, we also tried multilinear regressions, single decision trees and a simple neural network which all performed worse than RFs and GBMs. Since we include the MCE (which is often used as the sole predictor of EFs) from our RF model into the EF models as a predictor, one could argue that the EF models have a sort of gradient boosting step on that RF MCE model. |
| --- | --- |
| | Table 1 (bottom of this document) lists the RMSE and $R^2$ for different model runs (in which "All data" includes field observations, "Sat data includes Landsat and lower resolution >500m satellite data and "LR satellite data" only includes >500m resolution data). The random forest models have improved after this initial assessment. However, in line with your comment, we also found that in various runs the GBM regressor outperformed the RF regressor. |
| 4). To predict BB EFs, the authors included a series of factors for each group driver (seen in Table 2). However, it seems that some of them are highly correlated, e.g., NDVI VS LAI VS FPAR, VPD VS evaporative stress index VS Relative humidity. The rationale for including these redundant factors may need to be clarified. In addition, given the potential uncertainty in remote sensing and reanalysis data, it may not be wise to include all predictors without doing a feature selection. One way to include a specific driver or not is to compare its effect with a randomly generated variable. If its effect is equal or worse than the random variable, it may not be included in the final training. | We fully agree; many of the features from the full set of features (Table 2) are strongly seasonal and therefore correlated strongly to other features, or even calculated based on one another. The analyses using the full set was mainly to detect which performed better. We then did a feature selection going from a broader set of features (e.g. Fig. 4), to the five features that explained most of the observed variability (Fig. 5, listed in the bottom right box of the panels). The eventual models are trained based on only these features. |
| | In the discussion we state (P16 L8): "Cross-correlation between the features meant that feature importance scores (Fig. 4) varied over various model runs based on the test-train data split and bootstrap resampling. For example, a decision tree split based on VPD is most likely very similar to soil moisture or RH, and FTC in national parks is often closely correlated to the MAR, with our measurement sites in Brazil being the notable exception. Although we conducted model runs for various feature-subsets and selected the best, different features may |

| | |
|---|---|
| | also perform well in explaining much of the variability. For features with very high co-variation (e.g., FPAR and LAI or FWI and ISI), this meant only one feature was selected for the trimmed-down model even when both features scored high on the initial assessment."

Rather than taking the features with the highest feature scores, we realize that many of the features are correlated and therefore explain the same variability. We took the correlation between features into account when selecting the features for the final models. We added a Pearson correlation matrix (below this table, and as table A2 in the appendix of the manuscript) that lists the correlation between satellite features as well as the direct correlations of satellite features with the target variables. |
| 5). Satellite data over tropical regions usually suffers from the contamination of clouds. When deriving the global EFs, how the authors gap-filled relevant remote sensing data is not clear. | Indeed, particularly in the late dry season we found that cloudiness was a problem, especially for retrievals like NDVI before fire, dNDVI, dNBR, etc. If the scene before or after the fire was cloud-covered, the preceding or successive scene was used with a limit of 14 days before or after the fire. If no cloud-free scene was available in that time window, the fire was removed from the dataset.

For the features included in the final models, this was less of an issue given that the meteorological reanalysis data from ERA5-land is not impacted. Fractional tree and non-tree vegetation (MOD44bv006) as well as landcover classification (MCD12Q1C6) are annual while FPAR is based on an 8-day composite meaning the risk of no signal are much lower. When aggregated to 0.25 degree (while using a savanna mask to only take the average of the savanna classified pixels), there were no longer missing values.

We added the following sentence to the text (P9 L40): "Further simplification using a subset of features that are not directly correlated, reduced the data dependency |

| | and computational demands of the model as well as the loss of training data due to cloud cover, without losing much explained variance." |
| --- | --- |

**Table 1: Performance of various regression methods during our initial assessment. This assessment only included our own data. In the variable categories "All data" includes field observations, "Sat data includes Landsat and lower resolution >500m satellite data and "LR satellite data" only includes >500m resolution data.**

| Method | Variables | MCE RMSE | MCE R$^2$ | CH$_4$ EF RMSE | CH$_4$ EF R$^2$ | N$_2$O EF RMSE | N$_2$O EF R$^2$ |
|---|---|---|---|---|---|---|---|
| Multilinear regressor | All data: | 0.03 | 0.56 | 1.45 | 0.66 | 0.13 | 0.19 |
|  | Sat data: | 0.03 | 0.46 | 1.36 | 0.59 | 0.28 | 0.18 |
|  | LR Sat data: | 0.03 | 0.46 | 1.43 | 0.55 | 0.35 | 0.16 |
| Decision tree regressor | All data: | 0.031 | 0.70 | 1.38 | 0.76 | 0.05 | 0.21 |
|  | Sat data: | 0.023 | 0.61 | 1.05 | 0.71 | 0.08 | 0.40 |
|  | LR Sat data: | 0.022 | 0.66 | 1.00 | 0.70 | 0.07 | 0.59 |
| **Random forest regressor** | **All data:** | **0.026** | **0.80** | **1.16** | **0.87** | **0.04** | **0.47** |
|  | **Sat data:** | **0.027** | **0.70** | **0.98** | **0.65** | **0.06** | **0.65** |
|  | **LR Sat data:** | **0.021** | **0.70** | **0.88** | **0.76** | **0.06** | **0.64** |
| Gradient boosting machine regressor | All data: | 0.023 | 0.78 | 1.13 | 0.85 | 0.04 | 0.30 |
|  | Sat data: | 0.028 | 0.67 | 1.00 | 0.67 | 0.06 | 0.58 |
|  | LR Sat data: | 0.021 | 0.70 | 0.90 | 0.75 | 0.07 | 0.56 |
| Neural network regressor | All data: | 0.035 | 0.73 | 1.16 | 0.84 | 0.04 | 0.50 |
|  | Sat data: | 0.021 | 0.62 | 1.09 | 0.69 | 0.07 | 0.59 |
|  | LR Sat data: | 0.022 | 0.61 | 1.05 | 0.65 | 0.06 | 0.60 |

**Table A2. Spearman correlation matrix for the field measurements and the globally available satellite products. Positive correlations are presented in blue while negative correlations are presented in red.**

[Figure]

---

## Referee Report (RR1)

Second review by Bob Yokelson

Steps I took for my second review. I skimmed the responses to all the posted comments when they were first posted and did not notice any issues. When asked to re-review, I almost just okayed the article based on the responses, but I decided to read the track-changes document to double-check implementation and I do have some minor suggestions based on this second reading. I did not re-check the SI or the revised Excel spreadsheet, but could if requested.

Overview/summary:

1/ The authors are familiar with a large array of products at various spatial and temporal resolution, but the average reader may find this hard to sort out. I think I realized in the second reading that the new product is basically monthly and 0.25 degree resolution? Is that right? If so, maybe highlight that bottom line in the abstract and mention any plans to increase resolution in future in the conclusions.

2/ The fire terminology is "corrected," or at least I suggest how to harmonize with how it is implemented by US fire managers.

3/ There's a few places with repeated sentences or awkward flow.

4/ Earlier studies targeted a most representative single EF. This led them to attempt conducting random sampling of fires of opportunity using aircraft at the peak of fire season in the most active areas. There's little opportunity for detailed measurements at the burn in this approach. The approach the authors adopted here targets dynamic EFs, but then also relies on fires set by scientists rather than local farmers in order to facilitate collection of data pre and post burn on site. That adds a layer of uncertainty about how faithfully scientists reproduce native practices. It's also true that events can derail the accuracy of this higher resolution approach at the single fire level. I.e. if it rains one day in a dry month that will effect DM and EFs with successively less impact over the next few days.

Line by line on the track changes ms.

P, L# comment:

2, 28-9 How about: "Thus current global inventories are not designed to quantify any variation in average emissions at the local or monthly scale." Maybe replace "local" with something more quantitative? In general, this region of the paper could be a good place to integrate the first part of overview comment #4 above. The part of that comment about fire authenticity might be part of the fire description text in the methods.

2, 31 Add "weather," before "climate change"?

3, 4 Change "measuring EF measurements" to "measuring EFs"

4, 8-12 The same sentence appears twice in a row

5, 28 I think this was in first set of comments and might have been addressed in response. I wonder why NDVI is normalized to the previous year's range rather than a longer-term average. It seems a wet year could throw off the next year, but maybe that issue is not easy to fix?

6, 27 WA = weighted average? Check if already defined?

7, 15 CO EFs at 500 m resolution mentioned here is a bit confusing since the features often only have 0.25 res or in any case are not available at 500m

7, 18-24 It's still not explicitly clear how fine fuel moisture was computed. For instance, the daily cycle in fuel moisture could be important https://www.nwcg.gov/publications/pms425-1/weather-and-fuel-moisture. Typically fine fuel moisture could range from 15% in early AM to 5% in late afternoon. Is FFMC an average near 10%? If active fire products suggest burning peaks in afternoon, is the fine fuel moisture adjusted to assume afternoon conditions for every fire in a 0.25 degree grid box?

7, 24 So here is where we learn that this study is taking us from one static EF or an assigned EF for EDS and LDS to monthly EFs, but not higher?

7, 25 EFs, MCE, and DM consumed are all impacted by the environment. Does DM consumed vary based on these features or is that built in to GFED already?

8, 11 Not required but possible that this section could be a place to remind less experienced readers of the historical "one EF" motivation for simply reporting a study average measured on a big plane with lot's of instruments and a fast speed to access a lot of fires and operating from a base at the peak season/area.

8, 15 add any impact on MCE?

8, 18-19 isn't DM-consumed more important than burned area here?

8, 28 May have asked in round one, but how does measured fuel consumption compare to the fuel consumption predicted by GFED FC?

8, 30 "corresponding mixtures of fuel age" is a bit nebulous unless you define fuel age. Do you mean time since last burn, or something else? This fuel age concept comes up again on page 9, line 3.

8, 35-38 Here you seem to clearly indicate there was more RSC late in dry season, at least in the humid savanna. On line 38, I would move live foliage before the list of RSC-prone fuels. Have you defined RSC? You might find the Bertschi et al 2003 JGR paper useful for that. Or one could term this phenomenon as "post-frontal combustion."

9, 8 The dry Australian savannas had lower EFs, but was DM-consumed also lower or was that offset by longer fire-return intervals?

9, 9 One thing was obvious during our field work in Zambia. Within a radius of settlements, much of the woody debris is collected for household firewood. On the outskirts of Kaoma we saw people pushing bicycles with logs tied to the seat and handlebars headed to a local sawmill.

The landscape was clearly managed differently within say 50 miles of Lusaka compared to more remote areas.

10, 4 So to me, this means the MAE in MCE using static MCE is ~1.62*0.007 = 0.011. If that's not right, maybe explain more?

10, 24 Soil moisture may act more like a "long time-lag" (1000 hour) fuel?

10, 30 Again one wonders how common are grid cells that are a mixture of savanna and non-savanna? This may have been partly addressed in the added text about misclassification of savannas as cropland.

11, 7&10 I think it was decided in response to another reviewer that there's no such thing as "typical savanna"?

11, 25-26 There are no parentheses and two EFs have stdev of zero? The interannual variability seems low at < 1% maybe?, and certainly lower than real accuracy?

12, 11-13 & 17-19 same sentence twice

12, 16 As an alternative to saying fires get more intense, or as the probable cause, is the concept that "the fuels get more receptive."

12, 37-40 choose one version of sentence. I boycott the concept of "fires getting "hotter"" since no-one has ever defined how to measure the temperature or, much less, the extent of a "whole fire." This is in contrast to concepts like "flame temperature" or "combustion completeness", which have straightforward definitions. Also, you have already said there was more RSC in the wooded savanna in the LDS, which would lower the MCE.

13, 6-8 Standing alone this sentence could be interpreted to mean RSC is not a factor across the whole savanna biome. The previous sentence tries to qualify it, but with a new term "open savannas" Thus, I would qualify this sentence as follows: "We found that, in the xeric savannas, the composition …"

13, 20-22 I recommend changing "Laboratory experiments described by Selimovic et al. (2018) showed that the CH4 to CO ratio is strongly dependent on flaming or smouldering phases if the fire. Individual bag samples -which often hold emission from a single phase- therefore show much more variation compared to fire averages."

To "Laboratory experiments described by Selimovic et al. (2018) and others showed that the CH4 to CO ratio is more complex and variable in real-time than at the fire-average level. Individual bag samples, which are closer to a spot sample from a single point in time therefore show much more range and variation compared to fire averages."

The version of Fig 9 in track-changes is not the same as the revised Figure in the response showing the fit to both spot samples and fire-average data. I like the new version with two fits better and thought the authors intended to upgrade?

13, 22-25 Indicate if this applies only to mesic or wooded savannas.

13, 25-27 This is a lot in a short sentence. How do you know if studies are skewed, how would that effect slopes based on first principles, how does FTC fit in? I'd either explain all these things in full or just delete the sentence since it may not be that important.

13, 29-33 Should this summary of analytical uncertainty go in the methods section?

13, 36 EESGT defined?

13, 35-37 could a higher elevation of African sites mean cooler temps and less evaporation than in Brazil?

14, 8-17 is Mg/ha better than tonne/ha? Line 17 missing word, add "was" before "grass"?

14, 19 Is grazing taken into account in GFED or IGBP cover types?

14, 22-24 Don't capitalize "New" or delete "This".    Change "high" > "monthly". Are the burned areas new or EFs or both? I don't know what you're trying to say in this sentence, it needs major revision.

14, 32 and throughout. Technically there is no such thing as a "head fire"; it's "shorthand." Based on my four years as a wildland firefighter and with some back-up on terminology from: NWCG Glossary of Wildland Fire Terminology PMS 206 https://www.nwcg.gov/publications/pms205 I think it's best to encourage using the following terms clearly and consistently among fire professionals:

Heading fire: a fire burning (advancing) in the same direction as the wind, often at high rate of spread and patchy, especially if also uphill.

Backing fire: (sometimes "backburn" in the Queen's English>): a fire advancing into and against an opposing wind, generally at a slower rate of spread and with higher MCE and combustion completeness, but less RSC

Backfire: This is a specific type of fire set under special circumstances. A backfire is lit inside the fireline and must be drawn into an approaching flame front by the local surface wind induced by the convection column of the approaching fire. When done properly it will deprive the approaching fire of fuel to facilitate control or, alternatively, increase the intensity of a poorly-burning prescribed fire.

Blacklining or burning out: These are narrow strips of fire set along and inside the fireline to effectively thicken the control line one strip at a time

In use: Typically a road or handline is selected as the control line or fireline. Then, if there is time and the fire danger warrants it, by burning increasingly broader strips along the fireline, the area deprived of fuel can be enlarged to widen the control line (this step called burning out or blacklining). A back fire is when the approaching fire, often a heading fire, but can be backing too, creates a local sea breeze affect drawing surface air into the convection column from all directions. Standing on the control line, facing the oncoming fire, one starts to feel a wind on their back. At this point, the backfire can be lit and it is drawn into the main advancing fire to

deprive it of fuel. Setting a backfire is best left to experienced personnel with high-level decision authority because a backfire attempted too soon or too late can be ineffective and endanger crew.

14, 34 suggest changing "backburning" to "backing fire"

14, 35 suggest changing "backfires" > "backing fire"

14, 36-37 RSC can be increased in a heading fire because the high rate of spread and patchiness leaves fuels smoldering further from the convection associated with the advancing flame front.

15, 4 Higher resolution weather might be even more important than increased spatial resolution. If it rains before your fire, that changes a lot for a few days at least. And the duration of the rain is more important than total amount in terms of soaking the fuels.

15, 11-13 Is this topic out of place here and maybe fits better elsewhere?

15, 23-24 I don't think you mean these predictors worked better 20 years ago, but the sentence kind of gives that impression?

15, 28 can be, but were not, correct?

16, 10 It's a good discussion, but one starts to wonder if the discussion jumps around a bit and might be better organized. Maybe worth a small effort to improve, but okay.

16, 18-19 You might want to say "have high confidence in" rather than "be sure about." The part of the sentence after the comma doesn't add much.

16, 36 "can" > "could potentially" subtly acknowledges the work involved

17, 1 "Conclusion" or "Conclusions"

17, 2 delete "on"

17, 4 measured > sampled

17, 5 "using a UAS platform."

17, 9 could acknowledge impact of fuel receptiveness on fire intensity

17, 17 "dynamic EFs" > "monthly EFs at 0.25 degrees"? Or add specs in ()s after EFs. Retrievals > products?

17, 28 delete concept of "typical" savannas?

17, 30 RSC may not include live vegetation, which may burn better with more wind or more sustained ignition.

17, 39 "it's significance" > variability.   indicate > are

---

## Author Response (AR2)

**Response of the authors to the second round of comments by Bob Yokelson on the manuscript: "Dynamic savanna burning emission factors based on satellite data using a machine learning approach"**

Roland Vernooij (corresponding author) on behalf of the authors:

Again, we express our gratitude for the substantial time and effort dedicated to evaluating and improving our manuscript. The insightful and constructive feedback provided greatly contributed to enhancing the quality of this paper. Please find below our comprehensive response addressing each review point. The updated manuscript incorporates the revised text and updated figures, while a separate 'track-changes' document is provided to highlight the modifications made. Additionally, we have appended supplementary explanatory figures referenced in our responses at the bottom of this document.

| General comments by Bob Yokelson                                                                                                                                                                                                                                                                                                                                | Author's response, reasoning and                                                                                                                                                                                                                                                                                                                                                                                                                                                  |
|-----------------------------------------------------------------------------------------------------------------------------------------------------------------------------------------------------------------------------------------------------------------------------------------------------------------------------------------------------------------|-----------------------------------------------------------------------------------------------------------------------------------------------------------------------------------------------------------------------------------------------------------------------------------------------------------------------------------------------------------------------------------------------------------------------------------------------------------------------------------|
|                                                                                                                                                                                                                                                                                                                                                                 | comments                                                                                                                                                                                                                                                                                                                                                                                                                                                                          |
| 1/ The authors are familiar with a large
array of products at various spatial and
temporal resolution, but the average reader
may find this hard to sort out. I think I
realized in the second reading that the new
product is basically monthly and 0.25
degree resolution? Is that right? If so,
maybe highlight that bottom line in the | That is indeed correct, for the sake of this
assessment we have computed EFs at a
monthly base and 0.25-degree spatial
resolution. The future EF files will most
likely be made available at 0.25 degree in
both monthly and daily resolution to
integrate with GFED5.                                                                                                                                                                                          |
| abstract and mention any plans to increase
resolution in future in the conclusions.                                                                                                                                                                                                                                                                          | In the abstract, we clarified the text (P1
L34): "We then trained random forest (RF)
regressors to estimate EFs for CO 2 , CO,
CH 4 and N 2 O at a spatial resolution of
$0.25^{\circ}$ and a monthly timestep. Using these
modelled EFs, we calculated their
spatiotemporal impact on BB emission
estimates over the 2002–2016 period using
the Global Fire Emissions Database version
4 with small fires (GFED4s)." |
|                                                                                                                                                                                                                                                                                                                                                                 | In the conclusions we changed the last
sentence to (P18 L17): "Overall, the model
results are a first step towards more
dynamic and area specific emission
inventories, which we plan to make
available in monthly and daily resolution at
0.25° and will further improve as more
measurements and better remote sensing
products become available."                                                                                                      |
| 4a/ Earlier studies targeted a most
representative single EF. This led them to
attempt conducting random sampling of
fires of opportunity using aircraft at the                                                                                                                                                                                        | To structure our reply, we split point 4 into two separate issues.                                                                                                                                                                                                                                                                                                                                                                                                                |

| peak of fire season in the most active areas.
There's little opportunity for detailed
measurements at the burn in this approach.                                                                                                                                                                                       | We revised the text to (P2 L27): "Earlier
studies targeted a most representative single
EF. This led them to attempt conducting
random sampling of fires of opportunity
using aircraft at the peak of fire season in
the most active areas. There's little
opportunity for detailed measurements at
the burn in this approach. Although they
quantify overall variability (as summarized
in for example Akagi et al., 2011 and
Andreae, 2019), to date we cannot quantify
how specific factors such as moisture
content impact EFs (van Leeuwen and van
der Werf, 2011)."                            |
|------------------------------------------------------------------------------------------------------------------------------------------------------------------------------------------------------------------------------------------------------------------------------------------------------------------------------|---------------------------------------------------------------------------------------------------------------------------------------------------------------------------------------------------------------------------------------------------------------------------------------------------------------------------------------------------------------------------------------------------------------------------------------------------------------------------------------------------------------------------------------------------------------------------------------------------------------------------------------------|
|                                                                                                                                                                                                                                                                                                                              | Although we agree with the reviewer's statement that "the single representative EF is based on sampling of fires of opportunity using aircraft at the peak of fire season in the most active areas" we believe this applies somewhat less to savannas. In single EF estimates for savannas and grasslands, many of the underlying studies are local (small scale studies), that often include ground measurements and also rely on prescribed fires (e.g. experimental burn plots). Also, the timeframe and diversity in savanna vegetation types that frequently burn is incredibly diverse, making the biome less suited for a single EF. |
|                                                                                                                                                                                                                                                                                                                              | That being said, estimated EFs over the
whole savanna (at the time of burning)
were not far from previous estimates,
resulting in a limited effect on global
savanna emission estimates. Therefore,
when one is interested in year-on-year
estimates, using single EFs may suffice.
The added value of our work lies in the
spatio-temporal redistribution of these
emissions.                                                                                                                                                                                                                                   |
| 4b/ The approach the authors adopted here
targets dynamic EFs, but then also relies on
fires set by scientists rather than local
farmers in order to facilitate collection of
data pre and post burn on site. That adds a
layer of uncertainty about how faithfully
scientists reproduce native practices. | We agree that using prescribed fires raises
issues regarding the representativeness of
these fires to larger non-prescribed burns.
By coupling the estimation of the EFs to the
satellite derived local conditions this
uncertainty is somewhat mitigated. On top
of our own measurements, we also took                                                                                                                                                                                                                                                                                                                   |

|   | It's also true that events can derail the      | previous savanna studies into account.                                                   |
|---|------------------------------------------------|------------------------------------------------------------------------------------------|
|   | accuracy of this higher resolution approach    | fires.                                                                                   |
|   | at the single fire level. I.e. if it rains one |                                                                                          |
|   | day in a dry month that will effect DM and     | We took care in selecting widespread,                                                    |
|   | EFs with successively less impact over the     | homogenous and representative vegetation                                                 |
|   | next few days.                                 | types. However, there are many ways in                                                   |
|   |                                                | which areas differ (terrain, plant species,                                              |
|   |                                                | soil, grazing patters, fire and rainfall                                                 |
|   |                                                | all be accounted for at the same time. On                                                |
|   |                                                | top of that there are the weather conditions                                             |
|   |                                                | during the fire. The only way to remedy                                                  |
|   |                                                | this is more measurements. Our model                                                     |
|   |                                                | represent a first step in quantifying the                                                |
|   |                                                | broader variability that most studies agree                                              |
|   |                                                | exist. More measurements, indeed                                                         |
|   |                                                | improve the estimations of these patterns                                                |
|   |                                                | improve the estimations of these patterns.                                               |
|   |                                                | We added this to the discussion (P15 L1):                                                |
|   |                                                | "Most of the fires used to train the models                                              |
|   |                                                | were prescribed fires set by scientists or                                               |
|   |                                                | park rangers in protected areas in order to                                              |
|   |                                                | burn on site. It is common practice to                                                   |
|   |                                                | extrapolate these measurements in                                                        |
|   |                                                | relatively undisturbed savanna vegetation to                                             |
|   |                                                | the wider savanna. Even though these                                                     |
|   |                                                | protected natural areas tend to burn more                                                |
|   |                                                | frequently, they represent a minority of the                                             |
|   |                                                | area that is currently modelled using                                                    |
|   |                                                | savanna and grassland emission factors by                                                |
|   |                                                | gioual inventories (e.g. Fig. 1). Most of this area is to some degree affected by humans |
|   |                                                | though cattle ranging, wood harvesting                                                   |
|   |                                                | slash and burn agriculture, etc. This means                                              |
|   |                                                | fires in this study may not always represent                                             |
|   |                                                | the burning practices by local farmers and                                               |
|   |                                                | thus that representativeness of our work for                                             |
|   |                                                | the larger savanna area remains uncertain.                                               |
| ۱ |                                                |                                                                                          |

| Line by line comments by Bob Yokelson          | Author's response, reasoning and           |
|------------------------------------------------|--------------------------------------------|
|                                                | comments                                   |
| 2, 28-9 How about: "Thus current global        | As suggested, we added the text (P2 L27):  |
| inventories are not designed to quantify any   | "Earlier studies targeted a most           |
| variation in average emissions at the local or | representative single EF. This led them to |
| monthly scale." Maybe replace "local" with     | attempt conducting random sampling of      |
| something more quantitative? In general,       | fires of opportunity using aircraft at the |

| this region of the paper could be a good
place to integrate the first part of overview
comment #4 above. The part of that
comment about fire authenticity might be
part of the fire description text in the
methods.                                                                      |  <li>peak of fire season in the most active areas.</li> <li>There's little opportunity for detailed measurements at the burn in this approach."</li> <li>We also added the following sentence (P2 L31): 'Thus, current global inventories are not designed to quantify any variation in emissions at the local scale or at a monthly scale."</li>                                                                                                                                                                                                                                                                                                                                                                                                                                                                                                                                                                                                                                                            |
|----------------------------------------------------------------------------------------------------------------------------------------------------------------------------------------------------------------------------------------------------------------------------------------------------------|-----------------------------------------------------------------------------------------------------------------------------------------------------------------------------------------------------------------------------------------------------------------------------------------------------------------------------------------------------------------------------------------------------------------------------------------------------------------------------------------------------------------------------------------------------------------------------------------------------------------------------------------------------------------------------------------------------------------------------------------------------------------------------------------------------------------------------------------------------------------------------------------------------------------------------------------------------------------------------------------------------------------------|
| 2, 31 Add "weather," before "climate change"?                                                                                                                                                                                                                                                            | We changed the sentence to (P2 L33):
'Using historic averages also means that
EFs do not dynamically change while fire
regimes, weather patterns and
environmental burning conditions can shift
as a result of climate change or human
interaction.'                                                                                                                                                                                                                                                                                                                                                                                                                                                                                                                                                                                                                                                                                                                                                |
| 4, 8-12 The same sentence appears twice in a row                                                                                                                                                                                                                                                         | Thank you. This appears to be an issue with
turning the track changes doc into PDF
without comments. "Accepting" the
corrections removes the double sentences.                                                                                                                                                                                                                                                                                                                                                                                                                                                                                                                                                                                                                                                                                                                                                                                                                                               |
| 5, 28 I think this was in first set of
comments and might have been addressed in
response. I wonder why NDVI is normalized
to the previous year's range rather than a
longer-term average. It seems a wet year
could throw off the next year, but maybe
that issue is not easy to fix? | The formula for PGREEN was based on
equation 2 from Korontzi et al. (2015), who
found PGREEN to correlate with EFs for
CO and CH 4 . In their study they used the
NDVI range over the preceding growing
season. The reviewer makes a good point
that taking the average range over a longer
period would even out effects of atypical
seasons. In unproductive areas, fuel also
builds up over years meaning previous
years are likely to contain less biomass than
the fire year. On the other hand, including
longer periods would make it more likely
that fires or other disturbances artificially
influence the average. We would like to
consider the impact of this in future updates
and not change it at this point.
Korontzi, S.: Seasonal patterns in biomass
burning emissions from southern African
vegetation fires for the year 2000, Glob.
Chang. Biol., 11(10), 1680–1700,
doi:10.1111/j.1365-2486.2005.001024.x,
2005. |
| 6, 27 WA = weighted average? Check if already defined?                                                                                                                                                                                                                                                   | in P4 L16.                                                                                                                                                                                                                                                                                                                                                                                                                                                                                                                                                                                                                                                                                                                                                                                                                                                                                                                                                                                                            |
| 7, 15 CO EFs at 500 m resolution mentioned
here is a bit confusing since the features                                                                                                                                                                                                                 | You are right that many of the features have
native resolutions that are lower than 500                                                                                                                                                                                                                                                                                                                                                                                                                                                                                                                                                                                                                                                                                                                                                                                                                                                                                                                            |

| not available at 500mthis means that only the higher resolution
(MODIS based) features vary from pixel to
pixel, while the temperature estimate
(which is ERA5-Land derived at 0.10°)
may be the same in many adjacent 500-
meter pixels. As some of the features (like
FTC) are different, the EF estimate will still
be different between 500-meter pixels.The temporal resolution in the models on
the other hand is mostly driven by the
ERA5-land-derived variations. The FTC
retrievals will be the same the entire year
while differences in for instance VPD,
temperature and FWI make the EF estimate
vary each day.We added the following text (P7 L18):
"When computing to a higher resolution
e.g. 500-meter EFs, only the higher
resolution (MODIS-based) features exhibit
pixel-to-pixel variability, while
metorological conditions (derived from
ERA5-Land at 0.10°) may remain
consistent across many adjacent 500-meter
pixels. The presence of Modis-derived
features like FTC ensures that EF estimates
remain distinct between the grid cells. In
contrast, temporal resolution within the
models is more influenced by ERA5-Land-
derived fluctanions. While FTC criticvals
remain constant throughout the year,
variations in factors like VPD, temperature,
and EFVI cause EF estimates to fluctuate on
a daily basis."7, 18-24 It's still not explicitly clear how
fine fuel moisture was computed. For
instance, the daily cycle in fuel moisture
oud larage from 15% in early
AM to 5% in late afternoon, is the
fine fuel moisture adjusted to assume
afternon conditions for every fire in a 0.257, 18-24 It's still not explicitly clear how
fine fuel moisture adjusted to assume
afternon conditions for every fire in a 0.25                                                                                                                                                                                                                                          | often only have 0.25 res or in any case are   | meter. When computing the 500-meter EFs                                                                                                                                                                                                                                            |
|--------------------------------------------------------------------------------------------------------------------------------------------------------------------------------------------------------------------------------------------------------------------------------------------------------------------------------------------------------------------------------------------------------------------------------------------------------------------------------------------------------------------------------------------------------------------------------------------------------------------------------------------------------------------------------------------------------------------------------------------------------------------------------------------------------------------------------------------------------------------------------------------------------------------------------------------------------------------------------------------------------------------------------------------------------------------------------------------------------------------------------------------------------------------------------------------------------------------------------------------------------------------------------------------------------------------------------------------------------------------------------------------------------------------------------------------------------------------------------------------------------------------------------------------------------------------------------------------------------------------------------------------------------------------------------------------------------------------------------------------------------------------------------------------------------------------------------------------------------------------------------------------------------------------------------------------------------------------------------------------------------------------------------------------------------------------------------------------------------------------------|-----------------------------------------------|------------------------------------------------------------------------------------------------------------------------------------------------------------------------------------------------------------------------------------------------------------------------------------|
| (MODIS based) features vary from pixel to
pixel, while the temperature estimate
(which is ERA5-Land derived at 0.10°)
may be the same in many adjacent 500-
meter pixels. As some of the features (like
FTC) are different, the EF estimate will still
be different between 500-meter pixels.The temporal resolution in the models on
the other hand is mostly driven by the
ERA5-land-derived variations. The FTC
retrievals will be the same the entire year
while differences in for instance VPD,
temperature and FWI make the EF estimate
vary each day.We added the following text (P7 L18):
"When computing to a higher resolution
e.g. 500-meter EFs, only the higher
resolution (MODIS-based) features exhibit
pixel-to-pixel variability, while
meteorological conditions (derived from
ERA5-Land at 0.10°) may remain
consistent across many adjacent 500-meter
pixels. The presence of Modis-derived
features like FTC ensures that EF estimates
remain distinct between the grid cells. In
contrast, temporal resolution within the
models is more influenced by ERA5-Land-
derived fluctuations. While FTC retrievals7, 18-24 It's still not explicitly clear how
fine fuel moisture was computed. For
instance, the daily cycle in fuel moisture
could he important7, 18-24 It's still not explicitly clear how
fine fuel moisture was computed. For
instance, the daily cycle in fuel moisture
could range from 15% in early
AM to 5% in late afternoon, is the
fine fuel moisture diguted to assume
afternoor conditions for every fire in a 0.257. 18-24 It's still not explicitly clear how
fine fuel moisture adjusted to assume
afternoor conditions for every fire in a 0.257. 18-24 It's still not explicitly clear how
fine fuel moisture adjusted to assume
afternoor conditions for every fire in a 0.25 <td>not available at 500m</td> <td>this means that only the higher resolution</td>                                                                                  | not available at 500m                         | this means that only the higher resolution                                                                                                                                                                                                                                         |
|  <li>pixel, while the temperature estimate          <li>(which is ERA5-Land derived at 0.10°)</li> <li>may be the same in many adjacent 500-meter pixels. As some of the features (like FTC) are different, the EF estimate will still be different between 500-meter pixels.</li>  </li> <li>The temporal resolution in the models on the other hand is mostly driven by the ERA5-land-derived variations. The FTC retrievals will be the same the entire year while differences in for instance VPD, temperature and FWI make the EF estimate vary each day.</li> <li>We added the following text (P7 L18):          <li>"When computing to a higher resolution (MODIS-based) features exhibit pixel-to-pixel variability, while metcorological conditions (derived from ERA5-Land at 0.10°) may remain consistent across many adjacent 500-meter pixels. The presence of Modis-derived from ERA5-Land at 0.10°. May remain consistent across many adjacent 500-meter pixels. The presence of by ERA5-Land derived fluctuations. While FTC retrievals remain distinct between the grid cells. In contrast, temporal resolution within the models is more influenced by ERA5-Land-derived fluctuations. While FTC retrievals remain constant throughout the year, variations in factors like VPD, temperature, and FWI cause EF estimates to fluctuate on a daily basis."</li>  </li> <li>7, 18-24 It's still not explicitly clear how fine fuel moisture was computed. For instance, the daily cycle in fuel moisture or pixels in the afternoon, is FFMC an average near 10%? If active fire products suggest burning peaks in afternoon, is the fine fuel moisture adjusted to assume fuel moisture adjusted to assume fine fuel moisture is possed in acces. The products is product is product is product is product is product in the moisture is product is product is product is preading of the</li>                                                              |                                               | (MODIS based) features vary from pixel to                                                                                                                                                                                                                                          |
|  <li>(Which is ERAS-Land derived at 0.10°) may be the same in many adjacent 500-meter pixels. As some of the features (like FTC) are different, the EF estimate will still be different between 500-meter pixels.</li> <li>The temporal resolution in the models on the other hand is mostly driven by the ERAS-Land-derived variations. The FTC retrievals will be the same the entire year while differences in for instance VPD, temperature and FWI make the EF estimate vary each day.</li> <li>We added the following text (P7 L18): "When computing to a higher resolution e.g. 500-meter EFs, only the higher resolution (MODIS-based) features exhibit pixel-to-pixel variability, while meteorological conditions (derived from ERAS-Land at 0.10°) may remain consistent across many adjacent 500-meter pixels. The presence of Modis-derived freatures like FTC ensures that EF estimates remain distinct between the grid cells. In contrast, temporal resolution within the models is more influenced by ERAS-Land-derived freatures. While FTC retrievals remain constant throughout the year, variations in factors like VPD, temperature, and FWI cause EF estimates to fluctuate on a daily basis."</li> <li>7, 18-24 It's still not explicitly clear how fine fuel moisture was computed. For instance, the daily cycle in fuel moisture could range from 15% in early AM to 5% in late afternoon. Is FFMC an average near 10%? If active fire products suggest burning peaks in afternoon, is the fine fuel moisture adjusted to assume after on conditions for every fire in a 0.25</li>                                                                                                                                                                                                                                                                                                                                                                                                                                                                                                   |                                               | pixel, while the temperature estimate                                                                                                                                                                                                                                              |
|  <li>The vertex share in Haan agroup of the same in Haan agroup to both the same of the features (like FTC) are different, the EF estimate will still be different between 500-meter pixels.</li> <li>The temporal resolution in the models on the other hand is mostly driven by the ERA5-land-derived variations. The FTC retrievals will be the same the entire year while differences in for instance VPD, temperature and FWI make the EF estimate vary each day.</li> <li>We added the following text (P7 L18): "When computing to a higher resolution (MODIS-based) features exhibit pixel-to-pixel variability, while meteorological conditions (derived from ERA5-Land at 0.10°) may remain consistent across many adjacent 500-meter pixels. The presence of Modis-derived features like FTC ensures that EF estimates remain distinct between the grid cells. In contrast, temporal resolution within the models is more influenced by ERA5-Land-derived fluctuations. While FTC retrievals remain constant throughout the year, variations in factors like VPD, temperature, and FWI cause EF estimates to fluctuate on a daily basis."</li> <li>7, 18-24 It's still not explicitly clear how fine fuel moisture was computed. For instance, the daily cycle in fuel moisture could be important https://www.nwcg.gov/publications/pms425-1/weather-and-fuel-moisture. Typically fine fuel moisture adjusted to assume f</li> |                                               | (which is ERAS-Land derived at $0.10^{\circ}$ )
may be the same in many adjacent 500                                                                                                                                                                                            |
|  <li>FTC) are different, the EF estimate will still be different the terms of the other hand is mostly driven by the ERA5-land-derived variations. The FTC retrievals will be the same the entire year while differences in for instance VPD, temperature and FWI make the EF estimate vary each day.</li> <li>We added the following text (P7 L18): "When computing to a higher resolution c.g. 500-meter EFs, only the higher resolution (MODIS-based) features exhibit pixel-to-pixel variability, while meteorological conditions (derived from ERA5-Land at 0.10°) may remain consistent across many adjacent 500-meter pixels. The presence of Modis-derived ffeatures like FTC ensures that EF estimates remain distinct between the grid cells. In contrast, temporal resolution within the models is more influenced by ERA5-Land-derived ffeatures like FTC ensures that EF estimates remain distinct between the grid cells. In contrast, temporal resolution within the models is more influenced by ERA5-Land-derived fluctuations. While FTC retrievals remain constant throughout the year, variations in factors like VPD, temperature, and FWI cause EF estimates to fluctuate on a daily basis."</li> <li>7, 18-24 It's still not explicitly clear how fine fuel moisture could ange from 15% in early available is not a feature in figure 2. This code is a numeric rating of the moisture could range from 15% in early available is not a feature in figure 2. This code is a numeric rating of the moisture could range from 15% in early available is not a feature in figure 2. This code is a numeric rating of the moisture claver, 1-2 cm deep) and ranges from 2-101. It is based on Vitol et al (2019) who use the equations from vam targes from 2-101. It is based on Vitol et al (2019) who use the equations from vam and pricket (1987) combined with ECMWF model inputs. As this product is</li>                                                                                                                                                                            |                                               | may be the same in many adjacent 500-
meter pixels. As some of the features (like                                                                                                                                                                                               |
|  <li>be different between 500-meter pixels.</li> <li>be different between 500-meter pixels.</li> <li>The temporal resolution in the models on the other hand is mostly driven by the ERA5-land-derived variations. The FTC retrievals will be the same the entire year while differences in for instance VPD, temperature and FWI make the EF estimate vary each day.</li> <li>We added the following text (P7 L18): "When computing to a higher resolution e.g. 500-meter EFs, only the higher resolution (MODIS-based) features exhibit pixel-to-pixel variability, while meteorological conditions (derived from ERA5-Land at 0.10°) may remain consistent across many adjacent 500-meter pixels. The presence of Modis-derived features like FTC ensures that EF estimates remain distinct between the grid cells. In contrast, temporal resolution within the models is more influenced by ERA5-Land-derived fluctuations. While FTC retrievals remain constant throughout the year, variations in factors like VPD, temperature, and FWI cause EF estimates to fluctuate on a daily basis."</li> <li>7, 18-24 It's still not explicitly clear how fine fuel moisture was computed. For instance, the daily cycle in fuel moisture could be important https://www.nwcg.gov/publications/pms425-1/weather-and-fuel-moisture. Typically fine fuel moisture could range from 15% in early afternoon. Is FFMC an average near 10%? If active fire products suggest burning peaks in afternoon, is the fine fuel moisture adjusted to assume afternoon conditions for every fire in a 0.25</li>                                                                                                                                                                                                                                                                                                                                                                                                                                                                                                                 |                                               | FTC) are different, the EF estimate will still                                                                                                                                                                                                                                     |
|  <li>The temporal resolution in the models on the other hand is mostly driven by the ERA5-land-derived variations. The FTC retrievals will be the same the entire year while differences in for instance VPD, temperature and FWI make the EF estimate vary each day.</li> <li>We added the following text (P7 L18): "When computing to a higher resolution (MODIS-based) features exhibit pixel-to-pixel variability, while meteorological conditions (derived from ERA5-Land at 0.10°) may remain consistent across many adjacent 500-meter pixels. The presence of Modis-derived features like FTC ensures that EF estimates remain distinct between the grid cells. In contrast, temporal resolution within the models is more influenced by ERA5-Land derived fluctuations. While FTC retrievals remain constant throughout the year, variations in factors like VPD, temperature, and FWI cause EF estimates to fluctuate on a daily basis."</li> <li>7, 18-24 It's still not explicitly clear how fine fuel moisture acound ange from 15% in early AM to 5% in late afternoon. Is FFMC an average near 10%? If active fire products suggest burning peaks in afternoon, is the fuel moisture adjusted to assume afternoon conditions for every fire in a 0.25</li>                                                                                                                                                                                                                                                                                                                                                                                                                                                                                                                                                                                                                                                                                                                                                                                                                                       |                                               | be different between 500-meter pixels.                                                                                                                                                                                                                                             |
|  <li>We added the following text (P7 L18):
"When computing to a higher resolution
e.g. 500-meter EFs, only the higher
resolution (MODIS-based) features exhibit
pixel-to-pixel variability, while
meteorological conditions (derived from
ERA5-Land at 0.10°) may remain
consistent across many adjacent 500-meter
pixels. The presence of Modis-derived
features like FTC ensures that EF estimates
remain distinct between the grid cells. In
contrast, temporal resolution within the
models is more influenced by ERA5-Land-
derived fluctuations. While FTC retrievals
remain constant throughout the year,
variations in factors like VPD, temperature,
and FWI cause EF estimates to fluctuate on
a daily basis."</li> <li>7, 18-24 It's still not explicitly clear how
fine fuel moisture somputed. For
instance, the daily cycle in fuel moisture
could be important
https://www.nwcg.gov/publications/pms425-
1/weather-and-fuel-moisture. Typically fine
fuel moisture could range from 15% in early
AM to 5% in late afternoon. Is FFMC an
average near 10%? If active fire products
suggest burning peaks in afternoon, is the
fine fuel moisture adjusted to assume
afternoon conditions for every fire in a 0.25</li>                                                                                                                                                                                                                                                                                                                                                                                                                                                                                                                                                                                                                                                                                                                                                            |                                               | The temporal resolution in the models on
the other hand is mostly driven by the
ERA5-land-derived variations. The FTC
retrievals will be the same the entire year
while differences in for instance VPD,
temperature and FWI make the EF estimate
vary each day. |
|  <li>"When computing to a higher resolution
e.g. 500-meter EFs, only the higher
resolution (MODIS-based) features exhibit
pixel-to-pixel variability, while
meteorological conditions (derived from
ERA5-Land at 0.10°) may remain
consistent across many adjacent 500-meter
pixels. The presence of Modis-derived
features like FTC ensures that EF estimates
remain distinct between the grid cells. In
contrast, temporal resolution within the
models is more influenced by ERA5-Land-
derived fluctuations. While FTC retrievals
remain constant throughout the year,
variations in factors like VPD, temperature,
and FWI cause EF estimates to fluctuate on
a daily basis."</li> <li>7, 18-24 It's still not explicitly clear how
fine fuel moisture was computed. For
instance, the daily cycle in fuel moisture
could be important
https://www.nwcg.gov/publications/pms425-
1/weather-and-fuel-moisture. Typically fine
fuel moisture could range from 15% in early
AM to 5% in late afternoon. Is FFMC an
average near 10%? If active fire products
suggest burning peaks in afternoon, is the
fine fuel moisture adjusted to assume
afternoon conditions for every fire in a 0.25</li>                                                                                                                                                                                                                                                                                                                                                                                                                                                                                                                                                                                                                                                                                                                                                                                                  |                                               | We added the following text (P7 L18):                                                                                                                                                                                                                                              |
|  <li>e.g. 500-meter EFs, only the higher resolution (MODIS-based) features exhibit pixel-to-pixel variability, while meteorological conditions (derived from ERA5-Land at 0.10°) may remain consistent across many adjacent 500-meter pixels. The presence of Modis-derived features like FTC ensures that EF estimates remain distinct between the grid cells. In contrast, temporal resolution within the models is more influenced by ERA5-Land-derived fluctuations. While FTC retrievals remain constant throughout the year, variations in factors like VPD, temperature, and FWI cause EF estimates to fluctuate on a daily basis."</li> <li>7, 18-24 It's still not explicitly clear how fine fuel moisture was computed. For instance, the daily cycle in fuel moisture could be important https://www.nwcg.gov/publications/pms425-1/weather-and-fuel-moisture. Typically fine fuel moisture could range from 15% in early AM to 5% in late afternoon. Is FFMC an average near 10%? If active fire products suggest burning peaks in afternoon, is the fine fuel moisture adjusted to assume afternoon conditions for every fire in a 0.25</li>                                                                                                                                                                                                                                                                                                                                                                                                                                                                                                                                                                                                                                                                                                                                                                                                                                                                                                                                                       |                                               | "When computing to a higher resolution                                                                                                                                                                                                                                             |
|  <li>resolution (MODIS-based) reatures exhibit pixel-to-pixel variability, while meteorological conditions (derived from ERA5-Land at 0.10°) may remain consistent across many adjacent 500-meter pixels. The presence of Modis-derived features like FTC ensures that EF estimates remain distinct between the grid cells. In contrast, temporal resolution within the models is more influenced by ERA5-Land-derived fluctuations. While FTC retrievals remain constant throughout the year, variations in factors like VPD, temperature, and FWI cause EF estimates to fluctuate on a daily basis."</li> <li>7, 18-24 It's still not explicitly clear how fine fuel moisture was computed. For instance, the daily cycle in fuel moisture could be important https://www.nwcg.gov/publications/pms425-1/weather-and-fuel-moisture. Typically fine fuel moisture could range from 15% in early AM to 5% in late afternoon. Is FFMC an average near 10%? If active fire products suggest burning peaks in afternoon, is the fine fuel moisture adjusted to assume afternoon conditions for every fire in a 0.25</li>                                                                                                                                                                                                                                                                                                                                                                                                                                                                                                                                                                                                                                                                                                                                                                                                                                                                                                                                                                                           |                                               | e.g. 500-meter EFs, only the higher                                                                                                                                                                                                                                                |
|  <li>7, 18-24 It's still not explicitly clear how fine fuel moisture could be important https://www.nwcg.gov/publications/pms425-l/weather-and-fuel-moisture. Typically fine fuel moisture could range from 15% in early AM to 5% in late afternoon, is the fine fuel moisture adjusted to assume afternoon conditions for every fire in a 0.25</li> <li>Fire Fuel Moister Cote (1987) combined with atternoon conditions for every fire in a 0.25</li> <li>Fire Fuel Moister (1987) combined with ECMWF model inputs. As this product is</li>                                                                                                                                                                                                                                                                                                                                                                                                                                                                                                                                                                                                                                                                                                                                                                                                                                                                                                                                                                                                                                                                                                                                                                                                                                                                                                                                                                                                                                                                                                                                                                  |                                               | resolution (MODIS-based) features exhibit                                                                                                                                                                                                                                          |
|  <li>ERA5-Land at 0.10°) may remain
consistent across many adjacent 500-meter
pixels. The presence of Modis-derived
features like FTC ensures that EF estimates
remain distinct between the grid cells. In
contrast, temporal resolution within the
models is more influenced by ERA5-Land-
derived fluctuations. While FTC retrievals
remain constant throughout the year,
variations in factors like VPD, temperature,
and FWI cause EF estimates to fluctuate on
a daily basis."</li> <li>7, 18-24 It's still not explicitly clear how
fine fuel moisture was computed. For
instance, the daily cycle in fuel moisture
could be important</li> <li>https://www.nwcg.gov/publications/pms425-
1/weather-and-fuel-moisture. Typically fine
fuel moisture could range from 15% in early
AM to 5% in late afternoon. Is FFMC an
average near 10%? If active fire products
suggest burning peaks in afternoon, is the
fine fuel moisture adjusted to assume
afternoon conditions for every fire in a 0.25</li>                                                                                                                                                                                                                                                                                                                                                                                                                                                                                                                                                                                                                                                                                                                                                                                                                                                                                                                                                                                                                |                                               | meteorological conditions (derived from                                                                                                                                                                                                                                            |
|  <li>consistent across many adjacent 500-meter pixels. The presence of Modis-derived features like FTC ensures that EF estimates remain distinct between the grid cells. In contrast, temporal resolution within the models is more influenced by ERA5-Land-derived fluctuations. While FTC retrievals remain constant throughout the year, variations in factors like VPD, temperature, and FWI cause EF estimates to fluctuate on a daily basis."</li> <li>7, 18-24 It's still not explicitly clear how fine fuel moisture was computed. For instance, the daily cycle in fuel moisture could be important</li> <li>https://www.nwcg.gov/publications/pms425-1/weather-and-fuel-moisture. Typically fine fuel moisture could range from 15% in early AM to 5% in late afternoon. Is FFMC an average near 10%? If active fire products suggest burning peaks in afternoon, is the fine fuel moisture adjusted to assume afternoon conditions for every fire in a 0.25</li>                                                                                                                                                                                                                                                                                                                                                                                                                                                                                                                                                                                                                                                                                                                                                                                                                                                                                                                                                                                                                                                                                                                                     |                                               | ERA5-Land at 0.10°) may remain                                                                                                                                                                                                                                                     |
|  <li>pixels. The presence of Modis-derived features like FTC ensures that EF estimates remain distinct between the grid cells. In contrast, temporal resolution within the models is more influenced by ERA5-Land-derived fluctuations. While FTC retrievals remain constant throughout the year, variations in factors like VPD, temperature, and FWI cause EF estimates to fluctuate on a daily basis."</li> <li>7, 18-24 It's still not explicitly clear how fine fuel moisture was computed. For instance, the daily cycle in fuel moisture could be important https://www.nwcg.gov/publications/pms425-1/weather-and-fuel-moisture. Typically fine fuel moisture could range from 15% in early AM to 5% in late afternoon. Is FFMC an average near 10%? If active fire products suggest burning peaks in afternoon, is the fine fuel moisture adjusted to assume afternoon conditions for every fire in a 0.25</li>                                                                                                                                                                                                                                                                                                                                                                                                                                                                                                                                                                                                                                                                                                                                                                                                                                                                                                                                                                                                                                                                                                                                                                                        |                                               | consistent across many adjacent 500-meter                                                                                                                                                                                                                                          |
|  <li>remain distinct between the grid cells. In contrast, temporal resolution within the models is more influenced by ERA5-Land-derived fluctuations. While FTC retrievals remain constant throughout the year, variations in factors like VPD, temperature, and FWI cause EF estimates to fluctuate on a daily basis."</li> <li>7, 18-24 It's still not explicitly clear how fine fuel moisture was computed. For instance, the daily cycle in fuel moisture could be important https://www.nwcg.gov/publications/pms425-1/weather-and-fuel-moisture. Typically fine fuel moisture could range from 15% in early AM to 5% in late afternoon. Is FFMC an average near 10%? If active fire products suggest burning peaks in afternoon, is the fine fuel moisture adjusted to assume afternoon conditions for every fire in a 0.25</li>                                                                                                                                                                                                                                                                                                                                                                                                                                                                                                                                                                                                                                                                                                                                                                                                                                                                                                                                                                                                                                                                                                                                                                                                                                                                          |                                               | pixels. The presence of Modis-derived
features like FTC ensures that FF estimates                                                                                                                                                                                               |
|  <li>contrast, temporal resolution within the models is more influenced by ERA5-Land-derived fluctuations. While FTC retrievals remain constant throughout the year, variations in factors like VPD, temperature, and FWI cause EF estimates to fluctuate on a daily basis."</li> <li>7, 18-24 It's still not explicitly clear how fine fuel moisture was computed. For instance, the daily cycle in fuel moisture could be important https://www.nwcg.gov/publications/pms425-1/weather-and-fuel-moisture. Typically fine fuel moisture could range from 15% in early AM to 5% in late afternoon. Is FFMC an average near 10%? If active fire products suggest burning peaks in afternoon, is the fine fuel moisture adjusted to assume afternoon conditions for every fire in a 0.25</li>                                                                                                                                                                                                                                                                                                                                                                                                                                                                                                                                                                                                                                                                                                                                                                                                                                                                                                                                                                                                                                                                                                                                                                                                                                                                                                                     |                                               | remain distinct between the grid cells. In                                                                                                                                                                                                                                         |
|  <li>models is more influenced by ERA5-Land-derived fluctuations. While FTC retrievals remain constant throughout the year, variations in factors like VPD, temperature, and FWI cause EF estimates to fluctuate on a daily basis."</li> <li>7, 18-24 It's still not explicitly clear how fine fuel moisture was computed. For instance, the daily cycle in fuel moisture could be important https://www.nwcg.gov/publications/pms425-1/weather-and-fuel-moisture. Typically fine fuel moisture could range from 15% in early AM to 5% in late afternoon. Is FFMC an average near 10%? If active fire products suggest burning peaks in afternoon, is the fine fuel moisture adjusted to assume afternoon conditions for every fire in a 0.25</li>                                                                                                                                                                                                                                                                                                                                                                                                                                                                                                                                                                                                                                                                                                                                                                                                                                                                                                                                                                                                                                                                                                                                                                                                                                                                                                                                                              |                                               | contrast, temporal resolution within the                                                                                                                                                                                                                                           |
| derived fluctuations. While FTC retrievals
remain constant throughout the year,
variations in factors like VPD, temperature,
and FWI cause EF estimates to fluctuate on
a daily basis."7, 18-24 It's still not explicitly clear how
fine fuel moisture was computed. For
instance, the daily cycle in fuel moisture
could be importantIf understood correctly, you refer to
the Fine Fuel Moisture Code (FFMC),
which is included as a part of the FWI but
as a standalone variable is not a feature in
figure 2. This code is a numeric rating of
the moisture could range from 15% in early
AM to 5% in late afternoon. Is FFMC an
average near 10%? If active fire products
suggest burning peaks in afternoon, is the
fine fuel moisture adjusted to assume
afternoon conditions for every fire in a 0.25Utations. While FTC retrievals
remain constant throughout the year,
variations in factors like VPD, temperature,
and FWI cause EF estimates to fluctuate on
a daily basis."                                                                                                                                                                                                                                                                                                                                                                                                                                                                                                                                                                                                                                                                                                                                                                                                                                                                                                                                                                                                                                                                  |                                               | models is more influenced by ERA5-Land-                                                                                                                                                                                                                                            |
| 7, 18-24 It's still not explicitly clear how
fine fuel moisture was computed. For
instance, the daily cycle in fuel moisture
could be importantIf understood correctly, you refer to
the Fine Fuel Moisture Code (FFMC),
which is included as a part of the FWI but
as a standalone variable is not a feature in
figure 2. This code is a numeric rating of
the moisture content of litter and other
cured fine fuel moisture adjusted to assume
afternoon conditions for every fire in a 0.25remain constant throughout the year,
variations in factors like VPD, temperature,
and FWI cause EF estimates to fluctuate on
a daily basis."7, 18-24 It's still not explicitly clear how
fine fuel moisture was computed. For
instance, the daily cycle in fuel moisture
could be important
https://www.nwcg.gov/publications/pms425-
1/weather-and-fuel-moisture. Typically fine
fuel moisture could range from 15% in early
AM to 5% in late afternoon. Is FFMC an
average near 10%? If active fire products
suggest burning peaks in afternoon, is the
fine fuel moisture adjusted to assume
afternoon conditions for every fire in a 0.25                                                                                                                                                                                                                                                                                                                                                                                                                                                                                                                                                                                                                                                                                                                                                                                                                                                                                                      |                                               | derived fluctuations. While FTC retrievals                                                                                                                                                                                                                                         |
|  <li>and FWI cause EF estimates to fluctuate on a daily basis."</li> <li>7, 18-24 It's still not explicitly clear how fine fuel moisture was computed. For instance, the daily cycle in fuel moisture could be important https://www.nwcg.gov/publications/pms425-1/weather-and-fuel-moisture. Typically fine fuel moisture could range from 15% in early AM to 5% in late afternoon. Is FFMC an average near 10%? If active fire products suggest burning peaks in afternoon, is the fine fuel moisture adjusted to assume afternoon conditions for every fire in a 0.25</li> <li>Intervention in the or to be the product of the set of the set of the set of the fuel moisture could range from 15% in early AM to 5% in late afternoon. Is FFMC an average near 10%? If active fire products suggest burning peaks in afternoon, is the fine fuel moisture adjusted to assume afternoon conditions for every fire in a 0.25</li>                                                                                                                                                                                                                                                                                                                                                                                                                                                                                                                                                                                                                                                                                                                                                                                                                                                                                                                                                                                                                                                                                                                                                                            |                                               | variations in factors like VPD, temperature.                                                                                                                                                                                                                                       |
| a daily basis."7, 18-24 It's still not explicitly clear how
fine fuel moisture was computed. For
instance, the daily cycle in fuel moisture
could be importantIf understood correctly, you refer to
the Fine Fuel Moisture Code (FFMC),
which is included as a part of the FWI but
as a standalone variable is not a feature in
figure 2. This code is a numeric rating of
the moisture content of litter and other
cured fine fuels occupying the first fuel bed
layers (surface layer, 1-2 cm deep) and
ranges from 2-101. It is based on Vitolo et
al (2019) who use the equations from van
Wagner and Picket (1987) combined with
ECMWF model inputs. As this product is                                                                                                                                                                                                                                                                                                                                                                                                                                                                                                                                                                                                                                                                                                                                                                                                                                                                                                                                                                                                                                                                                                                                                                                                                                                                                                                                                                                                   |                                               | and FWI cause EF estimates to fluctuate on                                                                                                                                                                                                                                         |
| 7, 18-24 It's still not explicitly clear how
fine fuel moisture was computed. For
instance, the daily cycle in fuel moisture
could be importantIf understood correctly, you refer to
the Fine Fuel Moisture Code (FFMC),
which is included as a part of the FWI but
as a standalone variable is not a feature in
figure 2. This code is a numeric rating of
the moisture content of litter and other
cured fine fuels occupying the first fuel bed
layers (surface layer, 1-2 cm deep) and
ranges from 2-101. It is based on Vitolo et
al (2019) who use the equations from van
Wagner and Picket (1987) combined with
ECMWF model inputs. As this product is                                                                                                                                                                                                                                                                                                                                                                                                                                                                                                                                                                                                                                                                                                                                                                                                                                                                                                                                                                                                                                                                                                                                                                                                                                                                                                                                                                                                                  |                                               | a daily basis."                                                                                                                                                                                                                                                                    |
| interfuer moisture was computed. For
instance, the daily cycle in fuel moisture
could be important
https://www.nwcg.gov/publications/pms425-
1/weather-and-fuel-moisture. Typically fine
fuel moisture could range from 15% in early
AM to 5% in late afternoon. Is FFMC an
average near 10%? If active fire products
suggest burning peaks in afternoon, is the
fine fuel moisture adjusted to assume
afternoon conditions for every fire in a 0.25                                                                                                                                                                                                                                                                                                                                                                                                                                                                                                                                                                                                                                                                                                                                                                                                                                                                                                                                                                                                                                                                                                                                                                                                                                                                                                                                                                                                                                                                                                                                                                                                                                       | 7, 18-24 It's still not explicitly clear how  | If understood correctly, you refer to
the Fine Fuel Meisture Code (FEMC)                                                                                                                                                                                                        |
| could be important
https://www.nwcg.gov/publications/pms425-
1/weather-and-fuel-moisture. Typically fine
fuel moisture could range from 15% in early
AM to 5% in late afternoon. Is FFMC an
average near 10%? If active fire products
suggest burning peaks in afternoon, is the
fine fuel moisture adjusted to assume
afternoon conditions for every fire in a 0.25                                                                                                                                                                                                                                                                                                                                                                                                                                                                                                                                                                                                                                                                                                                                                                                                                                                                                                                                                                                                                                                                                                                                                                                                                                                                                                                                                                                                                                                                                                                                                                                                                                                                                                                             | instance, the daily cycle in fuel moisture    | which is included as a part of the FWI but                                                                                                                                                                                                                                         |
| https://www.nwcg.gov/publications/pms425-
1/weather-and-fuel-moisture. Typically fine
fuel moisture could range from 15% in early
AM to 5% in late afternoon. Is FFMC an
average near 10%? If active fire products
suggest burning peaks in afternoon, is the
fine fuel moisture adjusted to assume
afternoon conditions for every fire in a 0.25                                                                                                                                                                                                                                                                                                                                                                                                                                                                                                                                                                                                                                                                                                                                                                                                                                                                                                                                                                                                                                                                                                                                                                                                                                                                                                                                                                                                                                                                                                                                                                                                                                                                                                                                                   | could be important                            | as a standalone variable is not a feature in                                                                                                                                                                                                                                       |
| 1/weather-and-fuel-moisture. Typically fine
fuel moisture could range from 15% in early
AM to 5% in late afternoon. Is FFMC an
average near 10%? If active fire products
suggest burning peaks in afternoon, is the
fine fuel moisture adjusted to assume
afternoon conditions for every fire in a 0.25the moisture content of litter and other
cured fine fuels occupying the first fuel bed
layers (surface layer, 1-2 cm deep) and
ranges from 2-101. It is based on Vitolo et
al (2019) who use the equations from van
Wagner and Picket (1987) combined with
ECMWF model inputs. As this product is                                                                                                                                                                                                                                                                                                                                                                                                                                                                                                                                                                                                                                                                                                                                                                                                                                                                                                                                                                                                                                                                                                                                                                                                                                                                                                                                                                                                                                                                             | https://www.nwcg.gov/publications/pms425-     | figure 2. This code is a numeric rating of                                                                                                                                                                                                                                         |
| tuel moisture could range from 15% in early
AM to 5% in late afternoon. Is FFMC an
average near 10%? If active fire products
suggest burning peaks in afternoon, is the
fine fuel moisture adjusted to assume
afternoon conditions for every fire in a 0.25cured fine fuels occupying the first fuel bed
layers (surface layer, 1-2 cm deep) and
ranges from 2-101. It is based on Vitolo et
al (2019) who use the equations from van
Wagner and Picket (1987) combined with
ECMWF model inputs. As this product is                                                                                                                                                                                                                                                                                                                                                                                                                                                                                                                                                                                                                                                                                                                                                                                                                                                                                                                                                                                                                                                                                                                                                                                                                                                                                                                                                                                                                                                                                                                                                                        | 1/weather-and-fuel-moisture. Typically fine   | the moisture content of litter and other                                                                                                                                                                                                                                           |
| Aive to 5% in face alternoon. Is FFINC an
average near 10%? If active fire products
suggest burning peaks in afternoon, is the
fine fuel moisture adjusted to assume
afternoon conditions for every fire in a 0.25Tayers (surface layer, 1-2 cm deep) and
ranges from 2-101. It is based on Vitolo et
al (2019) who use the equations from van
Wagner and Picket (1987) combined with
ECMWF model inputs. As this product is                                                                                                                                                                                                                                                                                                                                                                                                                                                                                                                                                                                                                                                                                                                                                                                                                                                                                                                                                                                                                                                                                                                                                                                                                                                                                                                                                                                                                                                                                                                                                                                                                                                                     | tuel moisture could range from 15% in early   | cured fine fuels occupying the first fuel bed                                                                                                                                                                                                                                      |
| suggest burning peaks in afternoon, is the
fine fuel moisture adjusted to assume
afternoon conditions for every fire in a 0.25 ECMWF model inputs. As this product is                                                                                                                                                                                                                                                                                                                                                                                                                                                                                                                                                                                                                                                                                                                                                                                                                                                                                                                                                                                                                                                                                                                                                                                                                                                                                                                                                                                                                                                                                                                                                                                                                                                                                                                                                                                                                                                                                                                                              | average near 10%? If active fire products     | ranges from 2-101 It is based on Vitolo et                                                                                                                                                                                                                                         |
| fine fuel moisture adjusted to assume
afternoon conditions for every fire in a 0.25 Wagner and Picket (1987) combined with
ECMWF model inputs. As this product is                                                                                                                                                                                                                                                                                                                                                                                                                                                                                                                                                                                                                                                                                                                                                                                                                                                                                                                                                                                                                                                                                                                                                                                                                                                                                                                                                                                                                                                                                                                                                                                                                                                                                                                                                                                                                                                                                                                                                  | suggest burning peaks in afternoon, is the    | al (2019) who use the equations from van                                                                                                                                                                                                                                           |
| afternoon conditions for every fire in a 0.25   ECMWF model inputs. As this product is                                                                                                                                                                                                                                                                                                                                                                                                                                                                                                                                                                                                                                                                                                                                                                                                                                                                                                                                                                                                                                                                                                                                                                                                                                                                                                                                                                                                                                                                                                                                                                                                                                                                                                                                                                                                                                                                                                                                                                                                                                   | fine fuel moisture adjusted to assume         | Wagner and Picket (1987) combined with                                                                                                                                                                                                                                             |
|                                                                                                                                                                                                                                                                                                                                                                                                                                                                                                                                                                                                                                                                                                                                                                                                                                                                                                                                                                                                                                                                                                                                                                                                                                                                                                                                                                                                                                                                                                                                                                                                                                                                                                                                                                                                                                                                                                                                                                                                                                                                                                                          | afternoon conditions for every fire in a 0.25 | ECMWF model inputs. As this product is                                                                                                                                                                                                                                             |
| available daily, we did not consider the diurnal cycle of this parameter                                                                                                                                                                                                                                                                                                                                                                                                                                                                                                                                                                                                                                                                                                                                                                                                                                                                                                                                                                                                                                                                                                                                                                                                                                                                                                                                                                                                                                                                                                                                                                                                                                                                                                                                                                                                                                                                                                                                                                                                                                                 | aegree gria box?                              | available daily, we did not consider the diurnal cycle of this parameter                                                                                                                                                                                                           |

|                                                                                                                                                                                                                                                                                                                                                     |  <li>Vitolo, C., Di Giuseppe, F., Krzeminski, B. and San-Miguel-ayanz, J.: Data descriptor:
A 1980–2018 global fire danger re-analysis dataset for the Canadian fire weather indices, Sci. Data, 6, 1–10, doi:10.1038/sdata.2019.32, 2019.</li> <li>Van Wagner, C. E. & Forest, P. Development and structure of the canadian forest fire weather index system. Can. For. Serv. Forestry Tech. Rep. 35, (1987).</li> <li>It should be noted that this code did exhibit strong correlation with the fine fuel moisture contents we measured in the fuel (Table 4).</li>  |
|-----------------------------------------------------------------------------------------------------------------------------------------------------------------------------------------------------------------------------------------------------------------------------------------------------------------------------------------------------|-----------------------------------------------------------------------------------------------------------------------------------------------------------------------------------------------------------------------------------------------------------------------------------------------------------------------------------------------------------------------------------------------------------------------------------------------------------------------------------------------------------------------------------------------------------------------------------------|
| 7, 24 So here is where we learn that this
study is taking us from one static EF or an
assigned EF for EDS and LDS to monthly
EFs, but not higher?                                                                                                                                                                                          | For this global assessment over the 2002-
2016 period, we indeed calculated EFs on a
monthly basis. We have more recently
calculated global EFs on a daily basis
which will be made publicly available
alongside GFED5. The main temporal
patterns however will most likely be very
similar to those presented here. As
mentioned in the response to point 1, this is
now mentioned in the conclusions section.                                                                                                                                              |
| 7, 25 EFs, MCE, and DM consumed are all
impacted by the environment. Does DM
consumed vary based on these features or is
that built in to GFED already?                                                                                                                                                                                    | In GFED the combustion completeness is
directly scaled (between set minimum and
maximum values and for different fuel
types) based on moisture. Other
environmental influences are assumed to be
accounted for by NDVI which is used in the
light-use efficiency model to compute
carbon uptake. In future GFED values we
aim to build-in MCE and EFs based on this
work.                                                                                                                                                                                    |
| 8, 11 Not required but possible that this
section could be a place to remind less
experienced readers of the historical "one
EF" motivation for simply reporting a study
average measured on a big plane with lots of
instruments and a fast speed to access a lot
of fires and operating from a base at the
peak season/area. | We are very sorry if the reviewer feels that
we underrated the numerous previous
airplane and field measurements. That was
not our intention, we fully realize how
complicated and laborious these campaigns
are which yield information on a much
larger range of species and insights into
atmospheric chemistry than our work can
ever do.                                                                                                                                                                                                                   |
|                                                                                                                                                                                                                                                                                                                                                     | In the introduction we added (P3 L9):
"While lacking the extensive species
coverage and precision instruments found                                                                                                                                                                                                                                                                                                                                                                                                                                                               |

| 8, 15 add any impact on MCE?                                                                                                                                                                                      | in advanced aircraft campaigns, these UAS
measurements can effectively focus on
particular vegetation types, facilitating the
connection between ground conditions and
emissions."
We changed the sentence to (P8 L21): "EFs
of CO and CH 4 were lower (i.e. higher
MCE) in xeric open savannas compared to
woodland savannas."                                                                                                                                                             |
|-------------------------------------------------------------------------------------------------------------------------------------------------------------------------------------------------------------------|--------------------------------------------------------------------------------------------------------------------------------------------------------------------------------------------------------------------------------------------------------------------------------------------------------------------------------------------------------------------------------------------------------------------------------------------------------------------------------------------------------------------------------|
| 8, 18-19 isn't DM-consumed more important than burned area here?                                                                                                                                                  | That's correct. In relation to the DM
consumed, this sample bias would be even
larger. We have changed the sentence to
(P8 L23): "However, this may be largely
attributable to the fact that xeric savannas
were overly represented in our
measurements in terms of annual biomass
consumption (i.e. sample bias)"                                                                                                                                                                                        |
| 8, 28 May have asked in round one, but how
does measured fuel consumption compare to
the fuel consumption predicted by GFED
FC?                                                                          | We did not use these plots yet to compare
against the GFED4s modelled combustion
completeness. However, many of these
field measurements were used for the
calibration of the GFED 500-meter model
by van Wees et al. (2022). Figure 3 of that
paper shows a comparison of the fuel
consumption (and combustion
completeness) range found in our plots and
predicted by the GFED 500-meter model.
As mentioned, this is more a calibration
exercise rather than an independent
comparison. |
|                                                                                                                                                                                                                   | Van Wees, D., Van Der Werf, G. R.,
Randerson, J. T., Rogers, B. M., Chen, Y.,
Veraverbeke, S., Giglio, L. and Morton, D.
C.: Global biomass burning fuel
consumption and emissions at 500 m spatial
resolution based on the Global Fire
Emissions Database (GFED), Geosci.
Model Dev., 15(22), 8411–8437,
doi:10.5194/gmd-15-8411-2022, 2022.                                                                                                                                                          |
| 8, 30 "corresponding mixtures of fuel age" is
a bit nebulous unless you define fuel age. Do
you mean time since last burn, or something
else? This fuel age concept comes up again
on page 9, line 3. | We changed the sentence to (P9 L10): "For
some characteristics (e.g., the total fuel
load), it is important to note that the
average time since the last fire was not
necessarily equal between the listed
vegetation types. The higher fuel loads we
found in open savannas in Australia
compared to Botswana, may be partially
attributed to the longer fuel build-up."                                                                                                                              |

| 8, 35-38 Here you seem to clearly indicate
there was more RSC late in dry season, at
least in the humid savanna. On line 38, I
would move live foliage before the list of
RSC-prone fuels. Have you defined RSC?
You might find the Bertschi et al 2003 JGR
paper useful for that. Or one could term this
phenomenon as "post-frontal combustion." | We changed the text to (P9 L3): "As the
dry season progressed, there was a clear
shift towards the combustion of more live
foliage and Residual Smouldering
Combustion (RSC)-prone fuels like coarse
woody debris, stems and densely packed
litter, which after months of drought have
become more receptive to combustion. RSC
occurs after the passage of a flame front
and its emissions are not lofted by strong                                                                                                                                                                                                                                                                                          |
|-------------------------------------------------------------------------------------------------------------------------------------------------------------------------------------------------------------------------------------------------------------------------------------------------------------------------------------------------------------------------|------------------------------------------------------------------------------------------------------------------------------------------------------------------------------------------------------------------------------------------------------------------------------------------------------------------------------------------------------------------------------------------------------------------------------------------------------------------------------------------------------------------------------------------------------------------------------------------------------------------------------------------------------------------------------------------------------------------------------------------|
|                                                                                                                                                                                                                                                                                                                                                                         | fire-induced convection (Bertschi et al.,
2003). The late-LDS increase in the
consumption of live and course fuels
coincided with higher EFs for CO and CH 4
in the LDS."                                                                                                                                                                                                                                                                                                                                                                                                                                                                                                                                         |
| 9, 8 The dry Australian savannas had lower
EFs, but was DM-consumed also lower or
was that offset by longer fire-return
intervals?                                                                                                                                                                                                                             | As is listed in Table 3, both the fine fuel
load and the combustion completeness were
very high while these locations did not
contain much in terms of trees. Indeed, the
average time since last fire was also much
higher.                                                                                                                                                                                                                                                                                                                                                                                                                                                                                              |
|                                                                                                                                                                                                                                                                                                                                                                         | This combination of infrequent but intense
fires is related to the typical (hummock
style) growth pattern of the spinifex grasses
(e.g. Figure 2 below) that predominantly
carry the fire. They burn extremely hot and
at high MCE. However, it takes some years
for the fuel to become continuous enough
to carry.                                                                                                                                                                                                                                                                                                                                                                                                 |
| 9, 9 One thing was obvious during our field
work in Zambia. Within a radius of
settlements, much of the woody debris is
collected for household firewood. On the                                                                                                                                                                                               | Our measurements took place in the Kafue
national park where these practices are not
allowed.                                                                                                                                                                                                                                                                                                                                                                                                                                                                                                                                                                                                                                      |
| outskirts of Kaoma we saw people pushing
bicycles with logs tied to the seat and
handlebars headed to a local sawmill.
The landscape was clearly managed
differently within say 50 miles of Lusaka
compared to more remote areas.                                                                                                                        | We have added the following text to the
discussion (P15 L1): "Most of the fires
used to train the models were prescribed
fires set by scientists or park rangers in
protected areas in order to facilitate
collection of data pre and post burn on site.
It is common practice to extrapolate these
measurements in relatively undisturbed
savanna vegetation to the wider savanna.
Even though these protected natural areas
tend to burn more frequently, they represent
a minority of the area that is currently
modelled using savanna and grassland
emission factors by global inventories (e.g.
Fig.1). Most of this area is to some degree
affected by humans though cattle ranging. |

|                                                                                                                                                                                                                      |  <li>wood harvesting, slash and burn
agriculture, etc. This means fires in this
study may not always represent the burning
practices by local farmers and thus that
representativeness of our work for the
larger savanna area remains uncertain."</li> <li>In the conclusions we also added the
following sentence (P17 L22): "The
measured fires were predominantly
intentional burns conducted by scientists or
park rangers in protected areas for data
collection, and while these measurements
are extended to undisturbed savanna, the</li>  |
|----------------------------------------------------------------------------------------------------------------------------------------------------------------------------------------------------------------------|----------------------------------------------------------------------------------------------------------------------------------------------------------------------------------------------------------------------------------------------------------------------------------------------------------------------------------------------------------------------------------------------------------------------------------------------------------------------------------------------------------------------------------------------------------------------------------------------------------|
|                                                                                                                                                                                                                      | majority of the broader savanna used in
emission models is influenced by human
activities like cattle grazing and agriculture,
raising uncertainty about the
representativeness of the study's findings."                                                                                                                                                                                                                                                                                                                                                                                    |
| 10, 4 So to me, this means the MAE in MCE using static MCE is $\sim 1.60*0.006 = 0.010$ . If that's not right, maybe explain more?                                                                                   | Not exactly, that would be 60% compared
to the new MAE. Instead, the reduction is
given compared to the old value. Therefore,
the MAE in MCE using static MCE is ~
0.006 / (1-0.6) = 0.015                                                                                                                                                                                                                                                                                                                                                                                                   |
|                                                                                                                                                                                                                      | We changed the text to (P10 L9): "Overall,
we found that using only globally available
features covering a large (>20 year)
timespan, we could estimate the field-
measured MCE of the fires in the validation
set with a mean absolute error (MAE) of
0.006. Using the static MCE in GFED4
(MAE of 0.015 compared to the
measurements) as a baseline, this meant a
MAE reduction of 60%."                                                                                                                                                                                    |
| 10, 24 Soil moisture may act more like a
"long time-lag" (1000 hour) fuel?                                                                                                                                        | You are right that a time lag is to be
expected between soil and fuel moisture.
however, we also found that the soil
moisture variability was more spatial rather
than temporal in nature. This resulted in it
still being a strong predictor, even though
being a mediocre indicator of seasonality.                                                                                                                                                                                                                                                                                  |
| 10, 30 Again one wonders how common are
grid cells that are a mixture of savanna and
non-savanna? This may have been partly
addressed in the added text about
misclassification of savannas as cropland. | Given the fact that our measurements were
mostly taken in protected areas, the impact
of actual cropland on our measurements is
limited ("Cropland/Natural vegetation
mosaic" (6%) and "Croplands" (1%)).                                                                                                                                                                                                                                                                                                                                                                                    |

|                                                                                                                                                                                                                                                         | The IGBP classification classifies savannas                                                                                                                                                                                                                                                                                                                                                                                                                                                                                                                                                                                                                                                                                                                                                                                                                                                                                                                                                                                                                                                                                                                                                                                                                                                                                                                                                                                                                                                                                                                                                                                                                                                                                                                                                                                                                                                                                                                                                                                                                                                                                              |
|---------------------------------------------------------------------------------------------------------------------------------------------------------------------------------------------------------------------------------------------------------|------------------------------------------------------------------------------------------------------------------------------------------------------------------------------------------------------------------------------------------------------------------------------------------------------------------------------------------------------------------------------------------------------------------------------------------------------------------------------------------------------------------------------------------------------------------------------------------------------------------------------------------------------------------------------------------------------------------------------------------------------------------------------------------------------------------------------------------------------------------------------------------------------------------------------------------------------------------------------------------------------------------------------------------------------------------------------------------------------------------------------------------------------------------------------------------------------------------------------------------------------------------------------------------------------------------------------------------------------------------------------------------------------------------------------------------------------------------------------------------------------------------------------------------------------------------------------------------------------------------------------------------------------------------------------------------------------------------------------------------------------------------------------------------------------------------------------------------------------------------------------------------------------------------------------------------------------------------------------------------------------------------------------------------------------------------------------------------------------------------------------------------|
|                                                                                                                                                                                                                                                         | as "Lands covered with temporary crops                                                                                                                                                                                                                                                                                                                                                                                                                                                                                                                                                                                                                                                                                                                                                                                                                                                                                                                                                                                                                                                                                                                                                                                                                                                                                                                                                                                                                                                                                                                                                                                                                                                                                                                                                                                                                                                                                                                                                                                                                                                                                                   |
|                                                                                                                                                                                                                                                         | followed by harvest and a bare soil period                                                                                                                                                                                                                                                                                                                                                                                                                                                                                                                                                                                                                                                                                                                                                                                                                                                                                                                                                                                                                                                                                                                                                                                                                                                                                                                                                                                                                                                                                                                                                                                                                                                                                                                                                                                                                                                                                                                                                                                                                                                                                               |
|                                                                                                                                                                                                                                                         | (e.g., single and multiple cropping                                                                                                                                                                                                                                                                                                                                                                                                                                                                                                                                                                                                                                                                                                                                                                                                                                                                                                                                                                                                                                                                                                                                                                                                                                                                                                                                                                                                                                                                                                                                                                                                                                                                                                                                                                                                                                                                                                                                                                                                                                                                                                      |
|                                                                                                                                                                                                                                                         | systems). Note that perennial woody crops                                                                                                                                                                                                                                                                                                                                                                                                                                                                                                                                                                                                                                                                                                                                                                                                                                                                                                                                                                                                                                                                                                                                                                                                                                                                                                                                                                                                                                                                                                                                                                                                                                                                                                                                                                                                                                                                                                                                                                                                                                                                                                |
|                                                                                                                                                                                                                                                         | will be classified as the appropriate forest                                                                                                                                                                                                                                                                                                                                                                                                                                                                                                                                                                                                                                                                                                                                                                                                                                                                                                                                                                                                                                                                                                                                                                                                                                                                                                                                                                                                                                                                                                                                                                                                                                                                                                                                                                                                                                                                                                                                                                                                                                                                                             |
|                                                                                                                                                                                                                                                         | or shrub land cover type." We appreciate                                                                                                                                                                                                                                                                                                                                                                                                                                                                                                                                                                                                                                                                                                                                                                                                                                                                                                                                                                                                                                                                                                                                                                                                                                                                                                                                                                                                                                                                                                                                                                                                                                                                                                                                                                                                                                                                                                                                                                                                                                                                                                 |
|                                                                                                                                                                                                                                                         | that in some savanna areas with very                                                                                                                                                                                                                                                                                                                                                                                                                                                                                                                                                                                                                                                                                                                                                                                                                                                                                                                                                                                                                                                                                                                                                                                                                                                                                                                                                                                                                                                                                                                                                                                                                                                                                                                                                                                                                                                                                                                                                                                                                                                                                                     |
|                                                                                                                                                                                                                                                         | distinct seasonal signals, this distinction                                                                                                                                                                                                                                                                                                                                                                                                                                                                                                                                                                                                                                                                                                                                                                                                                                                                                                                                                                                                                                                                                                                                                                                                                                                                                                                                                                                                                                                                                                                                                                                                                                                                                                                                                                                                                                                                                                                                                                                                                                                                                              |
|                                                                                                                                                                                                                                                         | may be problematic. It would be very                                                                                                                                                                                                                                                                                                                                                                                                                                                                                                                                                                                                                                                                                                                                                                                                                                                                                                                                                                                                                                                                                                                                                                                                                                                                                                                                                                                                                                                                                                                                                                                                                                                                                                                                                                                                                                                                                                                                                                                                                                                                                                     |
|                                                                                                                                                                                                                                                         | interesting to see to what extent possible                                                                                                                                                                                                                                                                                                                                                                                                                                                                                                                                                                                                                                                                                                                                                                                                                                                                                                                                                                                                                                                                                                                                                                                                                                                                                                                                                                                                                                                                                                                                                                                                                                                                                                                                                                                                                                                                                                                                                                                                                                                                                               |
|                                                                                                                                                                                                                                                         | misclassifications affect our emission                                                                                                                                                                                                                                                                                                                                                                                                                                                                                                                                                                                                                                                                                                                                                                                                                                                                                                                                                                                                                                                                                                                                                                                                                                                                                                                                                                                                                                                                                                                                                                                                                                                                                                                                                                                                                                                                                                                                                                                                                                                                                                   |
|                                                                                                                                                                                                                                                         | estimations, albeit somewhat out of the                                                                                                                                                                                                                                                                                                                                                                                                                                                                                                                                                                                                                                                                                                                                                                                                                                                                                                                                                                                                                                                                                                                                                                                                                                                                                                                                                                                                                                                                                                                                                                                                                                                                                                                                                                                                                                                                                                                                                                                                                                                                                                  |
|                                                                                                                                                                                                                                                         | scope of this study.                                                                                                                                                                                                                                                                                                                                                                                                                                                                                                                                                                                                                                                                                                                                                                                                                                                                                                                                                                                                                                                                                                                                                                                                                                                                                                                                                                                                                                                                                                                                                                                                                                                                                                                                                                                                                                                                                                                                                                                                                                                                                                                     |
|                                                                                                                                                                                                                                                         |                                                                                                                                                                                                                                                                                                                                                                                                                                                                                                                                                                                                                                                                                                                                                                                                                                                                                                                                                                                                                                                                                                                                                                                                                                                                                                                                                                                                                                                                                                                                                                                                                                                                                                                                                                                                                                                                                                                                                                                                                                                                                                                                          |
|                                                                                                                                                                                                                                                         | In addition to the previously added classes                                                                                                                                                                                                                                                                                                                                                                                                                                                                                                                                                                                                                                                                                                                                                                                                                                                                                                                                                                                                                                                                                                                                                                                                                                                                                                                                                                                                                                                                                                                                                                                                                                                                                                                                                                                                                                                                                                                                                                                                                                                                                              |
|                                                                                                                                                                                                                                                         | and the classes in the Excel file, we added                                                                                                                                                                                                                                                                                                                                                                                                                                                                                                                                                                                                                                                                                                                                                                                                                                                                                                                                                                                                                                                                                                                                                                                                                                                                                                                                                                                                                                                                                                                                                                                                                                                                                                                                                                                                                                                                                                                                                                                                                                                                                              |
|                                                                                                                                                                                                                                                         | the previously mentioned section about the                                                                                                                                                                                                                                                                                                                                                                                                                                                                                                                                                                                                                                                                                                                                                                                                                                                                                                                                                                                                                                                                                                                                                                                                                                                                                                                                                                                                                                                                                                                                                                                                                                                                                                                                                                                                                                                                                                                                                                                                                                                                                               |
|                                                                                                                                                                                                                                                         | representativeness of protected areas to the                                                                                                                                                                                                                                                                                                                                                                                                                                                                                                                                                                                                                                                                                                                                                                                                                                                                                                                                                                                                                                                                                                                                                                                                                                                                                                                                                                                                                                                                                                                                                                                                                                                                                                                                                                                                                                                                                                                                                                                                                                                                                             |
|                                                                                                                                                                                                                                                         | discussion (P15 L1).                                                                                                                                                                                                                                                                                                                                                                                                                                                                                                                                                                                                                                                                                                                                                                                                                                                                                                                                                                                                                                                                                                                                                                                                                                                                                                                                                                                                                                                                                                                                                                                                                                                                                                                                                                                                                                                                                                                                                                                                                                                                                                                     |
| 11, 7&10 I think it was decided in response                                                                                                                                                                                                             | We have removed all references to "typical                                                                                                                                                                                                                                                                                                                                                                                                                                                                                                                                                                                                                                                                                                                                                                                                                                                                                                                                                                                                                                                                                                                                                                                                                                                                                                                                                                                                                                                                                                                                                                                                                                                                                                                                                                                                                                                                                                                                                                                                                                                                                               |
| to another reviewer that there's no such                                                                                                                                                                                                                | savanna" from the manuscript.                                                                                                                                                                                                                                                                                                                                                                                                                                                                                                                                                                                                                                                                                                                                                                                                                                                                                                                                                                                                                                                                                                                                                                                                                                                                                                                                                                                                                                                                                                                                                                                                                                                                                                                                                                                                                                                                                                                                                                                                                                                                                                            |
| thing as "typical savanna"?                                                                                                                                                                                                                             | 1                                                                                                                                                                                                                                                                                                                                                                                                                                                                                                                                                                                                                                                                                                                                                                                                                                                                                                                                                                                                                                                                                                                                                                                                                                                                                                                                                                                                                                                                                                                                                                                                                                                                                                                                                                                                                                                                                                                                                                                                                                                                                                                                        |
| 11, 25-26 There are no parentheses and two                                                                                                                                                                                                              | We added parentheses. We found that                                                                                                                                                                                                                                                                                                                                                                                                                                                                                                                                                                                                                                                                                                                                                                                                                                                                                                                                                                                                                                                                                                                                                                                                                                                                                                                                                                                                                                                                                                                                                                                                                                                                                                                                                                                                                                                                                                                                                                                                                                                                                                      |
| EFs have stdev of zero? The interannual                                                                                                                                                                                                                 | giving the numbers in 2 decimal precision                                                                                                                                                                                                                                                                                                                                                                                                                                                                                                                                                                                                                                                                                                                                                                                                                                                                                                                                                                                                                                                                                                                                                                                                                                                                                                                                                                                                                                                                                                                                                                                                                                                                                                                                                                                                                                                                                                                                                                                                                                                                                                |
| variability seems low at $< 1\%$ maybe?, and                                                                                                                                                                                                            | suggests the models to be more accurate                                                                                                                                                                                                                                                                                                                                                                                                                                                                                                                                                                                                                                                                                                                                                                                                                                                                                                                                                                                                                                                                                                                                                                                                                                                                                                                                                                                                                                                                                                                                                                                                                                                                                                                                                                                                                                                                                                                                                                                                                                                                                                  |
| certainly lower than real accuracy?                                                                                                                                                                                                                     | than they are. However, rounding to a                                                                                                                                                                                                                                                                                                                                                                                                                                                                                                                                                                                                                                                                                                                                                                                                                                                                                                                                                                                                                                                                                                                                                                                                                                                                                                                                                                                                                                                                                                                                                                                                                                                                                                                                                                                                                                                                                                                                                                                                                                                                                                    |
|                                                                                                                                                                                                                                                         | single decimal means the standard